

**1** **Anthropogenic Secondary Organic Aerosols Contribute Substantially to Air Pollution**
**2** **Mortality**

Benjamin A. Nault[1,2,*], Duseong S. Jo[1,2], Brian C. McDonald[2,3], Pedro Campuzano-Jost[1,2], Douglas A.
Day[1,2], Weiwei Hu[1,2,**], Jason C. Schroder[1,2,***], James Allan[4,5], Donald R. Blake[6], Manjula R.
Canagaratna[7], Hugh Coe[5], Matthew M. Coggon[2,3], Peter F. DeCarlo[8], Glenn S. Diskin[9], Rachel
Dunmore[10], Frank Flocke[11], Alan Fried[12], Jessica B. Gilman[3], Georgios Gkatzelis[2,3], Jacqui F. Hamilton[10],
Thomas F. Hanisco[13], Patrick L. Hayes[14], Daven K. Henze[15], Alma Hodzic[11,16], James Hopkins[10,17], Min
Hu[18], L. Greggory Huey[19], B. Thomas Jobson[20], William C. Kuster[3,****], Alastair Lewis[10,17], Meng Li[2,3], Jin
Liao[13,21], M. Omar Nawaz[15], Ilana B. Pollack[22], Jeffrey Peischl[2,3], Bernhard Rappenglück[23], Claire E.
Reeves[24], Dirk Richter[12], James M. Roberts[3], Thomas B. Ryerson[3], Min Shao[25], Jacob M. Sommers[14,26],
James Walega[12], Carsten Warneke[2,3], Petter Weibring[12], Glenn M. Wolfe[13,27], Dominique E. Young[5,*****],
Bin Yuan[25], Qiang Zhang[28], Joost A. de Gouw[1,2], and Jose L. Jimenez[1,2,+]
1. Department of Chemistry, University of Colorado, Boulder, Boulder, CO, USA
2. Cooperative Institute for Research in Environmental Sciences, Boulder, Colorado, USA
3. Chemical Sciences Division, NOAA Earth System Research Laboratory, Boulder, CO
4. National Centre for Atmospheric Sciences, School of Earth and Environmental Sciences, University of Manchester, Manchester, UK
5. Centre of Atmospheric Science, School of Earth and Environmental Sciences, University of Manchester, Manchester, UK
6. Department of Chemistry, University of California, Irvine, Irvine, CA, USA
7. Center for Aerosol and Cloud Chemistry, Aerodyne Research Inc., Billerica, MA, USA
8. Department of Environmental Health Engineering, Johns Hopkins University, Baltimore, MD, USA
9. NASA Langley Research Center, Hampton, Virginia, USA
10. Wolfson Atmospheric Chemistry Laboratories, Department of Chemistry, University of York, York, UK
11. Atmospheric Chemistry Observations and Modeling Laboratory, National Center for Atmospheric Research, Boulder, CO, USA
12. Institute of Arctic and Alpine Research, University of Colorado, Boulder, CO, USA
13. Atmospheric Chemistry and Dynamic Laboratory, NASA Goddard Space Flight Center, Greenbelt, MD, USA
14. Department of Chemistry, Université de Montréal, Montréal, QC, Canada
15. Department of Mechanical Engineering, University of Colorado, Boulder, Boulder, CO, USA
16. Laboratoires d'Aréologie, Université de Toulouse, CNRS, UPS, Toulouse, France
17. National Centre for Atmospheric Sciences, Department of Chemistry, University of York, York, UK
18. State Key Joint Laboratory of Environmental Simulation and Pollution Control, College of Environmental Sciences and Engineering, Peking
University, Beijing, China
19. School of Earth and Atmospheric Sciences, Georgia Institute of Technology, Atlanta, Georgia, USA
20. Laboratory for Atmospheric Research, Department of Civil and Environmental Engineering, Washington State University, Pullman, WA,
USA
21. Universities Space Research Association, GESTAR, Columbia, MD, USA
22. Department of Atmospheric Science, Colorado State University, Fort Collins, CO, USA
23. Department of Earth and Atmospheric Science, University of Houston, Houston, TX, USA
24. Centre for Ocean and Atmospheric Sciences, School of Environmental Sciences, University of East Anglia, Norwich, UK
25. Institute for Environmental and Climate Research, Jinan University, Guangzhou, China
26. Air Quality Research Division, Environment and Climate Change Canada, Toronto, Ontario, Canada
27. Joint Center for Earth Systems Technology, University of Maryland, Baltimore County, Baltimore, MD, USA
28. Ministry of Education Key Laboratory for Earth System Modeling, Department of Earth System Science, Tsinghua University, Beijing, China
*Now at Center for Aerosol and Cloud Chemistry, Aerodyne Research Inc., Billerica, MA, USA
**Now at State Key Laboratory at Organic Geochemistry, Guangzhou Institute of Geochemistry, Chinese Academy of Sciences, Guangzhou,
China
***Now at Colorado Department of Public Health and Environment, Denver, CO, USA
****Has retired and worked on this manuscript as an unaffiliated co-author.
*****Now at Air Quality Research Center, University of California, Davis, CA, USA
+Corresponding author: Jose L. Jimenez (jose.jimenez@colorado.edu)



## Abstract

Anthropogenic secondary organic aerosol (ASOA), formed from anthropogenic emissions of organic compounds, constitutes a substantial fraction of the mass of submicron aerosol in populated areas around the world and contributes to poor air quality and premature mortality. However, the precursor sources of ASOA are poorly understood, and there are large uncertainties in the health benefits that might accrue from reducing anthropogenic organic emissions. We show that the production of ASOA in 11 urban areas on three continents is strongly correlated with the anthropogenic reactivity of specific volatile organic compounds. The differences in ASOA production across different cities can be explained by differences in the emissions of aromatics and intermediate- and semi-volatile organic compounds, indicating the importance of controlling these ASOA precursors. With an improved modeling representation of ASOA driven by the observations, we attribute 340,000 $PM_{2.5}$ premature deaths per year to ASOA, which is over an order of magnitude higher than prior studies. A sensitivity case with a more recently proposed model for attributing mortality to $PM_{2.5}$ (the Global Exposure Mortality Model) results up to 900,000 deaths. A limitation of this study is the extrapolation from regions with detailed data to others where data is not available. Comprehensive air quality campaigns in the countries in South and Central America, Africa, South Asia, and the Middle East are needed for further progress in this area.



# 1. Introduction


Poor air quality is one of the leading causes of premature mortality worldwide (Cohen et
al., 2017; Landrigan et al., 2018). Roughly 95% of the world's population live in areas where
$PM_{2.5}$ (fine particulate matter with diameter smaller than 2.5 µm) exceeds the World Health
Organization's 10 µg m$^{-3}$ annual average guideline (Shaddick et al., 2018). This is especially true
for urban areas, where high population density is co-located with increased emissions of $PM_{2.5}$
and its gas-phase precursors from human activities. It is estimated that $PM_{2.5}$ leads to 3 to 4
million premature deaths per year, higher than the deaths associated with other air pollutants
(Cohen et al., 2017). More recent analysis using concentration-response relationships derived
from studies of populations exposure to high levels of ambient $PM_{2.5}$ suggest the global
premature death burden could be up to twice this value (Burnett et al., 2018).
The average measured chemical composition of submicron PM ($PM_1$, which typically
comprises most of $PM_{2.5}$ (Wang et al., 2015)) for various megacities, urban areas, and outflow
regions around the world is shown in Fig. 1. A substantial fraction of urban $PM_1$ is organic
aerosol (OA), which is composed of primary OA (POA, organic compounds emitted directly in
the particle phase) and secondary OA (SOA, formed from chemical reactions of precursor
organic gases). SOA is typically a factor of 2 to 3 higher than POA for these locations.
Understanding the gas-phase precursors of anthropogenic SOA (ASOA, defined as the
SOA formed from anthropogenic volatile organic compounds (AVOC) (de Gouw et al., 2005;
DeCarlo et al., 2010)) quantitatively is challenging (Hallquist et al., 2009). Though the
enhancement of ASOA is largest in large cities, these precursors and production of ASOA should
be important in any location impacted by anthropogenic emissions (e.g., Fig. 1). ASOA



comprises a wide range of condensable products generated by numerous chemical reactions
involving AVOC precursors (Hallquist et al., 2009; Hayes et al., 2015; Shrivastava et al., 2017).
These condensable products include intermediate volatile organic compounds (IVOCS, less
volatile than traditional VOCs and often not measured or considered (Robinson et al., 2007;
Hayes et al., 2015)) and semi volatile organic compounds (SVOCs, less volatile than IVOC and
similarly not measured or considered).

The main categories of gas-phase precursors that dominate ASOA have been the subject

of intensive research. Transportation-related emissions (e.g., tailpipe, evaporation, refueling)
were assumed to be the major precursors of ASOA, which was supported by field studies
(Parrish et al., 2009; Gentner et al., 2012; Warneke et al., 2012; Pollack et al., 2013). Yet, budget
closure of observed ASOA mass concentrations could not be achieved with
transportation-related VOCs (Ensberg et al., 2014). The contribution of urban-emitted biogenic
precursors to SOA in urban areas is typically small, and rather, the contribution of biogenic SOA
(BSOA) in urban areas is typically dominated by regionally advected SOA background (e.g.,
Hodzic et al., 2009, 2010a; Hayes et al., 2013; Janssen et al., 2017). BSOA is thought to
dominate globally (Hallquist et al., 2009), but as shown in Fig. 1, the contribution of BSOA (1%
to 20%) to urban concentrations, while often substantial, is typically smaller than that of ASOA
(17% to 39%) (see Sect. 2). Recent studies have indicated that emissions from volatile chemical
products (VCPs), defined as pesticides, coatings, inks, adhesives, personal care products, and
cleaning agents (McDonald et al., 2018), as well as cooking emissions (Hayes et al., 2015), are
important. While total amounts of ASOA precursors released in cities have dramatically declined
(largely due to three-way catalytic converters in cars (Warneke et al., 2012; Pollack et al., 2013;





Zhao et al., 2017; Khare and Gentner, 2018)), VCPs have not declined as quickly (Khare and
Gentner, 2018; McDonald et al., 2018). Besides a few cities in the US (Coggon et al., 2018;
Khare and Gentner, 2018; McDonald et al., 2018), extensive VCP emission quantification has
not yet been published.
Due to the uncertainty on the ASOA precursors and on the amount of ASOA formed
from them, the number of premature deaths associated with urban organic emissions is largely
unknown. Currently, most studies have not included ASOA realistically (e.g., Lelieveld et al.,
2015; Silva et al., 2016; Ridley et al., 2018) in source apportionment calculations of the
premature deaths associated with long-term exposure of $PM_{2.5}$. These models represented total
OA as non-volatile POA and "traditional" ASOA precursors (transportation-based VOCs), which
largely under-predict ASOA (Ensberg et al., 2014; Hayes et al., 2015; Nault et al., 2018;
Schroder et al., 2018) given that the current understanding is that POA is volatile and contributes
to ASOA mass concentration (e.g., Grieshop et al., 2009; Lu et al., 2018). As $PM_1$ and SOA
mass are highest in urban areas (Fig. 1), also shown in Jimenez et al. (2009), it is necessary to
quantify the amount and identify the sources of ASOA to target future emission standards that
will optimally improve air quality and the associated health impacts. As these emissions are from
human activities, they will contribute to SOA mass outside urban regions and to potential health
impacts outside urban regions as well.
Here, we investigate the factors that control ASOA using 11 major urban, including
megacities, field studies (Fig. 1 and Table 1). The empirical relationships and numerical models
are then used to quantify the attribution of premature mortality to ASOA around the world, using
the observations to improve the modeled representation of ASOA. The results provide insight



into the importance of ASOA to global premature mortality due to $PM_{2.5}$ and further
understanding the precursors and sources of ASOA in urban regions.

**2. Methods**
**2.1 Ambient Observations**
For values not previously reported in the literature (Table S4), observations taken
between 11:00 – 16:00 local time were used to determine the slopes of SOA versus
formaldehyde (HCHO) (Fig. S2), peroxy acetyl nitrate (PAN) (Fig. S3), and $O_x$ ($O_x = O_3 + NO_2$)
(Fig. S4). For CalNex, there was an approximate 48% difference between the two HCHO
measurements (Fig. S1). Therefore, the average between the two measurements were used in this
study, similar to what has been done in other studies for other gas-phase species (Bertram et al.,
2007). All linear fits, unless otherwise noted, use the orthogonal distance regression fitting
method (ODR).
For values in Table S4 through Table S8 not previously reported in the literature, the
following procedure was applied to determine the emissions ratios, similar to the methods of
Nault et al. (2018). An OH exposure ($OH_{exp}$ = [OH]×$\Delta$t), which is also the photochemical age
(PA), was estimated by using the ratio of $NO_x/NO_y$ (Eq. 1) or the ratio of
m+p-xylene/ethylbenzene (Eq. 2). For the m+p-xylene/ethylbenzene, the emission ratio
(Table S5) was determined by determining the average ratio during minimal photochemistry,
similar to prior studies (de Gouw et al., 2017). This was done for only one study, TexAQS 2000.
This method could be applied in that case as it was a ground campaign that operated both day
and night; therefore, a ratio at night could be determined when there was minimal loss of both



VOCs. The average emission ratio for the other VOCs was determined using Eq. 3 after the
$OH_{exp}$ was calculated in Eq. 1 or Eq. 2. The rate constants used for determining $OH_{exp}$ and
emission ratios are found in Table S11.
$$OH_{exp} = [OH] \times t = \ln\left(\frac{\left(\frac{[NO_x]}{[NO_y]}\right)}{k_{OH+NO_2}}\right)$$

Eq. 1

$$OH_{exp} = [OH] \times t = -\frac{1}{k_{m+p-xylene} - k_{ethylbenzene}} \times \ln\left(\frac{[m+p-xylene]_t}{[ethylbenzene]_t} - \frac{[m+p-xylene]_0}{[ethylbenzene]_0}\right)$$

Eq. 2

$$\frac{[VOC(i)]}{[CO]}(0) = -\frac{[VOC(i)]}{[CO]}(t) \times \left(1 - \frac{1}{exp\left(-k_i \times [OH] \times t\right)}\right) \times k_i + \frac{[VOC(i)]}{[CO]}(t) \times k_i$$

Eq. 3


**2.2 Error Analysis of Observations**

The errors that will be discussed here are in reference to Fig. 5 and Fig. 6 and Table S4

either come from the 1σ uncertainty in the slopes (the SOA versus $O_x$, HCHO, or PAN values) or
propagation of uncertainty in observations. For SOA, we estimate the 1σ uncertainty of ~15%,
which is lower than the typical 1σ uncertainty of the AMS (Bahreini et al., 2009) due to the
careful calibrations and excellent intercomparisons in the various campaigns (see Table 1 for
references for the AMS comparisons). For ΔCO, the largest uncertainty is associated with the
CO background (Hayes et al., 2013; Nault et al., 2018), and is estimated to be ~10% at 0.5
photochemical equivalent days (Hayes et al., 2013). The uncertainty in the emission ratios is
~10% (Wang et al., 2014; de Gouw et al., 2017); though, it may be higher for the values



calculated here (see above) due to the uncertainty in CO background, rate constants, and
photochemical age. Therefore, for Fig. 5a, the uncertainty in the y-values is 18% and the
uncertainty in the x-values is 10%. For Fig. 6, the uncertainty in the measurement is 21%.
Another potential source of uncertainty may stem from the fit of the data in Fig. 5a, as the
data point from Seoul (KORUS-AQ) could be impacting the fit due to the difference in its value
compared to the other locations. A sensitivity analysis, where one study was removed and a new
fit was derived, was conducted to determine the impact of any one study on the fit reported in
Fig. 5a (see Table S10). We find that though removing the Seoul data point increases the slope,
the value is still within the uncertainty and statistically significant at the 95% confidence
interval. Thus, the data from Seoul does not change the results and conclusions reported in this
study.

**2.3 Emission Inventories for Various Urban Areas around the World**

All BTEX (benzene, toluene, ethylbenze, and xylenes) and non-BTEX aromatic emissions
are shown in Table S5 (BTEX) or Table S8 (non-BTEX aromatics) and are described above. The
emission ratios are derived from ambient measurements utilizing photochemical aging
techniques (Nault et al., 2018).
Details of the emission inventories for cities in the US, for Beijing, and for London/UK
used here to estimate the IVOC:BTEX emission ratio (Fig. 2) and thus the IVOC emissions can
be found in SI Sect. 1 through 3. Briefly, emissions for the US are based on McDonald et al.
(2018), for China on the Multi-resolution Emission Inventory for China (MEIC) (Zhang et al.,
2009; Zheng et al., 2014, 2018; Liu et al., 2015; Li et al., 2017, 2019), and for the UK on the



National Atmospheric Emissions Inventory (NAEI) (EMEP/EEA, 2016). The IVOC:BTEX emission ratio from inventories are multiplied with the observed BTEX measured in urban air to estimate IVOCs emitted in each region (Table S5), including North America, Europe, and Asia. This ensures IVOC emissions used in our calculations properly reflect differences in mixtures of emission sources (e.g., mobile sources versus VCPs) that vary by continent for each field campaign. Additionally, we rely on inventories for estimating atmospheric abundances of IVOCs because it has been challenging to measure the full range of IVOC precursors that are emitted into urban air (Zhao et al., 2014, 2017; Lu et al., 2018). In particular, many of the IVOCs emitted from VCPs are oxygenated, which are challenging to measure using traditional gas chromatography-mass spectrometry (GC-MS) techniques. Oxygenated IVOCs may not elute completely through a non-polar column, and are likely underestimated (Zhao et al., 2014). The bottom-up IVOC:BTEX ratios for the US, Beijing, and UK are described in greater detail in SI Sect. S1 through S3. IVOC emissions are classified based on their vapor pressure (effective saturation concentration: $0.3 < C^* < 3 \times 10^6$ μg m$^{-3}$), with the vapor pressure estimated by the SIMPOL.1 model (Pankow and Asher, 2008). Unspeciated mass has been suggested as important SOA precursors from gasoline and diesel engines, and parameterized by n-tridecane and n-pentadecane, respectively (Jathar et al., 2014). For VCPs, the volatility distribution of VOCs is in-between that of gasoline and diesel fuel. Therefore, n-tetradecane was suggested as a surrogate for unspeciated mass of VCPs by McDonald et al. (2018).

Similar to IVOCs, the ability to measure the full range of SVOCs emitted into urban air is challenging. Therefore, we estimate SVOC emission ratios relative to POA mass concentrations (Table S9), as described by Ma et al. (2017). For the hydrocarbon-like portion, we used the



volatility distribution from Worton et al. (2014) to estimate SVOC, as this is associated with
fossil fuel emissions from transportation (Zhang et al., 2005). For the other POA, we used the
volatility distribution from Robinson et al. (2007), as this POA is typically cooking primary
aerosol.

Fig. 3 shows the calculated emission ratio versus saturation concentration (c*) for the

cities with emission inventories. The saturation concentration for SVOC was determined as part
of the estimation procedure discussed above. For IVOC, the emission ratios for the different
sources (gasoline, diesel, other fossil fuel sources, and VCP emissions) were split into the
volatility bins, as in McDonald et al. (2018). Finally, for BTEX and non-BTEX aromatics, and
other VOC emission ratios (see Fig. 3 for references for the other VOC emission ratios), CRC
(Rumble, 2019) or SIMPOL.1 (Pankow and Asher, 2008) (for estimating vapor pressures not in
CRC) was used to estimate the saturation concentrations.

**2.4 ASOA Budget Analysis of Ambient Observations**

To calculate the ASOA budget, we used the observed BTEX (Table S5) and non-BTEX

aromatic (Table S8) emission ratios, the emission inventories for IVOC (see above), and
estimated SVOCs from the primary OA emissions (see above). The methods to calculate ASOA
from emissions have been described in detail elsewhere (Hayes et al., 2015; Ma et al., 2017;
Schroder et al., 2018), and are briefly described here. All calculations described were conducted
with the KinSim v4.02 chemical kinetics simulator (Peng and Jimenez, 2019) within Igor Pro 7
(Lake Oswego, Oregon), and are summarized in Fig. S5. A typical average particle diameter for
urban environments of ~200 nm (Seinfeld and Pandis, 2006) is used to estimate the


condensational sink term for the partitioning of gas-to-particle, although condensation is always
fast compared to the experiment timescales. Further, we assume an average 250 g mol$^{-1}$ molar
mass for OA and an average SOA density of 1.4 g cm$^{-3}$ (Vaden et al., 2011; Kuwata et al., 2012).
Finally, all models are initialized with the campaign specific OA background (typically ~2 µg
sm$^{-3}$) and POA (Table S9) for partitioning of gases to the particle phase, and ran at the average
temperature for the campaign.
For the modeled VOCs (BTEX and non-BTEX aromatics), each species undergoes
temperature-dependent OH oxidation (Table S11), forming four SVOCs that partition between
gas- and particle-phase, using updated SOA yields that account for wall loss (Ma et al., 2017).
For IVOCs, the emission weighted SOA yields and rate constants from the "Zhao" option (Zhao
et al., 2014) of Ma et al. (2017) are used, and the products are apportioned into three SVOC bins
and one low-volatility organic compound (LVOC) bin (Fig. S5). Finally, SVOCs undergo
photooxidation at a rate of $4\times10^{-11}$ cm$^3$ molecules$^{-1}$ s$^{-1}$ (Dzepina et al., 2009; Hodzic et al.,
2010b; Tsimpidi et al., 2010; Hodzic and Jimenez, 2011; Hayes et al., 2015; Ma et al., 2017;
Schroder et al., 2018), producing one product per oxidation step, with yields from Robinson et al.
(2007) for cooking and other SVOCs and yields from Worton et al. (2014) for fossil fuel related
SVOCs, as recommended by Ma et al. (2017). The products from SVOC and IVOC oxidation are
allowed to further oxidize, as highlighted in Fig. S5 and described in prior studies (Hayes et al.,
2015; Ma et al., 2017; Schroder et al., 2018). Generally, each product reacts at a rate of $4\times10^{-11}$
cm$^3$ molecules$^{-1}$ s$^{-1}$ to produce some product at one volatility bin lower, adding one oxygen to the
compound for each oxidation (Dzepina et al., 2009; Tsimpidi et al., 2010; Hodzic and Jimenez,
2011; Hayes et al., 2015; Ma et al., 2017; Schroder et al., 2018). An update includes



fragmentation for a fraction of the molecules that are oxidized, as described in Schroder et al.
(2018) and Koo et al. (2014). As shown in Fig. S5, fragmentation of the compound occurs as it is
oxidized and goes down one volatility bin. For further oxidation of SVOCs from the oxidation of
primary IVOCs, one oxygen is added and 0.25 carbon is removed per step, leading to an increase
in mass of 1.03 (instead of 1.07) per oxidation step (Koo et al., 2014; Schroder et al., 2018). For
further oxidation of products from primary SVOC emissions, one oxygen is added and 0.5
carbon is removed per step, leading to an increase in mass of 0.99 (instead of 1.07) per oxidation
step (Koo et al., 2014; Nault et al., 2018).

**2.5 GEOS-Chem Modeling**
The model used in this study, for ASOA apportionment (Fig. 1), for apportionment of
ASOA to total PM2.5 for premature mortality calculations (Worldwide Premature Deaths Due to
ASOA), and for sensitivity analysis for ASOA production and emissions on premature mortality
calculations, is the GEOS-Chem v12.0.0 global chemical transport model (Bey et al., 2001; The
International GEOS-Chem User Community, 2018) to calculate global concentrations of $PM_{2.5}$
and ASOA at 2°×2.5° horizontal resolution. Goddard Earth Observing System – Forward
Processing (GEOS-FP) assimilated data from the NASA Global Modeling and Assimilation
Office (GMAO) were used for input meteorological fields. The model was run for 2013 to 2018
to take into account interannual variability of meteorological impacts onto $PM_{2.5}$ (therefore,
averaging $PM_{2.5}$ over variations in meteorology). However, the HTAPv2 emission inventory,
which was used for anthropogenic emissions (Janssens-Maenhout et al., 2015), was kept constant
for the 5 years. GEOS-Chem simulates gas and aerosol chemistry with ~700 chemical reactions.



GEOS-Chem calculates the following $PM_{2.5}$ species: sulfate, ammonium, nitrate (Park et al.,
2006); black carbon and POA (Park et al., 2005); SOA (Pye and Seinfeld, 2010; Marais et al.,
2016); sea salt (accumulation mode only (Jaeglé et al., 2011)); and, dust (Duncan Fairlie et al.,

2007).


### 2.5.1 Biogenic SOA

For monoterpene and sesquiterpene SOAs, we used the default complex SOA scheme
(without semi-volatile POA) using the two-product model framework (Pye and Seinfeld, 2010).
This scheme calculates initial oxidation of VOCs with OH, $O_3$, and $NO_3$, and resulting products
are assigned to four different gas-phase semi-volatile species (TSOA0–3) based on volatilities
($c^* = 0.1, 1, 10, 100 \ \mu g \ m^{-3}$). Aerosol and gas species fractions are calculated online using the
partitioning theory, and all are removed by dry and wet deposition processes.
For isoprene SOA, we used the explicit isoprene chemistry developed by Marais et al.
(2016). All the isoprene-derived gas-phase products, including isoprene peroxy radical,
ISOPOOH, IEPOX, glyoxal, and methylglyoxal, are explicitly simulated. Irreversible
heterogeneous uptake of precursors to aqueous aerosols are further calculated using online
aerosol pH and surface area.
GEOS-Chem was used to estimate the relative fractions of the measured SOA in our
studies between anthropogenic and biogenic (isoprene and monoterpene) sources (Fig. 1).
Extensive research has been conducted to evaluate and improve the models performance in
predicting BSOA, as summarized in Table S3. Though these evaluations mainly occurred in the
southeast US, a recent study has also included more global observations to compare with



GEOS-Chem (Pai et al., 2020). Generally, GEOS-Chem appears to overestimate biogenically
derived SOA; however, the model predicted SOA is typically within the uncertainty of the AMS
(Table S3). The overestimation, though, would suggest that the fraction of urban SOA may be
under-predicted by this method, whereas the BSOA may be over-predicted. Therefore, in urban
regions, the amount of SOA from biogenic sources may be lower, especially after the rapid SOA
enhancements (within 12 to 24 equivalent photochemical hours that have been observed around
the world (Nault et al., 2018)). Typically the BSOA is present as a regional background and
subtracted for the analyses used in this work, which focus on strong urban plumes on top of that
background (Hayes et al., 2013, 2015).

**2.5.2 Default GEOS-Chem Sensitivity to ASOA Simulations**
For the sensitivity calculation using the "traditional" ASOA precursors, we used the
two-product model framework (Pye and Seinfeld, 2010). Benzene, toluene, and xylene are
oxidized with OH and converted to peroxy radicals. These peroxy radicals react with $HO_2$ or NO,
resulting in non-volatile ASOA ($HO_2$ pathway, ASOAN species in GEOS-Chem) or
semi-volatile ASOA tracers (NO pathway, ASOA1-3 in GEOS-Chem). As is the case for
monoterpene and sesquiterpene SOA above, GEOS-Chem calculates online partitioning and
dry/wet deposition processes for semi-volatile ASOA tracers. Other conditions including
mortality calculation are kept the same as the base simulation above.

**2.6 Estimation of Premature Mortality Attribution**





Premature deaths were calculated for five disease categories: ischemic heart disease
(IHD), stroke, chronic obstructive pulmonary disease (COPD), acute lower respiratory illness
(ALRI), and lung cancer (LC). We calculated premature mortality for the population aged more
than 30 years, using Eq. 4.
$$Premature\ Death\ =\ Pop \times y_0 \times \frac{RR-1}{RR}$$    Eq. 4
Mortality rate, $y_0$, varies according to the particular disease category and geographic region,
which is available from Global Burden of Disease (GBD) Study 2015 database (IHME, 2016).
Population (Pop) was obtained from Columbia University Center for International Earth Science
Information Network (CIESIN) for 2010 (CIESIN, 2017). Relative risk, RR, can be calculated as
shown in Eq. 5.
$$RR\ =\ 1 + \alpha \times \left( 1 - exp\left( \beta \times \left( PM_{2.5} - PM_{2.5,Threshold} \right)^{\varrho} \right) \right)$$    Eq. 5
$\alpha$, $\beta$, and $\rho$ values depend on disease category and are calculated from Burnett et al. (2014) (see
Table S12 and associated file). If the $PM_{2.5}$ concentrations are below the $PM_{2.5}$ threshold value
(Table S12), premature deaths were computed as zero. However, there could be some health
impacts at concentrations below the $PM_{2.5}$ threshold values (Krewski et al., 2009); following the
methods of the GBD studies, these  can be viewed as lower bounds on estimates of premature
deaths.
We performed an additional sensitivity analysis using the Global Exposure Mortality
Model (GEMM) (Burnett et al., 2018). For the GEMM analysis, we also used age stratified
population data from GWPv3. Premature death is calculated the same as shown in Eq. 4;
however, the relative risk differs. For the GEMM model, the relative risk can be calculated as
shown in Eq. 6.


$$RR = exp(\theta \times \lambda) \text{ with } \lambda = \frac{\log\left(1 + \frac{z}{\alpha}\right)}{\left(1 + \exp\left(\frac{(\hat{\mu} - z)}{\pi}\right)\right)}$$

Eq. 6

Here $z = \max(0, PM_{2.5} - PM_{2.5,Threshold})$; $\theta$, $\pi$, $\hat{\mu}$, $\alpha$, and $PM_{2.5,Threshold}$ depends on disease category and
are from Burnett et al. (2018). Similar to the Eq. 5, if the concentrations are below the threshold
(2.4 µg m$^{-3}$, Burnett et al. (2018)), then premature deaths are computed as zero; however, the
GEMM has a lower threshold than the GBD method.

For GBD, we do not consider age-specific mortality rates or risks. For GEMM, we

calculate age-specific health impacts with age-specific parameters in the exposure response
function (Table S13). We combine the age-specific results of the exposure-response function
with age distributed population data from GPW (CIESIN, 2017) and a national mortality rate
across all ages to assess age-specific mortality.

We calculated total premature deaths using annual average total $PM_{2.5}$ concentrations

derived from satellite-based estimates at the resolution of 0.1°×0.1° from van Donkelaar et al.
(2016) . Application of the remote-sensing based $PM_{2.5}$ at the 0.1°×0.1° resolution rather than
direct use of the GEOS-Chem model concentrations at the 2°×2.5° resolution helps reduce
uncertainties in the quantification of $PM_{2.5}$ exposure inherent in coarser estimates (Punger and
West, 2013).  We also calculated deaths by subtracting from this amount the total annual average
ASOA concentrations derived from GEOS-Chem (Fig. S9). To reduce uncertainties related to
spatial gradients and total concentration magnitudes in our GEOS-Chem simulations of $PM_{2.5}$,
our modeled ASOA was calculated as the fraction of ASOA to total $PM_{2.5}$ in GEOS-Chem,
multiplied by the satellite-based PM2.5 concentrations (Eq. 7).

$ASOA_{sat} = (ASOA_{mod}/PM_{2.5,mod}) \times PM_{2.5,sat}$                    Eq. 7





Finally, this process for estimating $PM_{2.5}$ health impacts considers only $PM_{2.5}$ mass concentration
and does not distinguish toxicity by composition, consistent with the current US EPA position
expressed in Sacks et al. (2019).

**3. Observations of ASOA Production across Three Continents**
**3.1 Observational Constraints of ASOA Production across Three Continents**
Measurements during intensive field campaigns in large urban areas better constrain
concentrations and atmospheric formation of ASOA because the scale of ASOA enhancement is
large compared to SOA from regional background. Generally, ASOA increased with the amount
of urban precursor VOCs and with atmospheric PA (de Gouw et al., 2005; de Gouw and Jimenez,
2009; DeCarlo et al., 2010; Hayes et al., 2013; Nault et al., 2018; Schroder et al., 2018; Shah et
al., 2018). In addition, ASOA correlates strongly with gas-phase secondary photochemical
species, including $O_x$, HCHO, and PAN (Herndon et al., 2008; Wood et al., 2010; Hayes et al.,
2013; Zhang et al., 2015; Nault et al., 2018; Liao et al., 2019) (Table S4; Fig. S2 to Fig. S4),
which are indicators of photochemical processing of emissions.
However, as initially discussed by Nault et al. (2018) and shown in Fig. 4, there is large
variability in these various metrics across the urban areas evaluated here. To the best of the
authors' knowledge, this variability has not been explored and its physical meaning has not been
interpreted. As shown in Fig. 4, though, the trends in ΔSOA/ΔCO are similar to the trends in the
slopes of SOA versus $O_x$, PAN, or HCHO. For example, Seoul is the highest for nearly all
metrics, and is approximately a factor of 6 higher than the urban area, Houston, that generally





showed the lowest photochemical metrics. This suggests that the variability is related to a
physical factor, including emissions and chemistry.
The VOC concentration, together with how quickly the emitted VOCs react ($\Sigma k_i \times [VOC]_i$,
i.e., the hydroxyl radical, or OH, reactivity of VOCs), where k is the OH rate coefficient for each
VOC, are a determining parameter for ASOA formation over urban spatial scales (Eq. 8). ASOA
formation is normalized here to the excess CO mixing ratio ($\Delta$CO) to account for the effects of
meteorology, dilution, and non-urban background levels, and allow for easier comparison
between different studies:
$$\frac{\Delta \text{ASOA}}{\Delta \text{CO}} \propto [OH] \times \Delta t \times \left( \sum_i k_i \times \left[ \frac{\text{VOC}}{\text{CO}} \right]_i \times Y_i \right)$$
Eq. 8

where Y is the aerosol yield for each compound (mass of SOA formed per unit mass of precursor
reacted), and [OH]×Δt is the PA.
BTEX are one group of known ASOA precursors (Gentner et al., 2012; Hayes et al.,
2013), and their emission ratio (to CO) was determined for all campaigns (Table S5). BTEX can
thus provide insight into ASOA production. Fig. 5a shows that the variation in ASOA (at PA =
0.5 equivalent days) is highly correlated with the emission reactivity ratio of BTEX ($R_{BTEX}$,
$\sum_i \left[ \text{VOC}/\text{CO} \right]_i$) across all the studies. However, BTEX alone cannot account for much of the
ASOA formation (see budget closure discussion below), and instead, BTEX may be better
thought of as both partial contributors and also as indicators for the co-emission of other
(unmeasured) organic precursors that are also efficient at forming ASOA.
$O_x$, PAN, and HCHO are produced from the oxidation of a much wider set of VOC
precursors (including small alkenes, which do not appreciably produce SOA when oxidized).





These alkenes have similar reaction rate constants with OH as the most reactive BTEX
compounds (Table S11); however, their emissions and concentration can be higher than BTEX
(Table S7). Thus, alkenes would dominate $R_{Total}$, leading to $O_x$, HCHO, and PAN being produced
more rapidly than ASOA (Fig. 5b–d). When $R_{BTEX}$ becomes more important for $R_{Total}$, the emitted
VOCs are more efficient in producing ASOA. Thus, the ratio of ASOA to gas-phase
photochemical products shows a strong correlation with $R_{BTEX}/R_{Total}$ (Fig. 5b–d).

**3.2 Budget Closure of ASOA for 4 Urban Areas on 3 Continents Indicates Reasonable**

**Understanding of ASOA Sources**

We show that BTEX alone cannot explain the observed ASOA budget for urban areas
around the world. Fig. 6a shows that approximately 25±6% of the observed ASOA originates
from the photooxidation of BTEX. Therefore, other precursors must account for most of the
ASOA produced.
Because alkanes, alkenes, and oxygenated compounds with carbon numbers less than 6
are not significant ASOA precursors, we focus on emissions and sources of BTEX, other
mono-aromatics, IVOCs, and SVOCs. These three classes of VOCs, aromatics, IVOCs, and
SVOCs, have been suggested to be significant ASOA precursors in urban atmospheres
(Robinson et al., 2007; Hayes et al., 2015; Ma et al., 2017; McDonald et al., 2018; Nault et al.,
2018; Schroder et al., 2018; Shah et al., 2018), originating from both fossil fuel and VCP
emissions.
Using the best available emission inventories from cities on three continents
(EMEP/EEA, 2016; McDonald et al., 2018; Li et al., 2019) and observations, we quantify the





emissions of BTEX, other mono-aromatics, IVOCs, and SVOCs for both fossil fuel (e.g.,
gasoline, diesel, kerosene, etc.), VCPs (e.g., coatings, inks, adhesives, personal care products,
and cleaning agents), and cooking sources (Fig. 2 and Fig. 3). This builds off the work of
McDonald et al. (2018) for urban regions on three different continents. Combining these
inventories and observations for the various locations provide the following insights about the
potential ASOA precursors not easily measured or quantified in urban environments (e.g., Zhao
et al., 2014; Lu et al., 2018):  (1) aromatics from fossil fuel accounts for 14-40% (mean 22%) of
the total BTEX and IVOC emissions for the five urban areas investigated in-depth (Fig. 2),
agreeing with prior studies that have shown that the observed ASOA cannot be reconciled by the
observations or emission inventory of aromatics from fossil fuels (e.g., Ensberg et al., 2014;
Hayes et al., 2015). (2) BTEX from both fossil fuels and VCPs account for 25-95% (mean 43%)
of BTEX and IVOC emissions (Fig. 2). China has the lowest contribution of IVOCs, potentially
due to differences in chemical make-up of the solvents used daily (Li et al., 2019), but more
research is needed to investigate the differences in IVOCs:BTEX from Beijing versus US and
UK emission inventories. Nonetheless, this shows the importance of IVOCs for both emissions
and ASOA precursors. (3) IVOCs are generally equal to, if not greater than, the emissions of
BTEX in 4 of the 5 urban areas investigated here (Fig. 2). (4) Overall, VCPs account for a large
fraction of the BTEX and IVOC emissions for all five cities. (5) Finally, SVOCs account for
27-88% (mean 53%) of VOCs generally considered ASOA precursors (VOCs with volatility
saturation concentrations $\leq 10^7 \ \mu g \ m^{-3}$) (Fig. 3). Beijing has the highest contribution of SVOCs
to ASOA precursors due to the use of solid fuels and cooking emissions (Hu et al., 2016). Also,
this indicates the large contribution of a class of VOCs difficult to measure (Robinson et al.,





2007) that are an important ASOA precursor (e.g., Hayes et al., 2015), showing further emphasis
should be placed in quantifying the emissions of this class of compounds.

These results provide an ability to further investigate the mass balance of predicted and

observed ASOA for these urban locations (Fig. 6). The inclusion of IVOCs, other aromatics not
including BTEX, and SVOCs leads to the ability to explain, on average, 85±12% of the observed
ASOA for these urban locations around the world (Fig. 6a). Further, VCP contribution to ASOA
is important for all these urban locations, accounting or, on average, 37±3% of the observed
ASOA (Fig. 6b).

This bottom-up mass budget analysis provides important insights to further explain the

correlation observed in Fig. 5. First, IVOCs are generally co-emitted from similar sources as
BTEX for the urban areas investigated in-depth (Fig. 2). The oxidation of these co-emitted
species leads to the ASOA production observed across the urban areas around the world. Second,
S/IVOCs generally have similar rate constants as toluene and xylenes ($\geq 1\times10^{-11}$ cm$^3$ molec.$^{-1}$ s$^{-1}$)
(Zhao et al., 2014, 2017), the compounds that contribute the most to R$_{BTEX}$, explaining the rapid
ASOA production that has been observed in various studies (de Gouw and Jimenez, 2009;
DeCarlo et al., 2010; Hayes et al., 2013; Hu et al., 2013, 2016; Nault et al., 2018; Schroder et al.,
2018) and correlation (Fig. 5). Finally, the contribution of VCPs and fossil fuel sources to ASOA
is similar across the cities, expanding upon and further supporting the conclusion of McDonald
et al. (2018) in the importance of identifying and understanding VCP emissions in order to
explain ASOA.

**4. Improved Urban SIMPLE Model Using Multi-Cities to Constrain**



## 4.1 Updates to the SIMPLE Model

With the combination of the new dataset, which expands across urban areas on three continents, the SIMPLE parameterization for ASOA (Hodzic and Jimenez, 2011) is updated in the standard GEOS-Chem model to reproduce observed ASOA in Fig. 5a. The parameterization operates as represented by Eq. 9.

$$\text{Emissions} \rightarrow \text{SOAP} \xrightarrow{\ k \times [\text{OH}]\ } \text{ASOA} \qquad \text{Eq. 9}$$

SOAP represents the lumped precursors of ASOA, k is the reaction rate coefficient with OH ($1.25 \times 10^{-11}$ cm$^3$ molecules$^{-1}$ s$^{-1}$), and [OH] is the OH concentration in molecules cm$^{-3}$.

SOAP emissions were calculated based on the relationship between $\Delta\text{SOA}/\Delta\text{CO}$ and $R_{\text{aromatics}}/\Delta\text{CO}$ in Fig. 5a. First, we calculated $R_{\text{aromatics}}/\Delta\text{CO}$ (Eq. 10) for each grid cell and time step as follows:

$$\frac{R_{\text{aromatics}}}{\Delta\text{CO}} = \frac{E_B \times k_B + E_T \times k_T + E_X \times k_X}{E_{\text{CO}}} \qquad \text{Eq. 10}$$

Where E and k stand for the emission rate and reaction rate coefficient with OH, respectively, for benzene (B), toluene (T), and xylenes (X). Ethylbenzene was not included in this calculation because its emission was not available in HTAPv2 emission inventory. However, ethylbenzene contributed a minor fraction of the mixing ratio (~ 7%, Table S5) and reactivity (~6%) of the total BTEX across the campaigns. Reaction rate constants used in this study were $1.22 \times 10^{-12}$, $5.63 \times 10^{-12}$, and $1.72 \times 10^{-11}$ cm$^3$ molec.$^{-1}$ s$^{-1}$ for benzene, toluene, and xylene, respectively (Atkinson and Arey, 2003; Atkinson et al., 2006).





Second, $E_{SOAP}/E_{CO}$ can be obtained from the result of Eq. 11, using slope and intercept in
Fig. 5a, with a correction factor (F) to consider additional SOA production after 0.5 PA
equivalent days, since Fig. 5a shows the comparison at 0.5 PA equivalent days.
$$\frac{E_{SOAP}}{E_{CO}} = \left( Slope \times \frac{R_{Aromatics}}{\Delta CO} + Intercept \right) \times F$$
Eq. 11

Where slope is 24.8 and intercept is –1.7 from Fig. 5a. F (Eq. 12) can be calculated as follows:
$$F = \frac{ASOA_{t=\infty}}{ASOA_{t=0.5d}} = \frac{SOAP_{t=0}}{SOAP_{t=0} \times (1 - exp(-k \times \Delta t \times [OH]))}, \Delta t = 43200\, s$$
Eq. 12

F was calculated as 1.8 by using [OH] = $1.5 \times 10^6$ molecules cm$^{-3}$, which was used in the
definition of 0.5 PA equivalent days for Fig. 5a.
Finally, $E_{SOAP}$ can be computed by multiplying CO emissions ($E_{CO}$) for every grid point
and time step in GEOS-Chem by the $E_{SOAP}/E_{CO}$ ratio.

**4.2 Results of Updated SIMPLE Model**
The SIMPLE model was originally designed and tested against the observations collected
around Mexico City (Hodzic and Jimenez, 2011). It was then tested against observations
collected in Los Angeles (Hayes et al., 2015; Ma et al., 2017). As both data sets have nearly
identical ΔSOA/ΔCO and $R_{BTEX}$ (Fig. 4 and Fig. 5), it is not surprising that the SIMPLE model
did well in predicting the observed ΔSOA/ΔCO for these two urban regions with consistent
parameters. Though the SIMPLE model generally performed better than more explicit models, it
generally had lower skill in predicting the observed ASOA in urban regions outside of Mexico
City and Los Angeles (Shah et al., 2019; Pai et al., 2020).





This may stem from the original SIMPLE model with constant parameters missing the

ability to change the amount and reactivity of the emissions, which are different for the various
urban regions, versus the ASOA precursors being emitted proportionally to only CO (Hodzic and
Jimenez, 2011; Hayes et al., 2015). For example, in the HTAP emissions inventory, the CO
emissions for Seoul, Los Angeles, and Mexico City are all similar (Fig. S6); thus, the original
SIMPLE model would suggest similar $\Delta SOA/\Delta CO$ for all three urban locations. However, as
shown in Fig. 4 and Fig. 5, the $\Delta SOA/\Delta CO$ is different by nearly a factor of 2. The inclusion of
the emissions and reactivity, where $R_{BTEX}$ for Seoul is approximately a factor of 2.5 higher than
Los Angeles and Seoul, into the improved SIMPLE model better accounts for the variability in
SOA production, as shown in Fig. 5. Thus, the inclusion and use of this improved SIMPLE
model refines the simplified representation of ASOA in chemical transport models and/or box
models.

**534 5. Preliminary Evaluation of Worldwide Premature Deaths Due to ASOA with Updated**

**535 SIMPLE Parameterization**

The improved SIMPLE parameterization is used along with GEOS-Chem to provide an

accurate estimation of ASOA formation in urban areas worldwide and provide an ability to
obtain realistic simulations of ASOA based on measurement data. We use this model to quantify
the attribution of $PM_{2.5}$ ASOA to premature deaths. Analysis up to this point has been for $PM_1$;
however, both the chemical transport model and epidemiological studies utilize $PM_{2.5}$. For
ASOA, this will not impact the discussion and results here because the mass of OA (typically
80–90%) is dominated by $PM_1$ (e.g., Bae et al., 2006; Seinfeld and Pandis, 2006), and ASOA is



formed mostly through condensation of oxidized species, which favors partitioning onto smaller
particles (Seinfeld and Pandis, 2006).
The procedure for this analysis is described in Fig. 7 and Sect. 2.5 and 2.6. Briefly, we
combine high-resolution satellite-based $PM_{2.5}$ estimates (for exposure) and a chemical transport
model (GEOS-Chem, for fractional composition) to estimate ASOA concentrations and various
sensitivity analysis (van Donkelaar et al., 2015). We calculated ~3.3 million premature deaths
(using the Integrated Exposure-Response, IER, function) are due to long-term exposure of
ambient $PM_{2.5}$ (Fig. S7, Table S14), consistent with recent literature (Cohen et al., 2017).
The attribution of ASOA $PM_{2.5}$ premature deaths can be calculated one of two ways: (a)
marginal method (Silva et al., 2016) or (b) attributable fraction method (Anenberg et al., 2019).
For method (a), it is assumed that a fraction of the ASOA is removed, keeping the rest of the
$PM_{2.5}$ components approximately constant, and the change in deaths is calculated from the deaths
associated with the total concentration less the deaths calculated using the reduced total $PM_{2.5}$
concentrations. For method (b), the health impact is attributed to each $PM_{2.5}$ component by
multiplying the total deaths by the fractional contribution of each component to total $PM_{2.5}$. For
method (a), the deaths attributed to ASOA are ~340,000 people per year (Fig. 8); whereas, for
method (b), the deaths are ~370,000 people per year. Both of these are based on the IER response
function (Cohen et al., 2017).
Additional recent work (Burnett et al., 2018) has suggested less reduction in the
premature deaths versus $PM_{2.5}$ concentration relationship at higher $PM_{2.5}$ concentrations, and
lower concentration limits for the threshold below which this relationship is negligible, both of
which lead to much higher estimates of $PM_{2.5}$ associated premature deaths. This is generally





termed the Global Exposure Mortality Model (GEMM). Using the two attribution methods
described above (a and b), the ASOA $PM_{2.5}$ premature deaths are estimated to be ~640,000
(method a) and ~900,000 (method b) (Fig. S7 and Fig. S10 and Table S15).
Compared to prior studies using chemical transport models to estimate premature deaths
associated with ASOA (e.g., Silva et al., 2016; Ridley et al., 2018), which assumed non-volatile
POA and "traditional" ASOA precursors, the attribution of premature mortality due to ASOA is
over an order of magnitude higher in this study (Fig. 9). This occurs using either the IER and
GEMM approach for estimating premature mortality (Fig. 9). For regions with larger populations
and more $PM_{2.5}$ pollution, the attribution is between a factor of 40 to 80 higher. This stems from
the non-volatile POA and "traditional" ASOA precursors over-estimating POA and
under-estimating ASOA compared to observations (Schroder et al., 2018). These offsetting
errors will lead to model predicted total OA similar to observations (Ridley et al., 2018; Schroder
et al., 2018), yet different conclusions on whether POA versus SOA is more important for
reducing $PM_{2.5}$ associated premature mortality. Using a model constrained to atmospheric
observations (Fig. 5 and Fig. 6, see Sect. 4) leads to a more accurate estimation of the
contribution of ASOA to $PM_{2.5}$ associated premature mortality that has not been possible in prior
studies. We note that ozone concentrations change little as we change the ASOA simulation (see
Sect. S4 in the SI and Fig. S12).
A limitation in this study is the lack of sufficient measurements in South and Southeast
Asia, Eastern Europe, Africa, and South America (Fig. 1), though these areas account for 44% of
the predicted reduction in premature mortality for the world (Table S14). However, as
highlighted in Table S16, these regions likely still consume both transportation fuels and VCPs,





although in lower per capita amounts than more industrialized countries. This consumption is
expected to lead to the same types of emissions as for the cities studied here, though more field
measurements are needed to validate global inventories of VOCs and resulting oxidation
products in the developing world. Transportation emissions of VOCs are expected to be more
dominant in the developing world due to higher VOC emission factors associated with inefficient
combustion engines, such as two-stroke scooters (Platt et al., 2014) and auto-rickshaws (e.g.,
Goel and Guttikunda, 2015). Also, unlike many of the cities studied here, solid fuels are used for
residential heating and cooking, which impact the outdoor air quality as well (Hu et al., 2013,
2016; Lacey et al., 2017; Stewart et al., 2020), and which also lead to SOA (Heringa et al.,
2011). Recently, emission factors from Abidjan, Côte d'Ivoire, a developing urban area, showed
the dominance of emissions from transportation and solid fuel burning, with BTEX being an
important fraction of the total emissions, and that all the emissions were efficient in producing
ASOA (Dominutti et al., 2019). Further, investigation of emissions in New Delhi region of India
demonstrated the importance of both transportation and solid fuel emissions (Stewart et al.,
2020; Wang et al., 2020) while model comparisons with observations show an underestimation
of OA compared to observations due to a combination of emissions and OA representation (Jena
et al., 2020). Despite emission source differences, SOA is still an important component of $PM_{2.5}$
(e.g., Singh et al., 2019) and thus will impact air quality and premature mortality in developing
regions. Admittedly, though, our estimates will be less accurate for these regions.

**6. Conclusions**





In summary, ASOA is an important, though inadequately constrained component of air
pollution in megacities and urban areas around the world. This stems from the complexity
associated with the numerous precursor emission sources, chemical reactions, and oxidation
products that lead to observed ASOA concentrations. We have shown here that the variability in
observed ASOA across urban areas is correlated with $R_{BTEX}$, a marker for the co-emissions of
IVOC from both transportation and VCP emissions. Global simulations indicate ASOA
contributes to a substantial fraction of the premature mortality associated with $PM_{2.5}$. Reductions
of the ASOA precursors will reduce the premature deaths associated with $PM_{2.5}$, indicating the
importance of identifying and reducing exposure to sources of ASOA. These sources include
emissions that are both traditional (transportation) as well as non-traditional emissions of
emerging importance (VCPs) to ambient $PM_{2.5}$ concentrations in cities around the world. Further
investigation of speciated IVOCs and SVOCs for urban areas around the world along with SOA
mass concentration and other photochemical products (e.g., $O_x$, PAN, and HCHO) for other
urban areas, especially in South Asia, throughout Africa, and throughout South America, would
provide further constraints to improve the SIMPLE model and our understanding of the emission
sources and chemistry that leads to the observed SOA and its impact on premature mortality.





## Acknowledgements

This study was partially supported by grants from NASA NNX15AT96G, NNX16AQ26G, Sloan Foundation 2016-7173, NSF AGS-1822664, EPA STAR 83587701-0, NERC NE/H003510/1, NERC NE/H003177/1, NERC NE/H003223/1, NOAA NA17OAR4320101, NCAS R8/H12/83/037, Natural Science and Engineering Research Council of Canada (NSERC, RGPIN/05002-2014), and the Fonds de Recherche du Québec —Nature et technologies (FRQNT, 2016-PR-192364). This manuscript has not been formally reviewed by EPA. The views expressed in this document are solely those of the authors and do not necessarily reflect those of the Agency. EPA does not endorse any products or commercial services mentioned in this publication. We thank Katherine Travis for useful discussions. We acknowledge B J. Bandy, J. Lee, G. P. Mills, d. D. Montzka, J. Stutz, A. J. Weinheimer E. J. Williams, E. C. Wood, and D. R. Worsnop for use of their data.

## Data Availability

TexAQS measurements are available at https://esrl.noaa.gov/csl/groups/csl7/measurements/2000TexAQS/LaPorte/DataDownload/ and upon request. NEAQS measurements are available at https://www.esrl.noaa.gov/csl/groups/csl7/measurements/2002NEAQS/. MILAGRO measurements are available at http://doi.org/10.5067/Aircraft/INTEXB/Aerosol-TraceGas. CalNex measurements are available at https://esrl.noaa.gov/csl/groups/csl7/measurements/2010calnex/Ground/DataDownload/. ClearfLo measurements are available at https://catalogue.ceda.ac.uk/uuid/6a5f9eedd68f43348692b3bace3eba45. SEAC$^4$RS measurements are available at http://doi.org/10.5067/Aircraft/SEAC4RS/Aerosol-TraceGas-Cloud. WINTER measurements are available at https://data.eol.ucar.edu/master_lists/generated/winter/. KORUS-AQ measurements are available at http://doi.org/10.5067/Suborbital/KORUSAQ/DATA01. Data from Chinese campaigns are available upon request, and rest of data used were located in papers cited. GEOS-Chem data available upon request. Figures will become accessible at cires1.colorado.edu/jimenez/group_pubs.html.

## Competing Interests

The authors declare no competing interests.

## Author Contribution

B.A.N., D.S.J., B.C.M., J.A.dG., and J.L.J designed the experiment and wrote the paper. B.A.N., PC.-J., D.A.D., W.H., J.C.S, J.A., D.R.B., M.R.C., H.C., M.M.C., P.F.D, G.S.D., R.D., F.F, A.F., J.B.G., G.G., J.F.H, T.F.H., P.L.H., J.H., M.H., L.G.H., B.T.J., W.C.K., J.L., I.B.P., J.P., B.R.,



C.E.R., D.R., J.M.R, T.B.R, M.S., J.W., C.W., P.W., G.M.W., D.E.Y., B.Y., J.A.dG., and J.L.J.
collected and analyzed the data. D.S.J. and A.H. ran the GEOS-Chem model and B.A.N., D.S.J,
and J.L.J. analyzed the model output. B.A.N., P.L.H., J.M.S., and J.L.J. ran and analyzed the 0-D
model used for ASOA budget analysis of ambient observations. B.C.M., A.L., M.L., and Q.Z.
analyzed and provided the emission inventories used for the 0-D box model. D.S.J., D.K.H., and
M.O.N. conducted the ASOA attribution to mortality calculation, and B.A.N., D.S.J., D.K.H.,
M.O.N., J.A.dG, and J.L.J analyzed the results. All authors reviewed the paper.



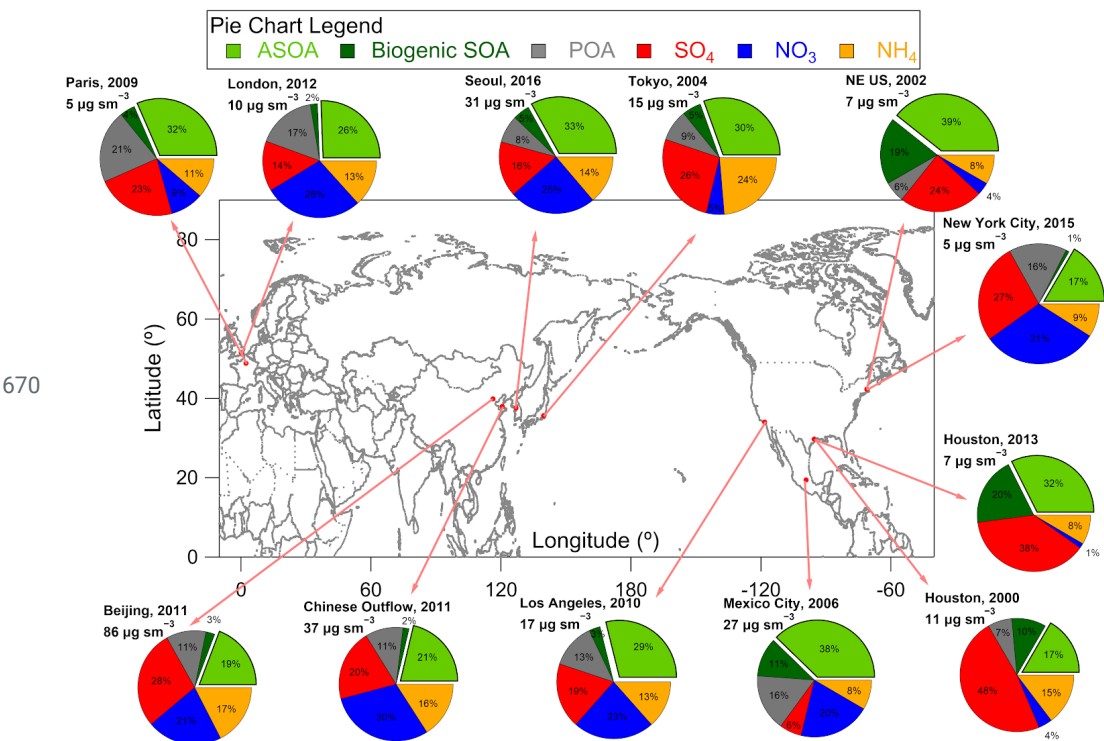

**Figure 1**. Non-refractory submicron aerosol composition measured in urban and urban outflow regions from field campaigns used in this study, all in units of µg m$^{-3}$, at standard temperature (273 K) and pressure (1013 hPa) (sm$^{-3}$). See Sect. 2 (GEOS-Chem Section and Table 1) for further information on measurements, studies, and apportionment of SOA into ASOA and BSOA.

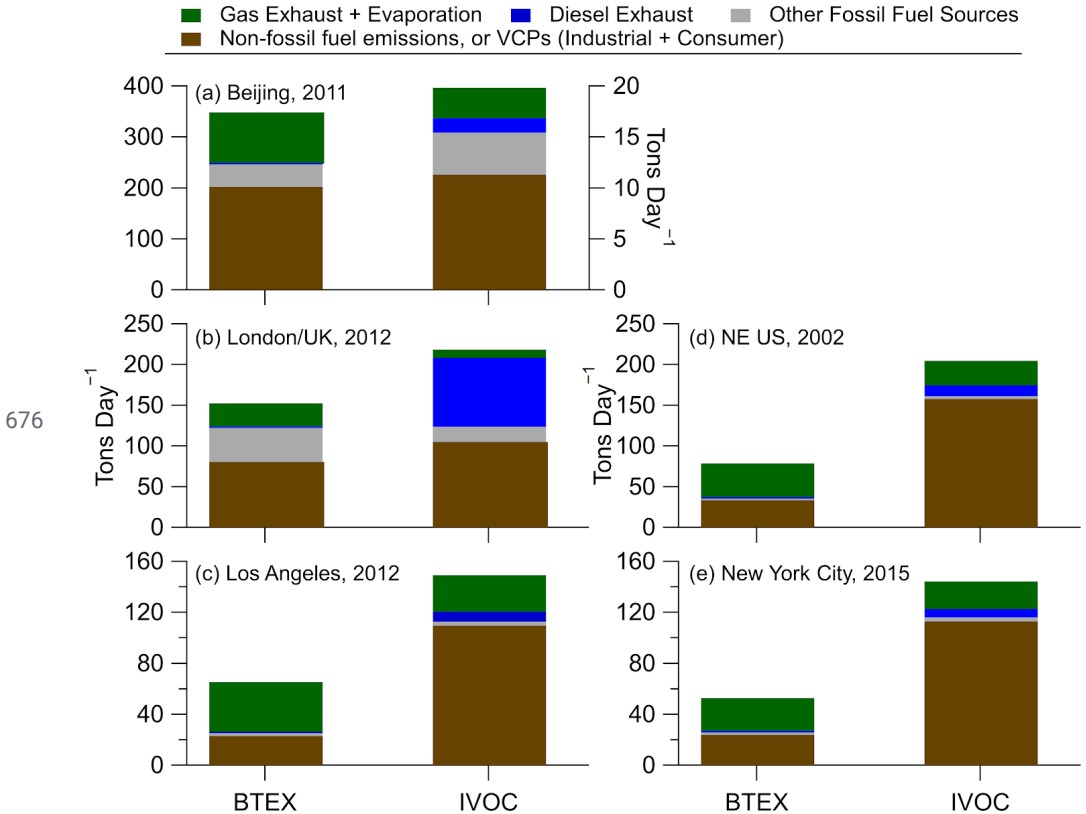

**Figure 2.** Comparison of BTEX and IVOC sources for (a) Beijing (see SI section about Beijing emission inventory), (b) London (see SI section about London/UK emission inventory), and (c) Los Angeles, (d) Northeast United States, and (e) New York City (see SI section about United States for (c) – (e)). For (a), BTEX is on the left axis and IVOC is on the right axis, due to the small emissions per day for IVOC.

682

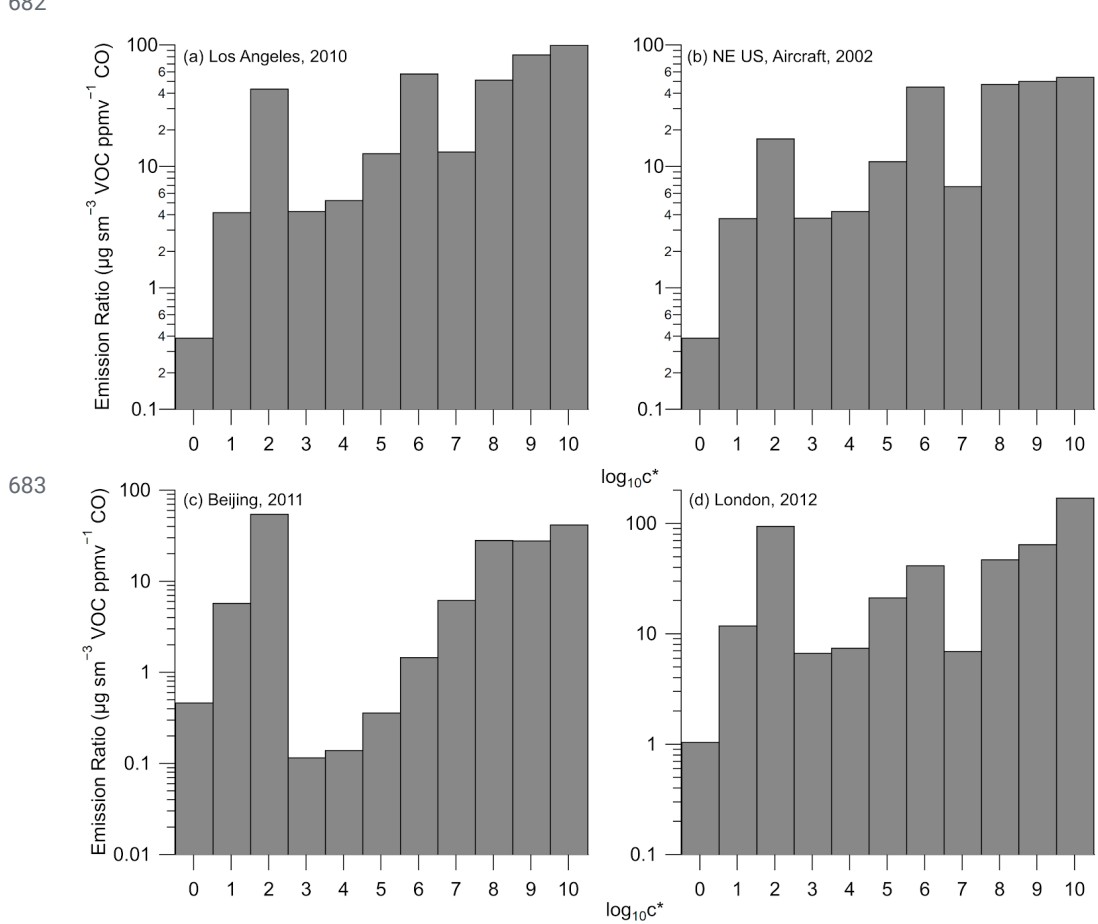

683

**Figure 3**. Emission ratio versus saturation concentration ($\log_{10}(c^*)$) for (a) Los Angeles, (b) NE US, aircraft, (c) Beijing, and (d) London. The emission ratios for VOCs ($\log_{10}(c^*) \geq 7$) were taken from de Gouw et al. (2017) and Ma et al. (2017) for Los Angeles, Warneke et al. (2007) for NE US, aircraft, and Wang et al. (2014) for Beijing while the VOC emission ratio for London is from Table S6 to Table S8. For VOCs between $\log_{10}(c^*)$ of 3 and 6 (IVOCs), the volatility distribution from McDonald et al. (2018), along with the ratio of IVOC to BTEX from Figure SI-6 and the emission ratio of BTEX (Table S6), were used to determine the emission ratio versus saturation concentration. Finally, for VOCs between $\log_{10}(c^*)$ 0 and 2 (SVOCs), the volatility distributions from Robinson et al. (2007) for non-fossil fuel POA and from Worton et al. (2014) for fossil fuel POA were used to convert the normalized POA mass concentration (Table S9) to VOC emission ratios. Note, the emission ratio versus saturation concentration for New York City, 2015, was similar to (b), as the emissions were similar (Fig. 2) and the BTEX for New York City is the same as NE US (Table S5).






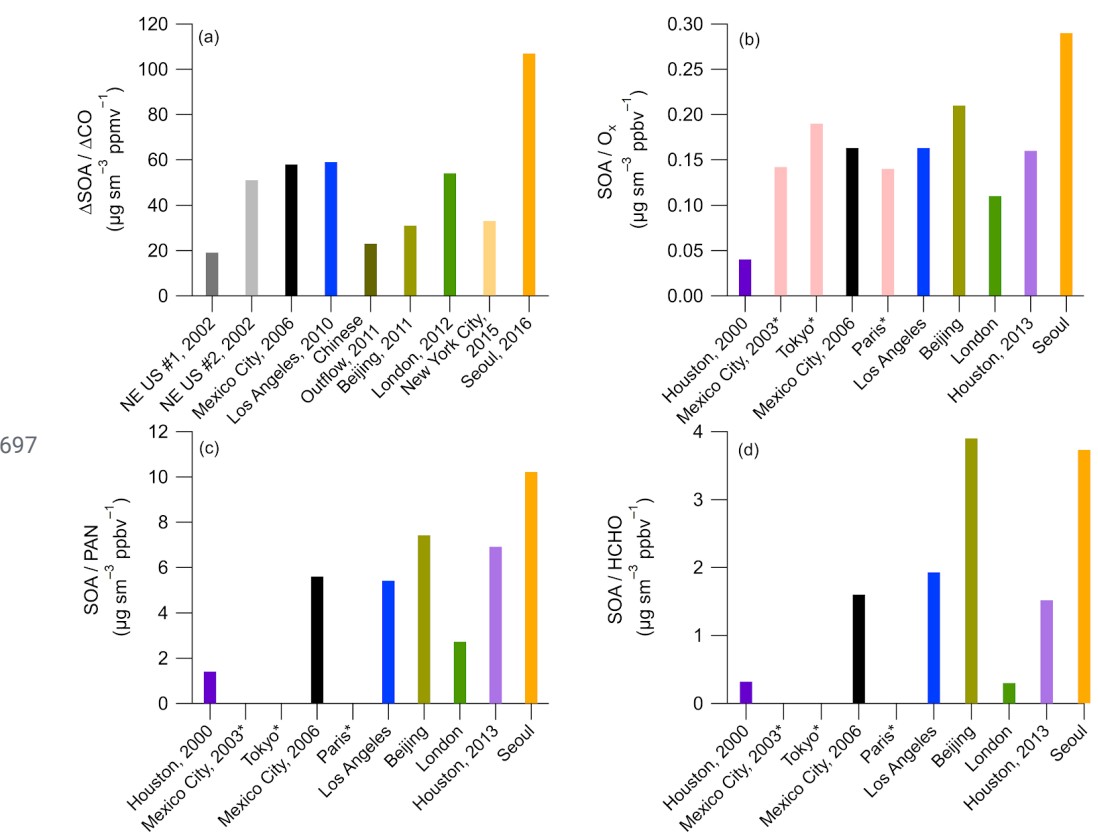

**Figure 4**. (a) A comparison of the ΔSOA/ΔCO for the urban campaigns on three continents.
Comparison of (b) SOA/Ox, (c) SOA/HCHO, and (d) SOA/PAN slopes for the urban areas
(Table S4). For (b) through (d), cities marked with * have no HCHO, PAN, or hydrocarbon data.

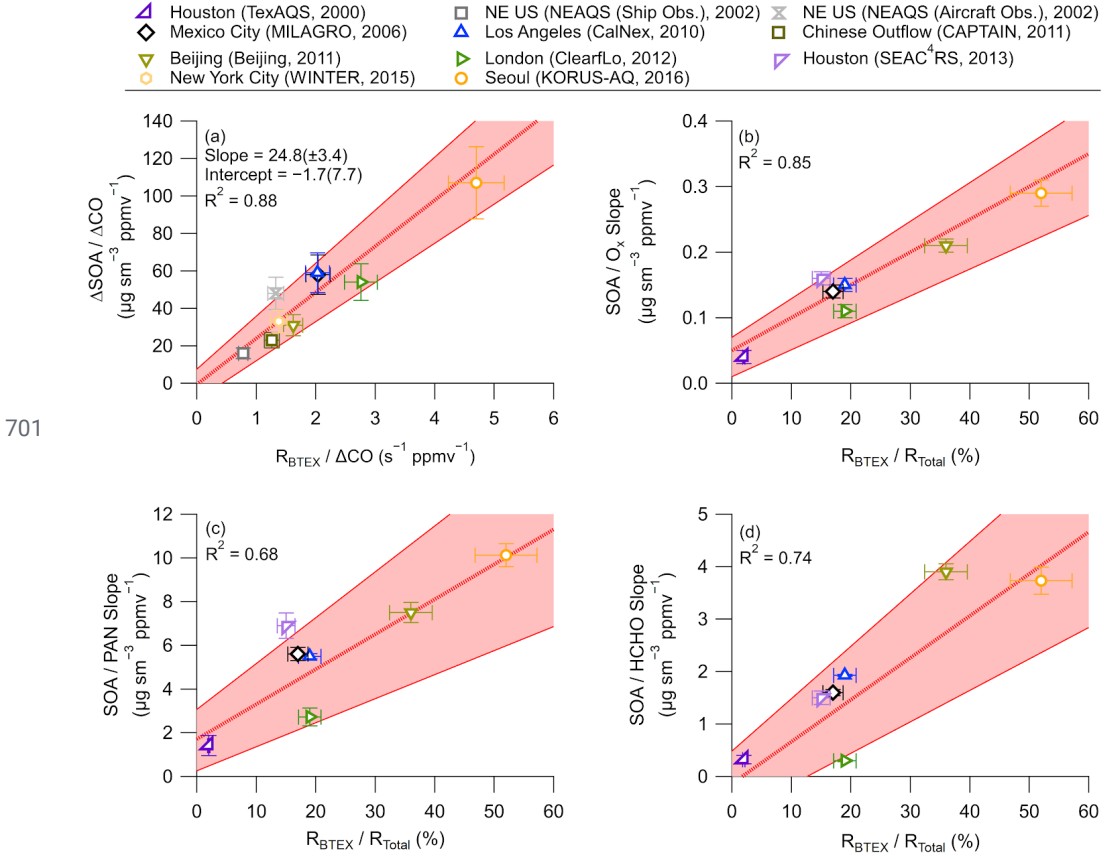

**Figure 5.** (a) Scatter plot of background and dilution corrected ASOA concentrations ($\Delta$ASOA/$\Delta$CO at PA = 0.5 equivalent days) versus BTEX emission reactivity ratio ($R_{BTEX}= \sum_i \left[ {}^{VOC}/_{CO} \right]_i$) for multiple major field campaigns on three continents. Comparison of ASOA versus (b) Ox, (c) PAN, and (d) HCHO slopes versus the ratio of the BTEX/Total emission reactivity, where total is the OH reactivity for the emissions of BTEX + C¬2-3 alkenes + C2-6 alkanes (Table S5 through Table S7), for the campaigns studied here. For all figures, red shading is the ±1σ uncertainty of the slope, and the bars are ±1σ uncertainty of the data (see Sect. 2.2).



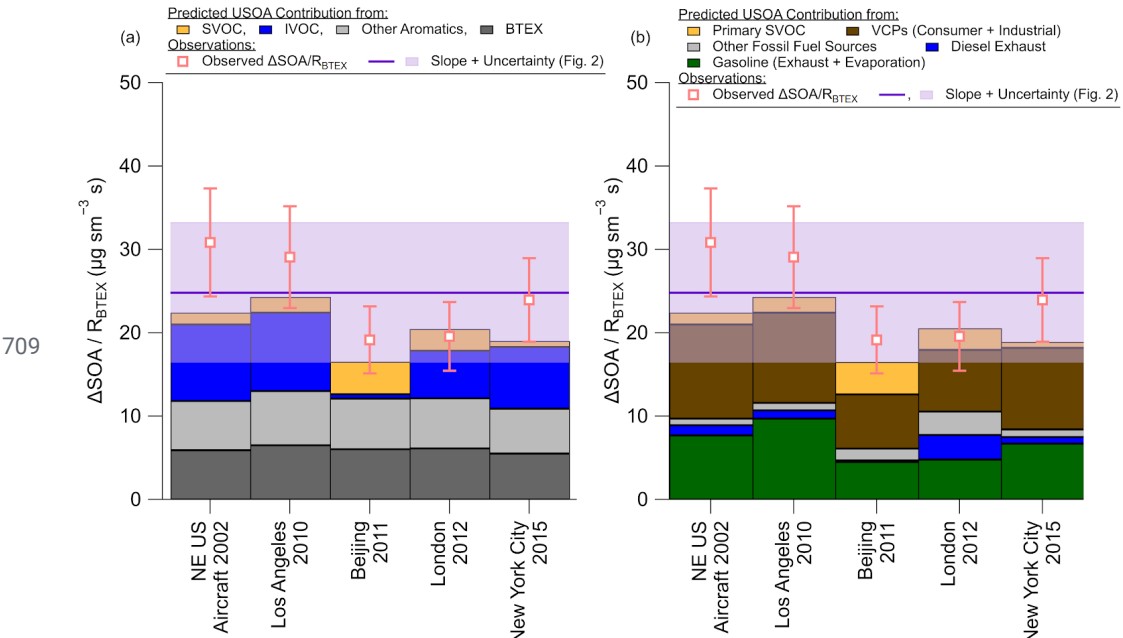

**Figure 6**. (a) Budget analysis for the contribution of the observed $\Delta SOA/R_{BTEX}$ (Fig. 5) for cities with known emissions inventories for different volatility classes (see SI and Fig. 2 and Fig. 3). (b) Same as (a), but for sources of emissions. For (a) and (b), SVOC is the contribution from both vehicle and other (cooking, etc.) sources. See Sect. 2 and SI for information about the emissions, ASOA precursor contribution, error analysis, and discussion about sensitivity of emission inventory IVOC/BTEX ratios for different cities and years in the US.





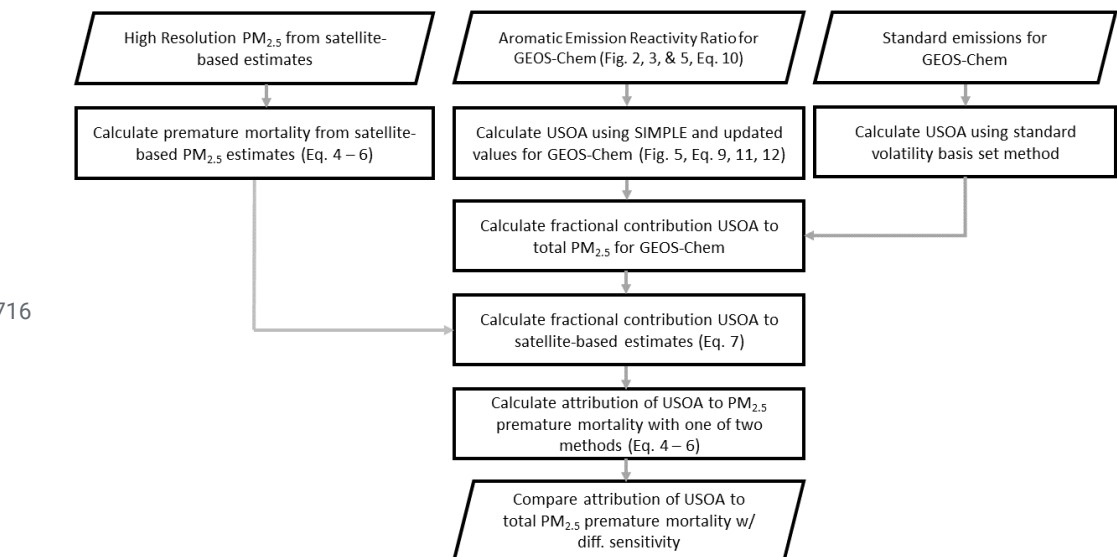

716

**Figure 7**. Flowchart describing how observed ASOA production was used to calculate ASOA in GEOS-Chem, and how the satellite-based $PM_{2.5}$ estimates and GEOS-Chem $PM_{2.5}$ speciation was used to estimate the premature mortality and attribution of premature mortality by ASOA. See Sect. 2 for further information about the details in the figure. SIMPLE is described in Eq. 9 and by Hodzic and Jimenez (2011) and Hayes et al. (2015). The one of two methods mentioned include either the Integrated Exposure-Response (IER) (Burnett et al., 2014) with Global Burden of Disease (GBD) dataset (IHME, 2016) or the new Global Exposure Mortality Model (GEMM) (Burnett et al., 2018) methods. For both IER and GEMM, the marginal method (Silva et al., 2016) or attributable fraction method (Anenberg et al., 2019) are used.



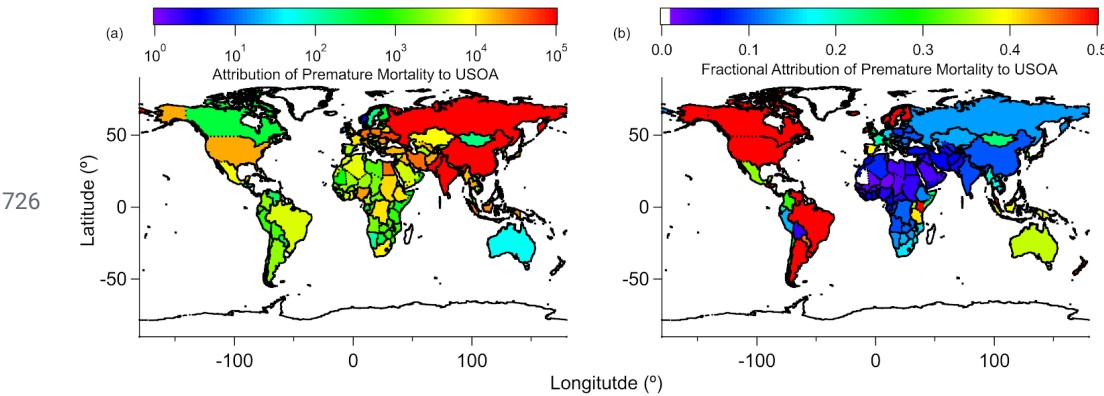

**Figure 8**. Five-year average (a) estimated reduction in PM$_{2.5}$-associated premature deaths, by country, upon removing ASOA from total PM$_{2.5}$, and (b) fractional reduction (reduction PM$_{2.5}$ premature deaths / total PM$_{2.5}$ premature deaths) in PM$_{2.5}$-associated premature deaths, by country, upon removing ASOA from GEOS-Chem. The IER methods are used here. See Fig. S7 and Fig. S10 for results using GEMM. See Fig. S8 for 10×10 km$^2$ area results in comparison with country-level results.

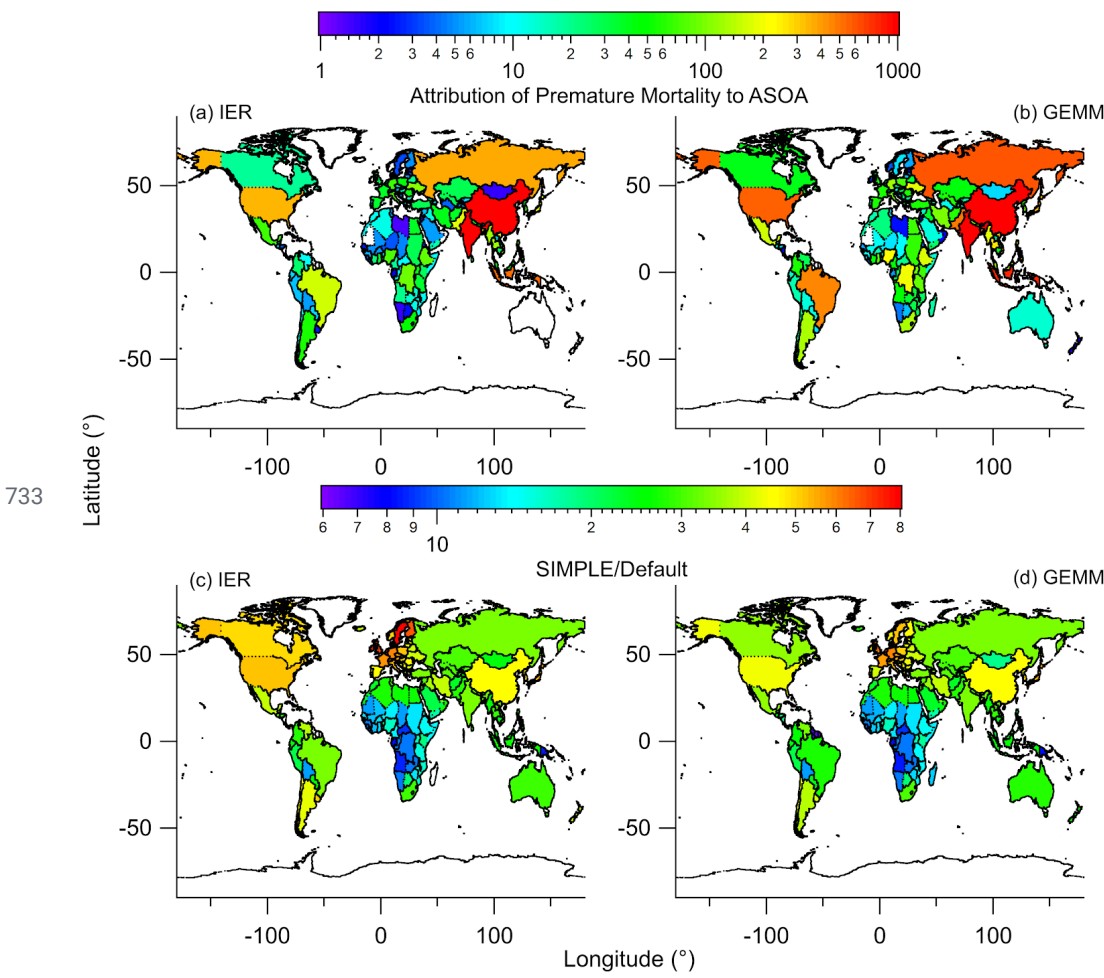

**Figure 9**. Attribution of premature mortality to ASOA using (a) IER or (b) GEMM, using the non-volatile primary OA and traditional SOA precursors method in prior studies (e.g., Ridley et al., 2018). The increase in attribution of premature mortality to ASOA for the "SIMPLE" model (Fig. 8) versus the non-volatile primary OA and traditional SOA precursor method ("Default"), for (c) IER and (d) GEMM.





Table 1. List of campaigns used here. For values previously reported for those campaigns, they
are noted. For Seasons, W = Winter, Sp = Spring, and Su = Summer.

| Location | Field Campaign | Coordinates | | Time Period | Season | Previous Publication/Campaign Overview |
|---|---|---|---|---|---|---|
| | | Long. (°) | Lat. (°) | | | |
| Houston, TX, USA (2000) | TexAQS 2000 | −95.4 | 29.8 | 15/Aug/2000 - 15/Sept/2000 | Su | Jimenez et al. (2009)[a], Wood et al. (2010)[b] |
| Northeast USA (2002) | NEAQS 2002 | −78.1 - −70.5 | 32.8 - 43.1 | 26/July/2002; 29/July/2002 - 10/Aug/2002 | Su | Jimenez et al. (2009)[a], de Gouw and Jimenez (2009)[c], Kleinman et al. (2007)[c] |
| Mexico City, Mexico (2003) | MCMA-2003 | −99.2 | 19.5 | 31/Mar/2003 - 04/May/2003 | Sp | Molina et al. (2007), Herndon et al. (2008)[b] |
| Tokyo, Japan (2004) | | 139.7 | 35.7 | 24/July/2004 - 14/Aug/2004 | Su | Kondo et al. (2008)[a], Miyakawa et al. (2008)[a], Morino et al. (2014)[b] |
| Mexico City, Mexico (2006) | MILAGRO | −99.4 - −98.6 | 19.0 − 19.8 | 04/Mar/2006 - 29/Mar/2006 | Sp | Molina et al. (2010), DeCarlo et al. (2008)[a], Wood et al. (2010)[b], DeCarlo et al. (2010)[c] |
| Paris, France (2009) | MEGAPOLI | 48.9 | 2.4 | 13/July/2009 - 29/July/2009 | Su | Freney et al. (2014)[a], Zhang et al. (2015)[b] |
| Pasadena, CA, USA (2010) | CalNex | −118.1 | 34.1 | 15/May/2010 - 16/June/2010 | Sp | Ryerson et al. (2013), Hayes et al. (2013)[a,b,c] |
| Changdao Island, China (2011) | CAPTAIN | 120.7 | 38.0 | 21/Mar/2011 - 24/Apr/2011 | Sp | Hu et al. (2013)[a,c] |
| Beijing, China (2011) | CareBeijing 2011 | 116.4 | 39.9 | 03/Aug/2011 - 15/Sept/2011 | Su | Hu et al. (2016)[a,b,c] |
| London, UK (2012) | ClearfLo | 0.1 | 51.5 | 22/July/2012 - 18/Aug/2012 | Su | Bohnenstengel et al. (2015) |
| Houston, TX, USA (2013) | SEAC⁴RS | −96.0 - −94.0 | 29.2 - 30.3 | 01/Aug/2013 - 23/Sept/2013 | Su | Toon et al. (2016) |
| New York City, NY, USA (2015) | WINTER | −74.0 - −69.0 | 39.5 - 42.5 | 07/Feb/2015 | W | Schroder et al. (2018)[a,c] |
| Seoul, South Korea (2016) | KORUS-AQ | 124.6 - 128.0 | 36.8 - 37.6 | 01/May/2016 - 10/June/2016 | Sp | Nault et al. (2018)[a,b,c,d] |

[a]Reference used for $PM_1$ composition. [b]Reference used for $SOA/O_x$ slope. [c]Reference used for
$\Delta OA/\Delta CO$ value. [d]Reference used for SOA/HCHO and SOA/PAN slopes





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
