# Peer review of "Anthropogenic Secondary Organic Aerosols Contribute Substantially to Air Pollution 2 Mortality"

_Atmospheric Chemistry and Physics, 2020_

## Referee Comment (RC1) · Anonymous Referee #1 · 8 Jan 2021

This manuscript leverages an impressive breadth and diversity of data to shed light on a critical public health and environmental policy question: how many premature deaths can be avoided annually with reductions in emissions of organic compounds that humans have direct control over? The methods and approaches that are developed are generally sound, although far from perfect, for this kind of high-level endeavor. My main issues with the paper in its current form are the confusing organization and presentation of ideas, some slight misplaced focus on particular organic aerosol model updates, and the decision to ignore solid fuel burning in the model formulation. These and other issues below should be addressed before publication.

[Figure]

General Comments:

1. I think the main ideas in the paper are quite compelling: a) reconstruct measured SOA from in situ campaigns using correlations with likely predictors, b) incrementally improve a streamlined parameterization for SOA prediction (SIMPLE), c) integrate the SIMPLE predictions into a full-science CTM prediction of PM2.5 and use satellite data to further refine the predictions, d) feed those predictions to a premature death parameterization to quantify human health impact, and e) investigate key sensitivities. I would reorganize the entire paper so that the methods, results and discussion each flow in that order. Currently the introduction gives little clue about how the pieces will fit together or the goals of the paper, beyond showing that ASOA is important. Much of (b) above is discussed inappropriately in the current results section 4. Manuscript sections 3, 4 and 5 contain quite a lot of methods discussion that should be moved out to section 2. For example, equations rarely belong in a results section. I could even suggest that most of sections 2.2 and 2.4 be moved to SI. The details of the chemical mechanism used in 2.4 are a bit irrelevant once the SOA/R_BTEX enhancement ratio is confirmed for use in SIMPLE. If the mechanism were more sophisticated (e.g. HOM formation, carbon-conserving fragmentation to lower MW products, oligomerization, etc) then I think there would be more cause for focusing on it, but the schemes used here are relatively close to the SIMPLE approach in terms of one-way generation of SOA.

2. The authors repeatedly compare their updated approach including semivolatile POA to previous efforts to assess ASOA impacts on human health using nonvolatile POA assumptions. The implication here is that treating POA as semivolatile might be as important to getting ASOA correct as the dramatic increase in ASOA precursors. But one look at Fig. 6 shows that in most cases it's the way BTEX, IVOCs and Other Aromatics are treated (emissions, SOA yields, aging) that is really driving the ASOA formation. So while I think the update to semivolatile POA is a good one and it gives room for greater ASOA production, I think the authors focus on it too much in this

study (see specific comment #3 below and lines 572-581 for examples that should be addressed). Instead there should be much more focus in the budget section on how the parameters they have chosen for their VCP emissions and SOA chemistry scheme are driving larger IVOC and VOC contributions and how confident they are in those parameters. For the SIMPLE and GEOS-Chem sections, the key sensitivity is the SOA/R_BTEX ratio (see specific comment #7) and there should be more discussion on its impact. Also, the SIMPLE rate constant k is parameterized from the CalNex data, correct? Why was that not revisited and optimized for performance among all the measurement campaigns?

3. The authors spend some time in the discussion addressing the fact that solid fuel combustion emissions are missing from this study. I'm still very concerned that much of the global results they show are corrupted by this omission, not just in southern Asia and Africa. Large regions of Northern/Western Europe and North America will also be affected by residential wood fuel burning, especially in the winter. The authors should at least justify their omission of solid fuels for the measurement campaigns citing tracer analyses, for example. To address the global comparisons, can you add a reference to SOA/CO ratios for wood combustion and comment on their similarity or difference from what has gone into SIMPLE for this study?

4. Why does the SIMPLE model now rely on BTEX alone? It seems to be doing better overall than when it just relied on CO, but why not use more than one variable with BTEX to develop a multilinear fit for the SOAP emissions? For example, Seoul is likely a problematic point in the SOA vs. BTEX regression (see specific comment #7). It's driving down the ratio and thus probably leading to under-representing impacts in the Northeast US and LA. Taken together, Figs. 3 and 4 suggest that in Seoul there are SOA sources associated with CO emissions that are not as highly associated with BTEX. Using too many independent variables would surely end up overfitting, but why not add 1 or 2 key variables (like CO and POA) since you have a good idea that the relative contributions of sources (e.g. vehicles, VCPs, and solid fuel use) vary from city

to city?

5. Lines 455-457: This observation about SVOCs is difficult to believe based on existing NMOG and POA profiles in the literature. I have yet to see a volatility profile for any source where the SVOC accounted for half of the total ASOA precursor, let alone 88%. Is there something unexpected going on with the CO normalization of POA vs. VOCs here? I confess this one catches me completely by surprise, and likewise the large influence of SVOCs in Fig. 3 looks strange as well. The authors have made an emphatic case for the dominant and growing role of VCPs. Wouldn't these be overwhelmingly VOCs and IVOCs? Cooking emissions are used to explain this to some degree, but if the Robinson et al. (2007) profile is used for the cooking emissions, I would expect a lot more 10ˆ3 and 10ˆ4 C* compounds. Regardless, SVOC should probably be included on Fig. 2 as a separate series like BTEX and IVOC. And I recommend adding more description about how SVOC emission ratios were derived.

Specific Comments:

1. Line 95-98: These generalities about IVOCs and SVOCs are perhaps useful as an introduction for those who may be unfamiliar but based on more current understanding of emissions sampling and speciation, they may be more confusing than helpful. Lu et al. (2018) show in their Fig. 1 that most of the IVOC would have missed the filters for the vehicles they studied, but much of the SVOC is expected to be captured by the filters. Even more SVOC would presumably be captured for stationary sources at conditions relevant for "condensable particulate matter" measurements (i.e. low dilution, cooled temperatures); see Morino et al. (2018). As for IVOCs, VBS profiles for biomass burning sources like those in May et al. (2013) show that IVOCs are probably included in many if not most PM emission factors measured for these sources if the emissions are not diluted enough. The authors here are not focused on wildfires, but certainly cooking/residential wood-burning PM emission factors may include these IVOCs. Admittedly, the problem is even more complicated by the fact that many countries report wood-burning PM emission factors at high temperature conditions, so they

may not actually be capturing the IVOCs. Still, it's highly uncertain to what extent they are already measured. I urge the authors to update their discussion of these classes of compounds to better reflect some of the nuances we now understand better.

2. Line 99-118: I encourage the authors to add residential wood burning/cookstoves to their list, and possibly also the recent work on asphalt emissions (Khare et al., 2020).

3. Line 119-132: I think the authors get somewhat stuck on the SVOC portion of the ASOA problem in this paragraph and would do well to keep the broad focus on both IVOC-SOA and SVOC-SOA they have been introducing so far. For one thing, I'm not sure how important revising the (terrible) assumption of POA nonvolatility is for connecting urban PM to health impacts in the context of annual mean guidelines. Of course it's important to know how much of the PM started as an SVOC vapor for the purposes of control. But meanwhile, if we think that a portion of the SOA mass was emitted in the particle phase and then evaporated, oxidized and recondensed after dilution, then how does updating our conceptual picture to consider that portion volatile necessarily help us control it better – we could still control it with particulate filters. To me, the important reasons to update the conceptual model from nonvolatile POA to semivolatile are to 1) better track the composition of the OA because maybe it has different toxicity or efficiencies for losses as it is oxidized, 2) sensitivity to temperature and concentration swings might have an impact on urban scale versus suburban or rural exposure or diurnal timing of concentration peaks and thus impacts on human exposure. Adding in the SVOC and IVOC vapors helps us achieve a total mass balance on the amount of carbon with the potential to make SOA and this is really a separate point. In short, the authors could make it more clear in this paragraph, at least qualitatively, which sources of uncertainty they are most concerned about in previous estimations of PM mortality. Is it a) poor traditional POA models, b) undersampled SVOC and IVOC emissions from known sources, c) underestimated yields (i.e. vapor wall-losses, etc), d) missing or unacknowledged sources of vapor precursors or e) something else. Right now, it seems like (a) is their chief concern.

[Figure]

4. Line 141: A complete introduction or general description of the modeling approach is needed to begin the methods section. Before the authors get into the extreme details (e.g. how data were averaged), we readers would do well to learn what the basic idea of the study is going to be (i.e. parameterize ASOA in cities using campaign data, replace ASOA in GEOS-Chem with these results, plug new PM2.5 into relative risk and premature death parameterizations, assess the impact, and explore some key sensitivities). For example, on line 142, I'm not sure what 'values' are being discussed, how they were measured, or how they will be used.

5. Figures are introduced out of order in the methods section.

6. It looks to me like the emission ratio in Tables S5-S8 that were calculated with Eq. 3 are in most (though not all) cases well outside the range of measured emission ratios from other campaigns. For example, o-xylene in Table S5 is all as high or higher than the maximum observations, propene in Table S7 as well. The values for London in Table S8 are either below the minimum observed or above the maximum, depending on the species. Are these predictions expected by the authors? Can they be explained by variations among cities? I recommend calculating and reporting the performance of the Eq. 3 model in reproducing the observed values in Tables S5-S8. Also, what values of t were used to calculate the emission ratios in Eq. 3? I assume many values used and then all averaged together? Or were the values for each daily averaged and then a campaign average derived from that? What is the spread in the intermediate emission ratio values? I think this paragraph (lines 150-161) could be written more clearly to better describe the multiple levels of averaging and error analysis taking place here.

7. Lines 181-188: I appreciate the spirit of the leave-one-out sensitivity study and the results presented in Table S10. However, I do not think it accomplishes what the authors intend, which is to justify their regressed slope of 24.8. The reference to 95% confidence intervals seems misleading, perhaps because a clear null hypothesis is not stated. I'm not sure I've seen confidence intervals used to prove two slopes are statistically similar before, but I'd be interested to learn if the authors can show their

work. A conventional leave-one-out would calculate the error in predicting the removed point and then average the errors across all trials. I'm not sure how knowing this error statistic would be helpful either though, except to perhaps compare among similar leave-one-out analyses for the other slopes in Fig. 5. In my opinion, a better analysis would involve an assessment of the degree to which the Seoul data point is influencing the slope parameter. For example, the Cook's distance is commonly used in regression approaches to flag highly influential data points. If the point is determined to be influential, then the authors need to discuss what impact the change in slope from 24.8 to 34.0 has on the conclusions of the paper.

8. The SIMPLE model relies on having an accurate BTEX field for input. So how consistent were the HTAPv2 emission inputs with each of the measurement campaigns, allowing for expected deviations for year to year trends?

9. Why not add a supplemental figure showing the average spatial distribution of CO and R_btex emissions so readers can get a sense for which is driving the SIMPLE predictions in the various countries? I recommend at least plotting this as country averages, if not both country averages and grid cells. 10. Consider adding to the conclusions the ASOA-associated premature death estimates that you are most confident in.

Minor Comments:

1. Line 60: Rewrite "anthropogenic reactivity of specific volatile organic compounds" to "reactivity of specific anthropogenic volatile organic compounds" ?

2. Line 66: "results in up to. . ."

3. Line 67: "extrapolation" of what data specifically? Is it more informative to say "extrapolation from regions where detailed emission inventory data are available to other regions where uncertainties in emissions are larger." ?

4. Line 68: I agree that comprehensive air quality campaigns are certainly helpful and

possibly necessary, but it seems that robust national-scale institutions (government, academic, or private) are absolutely necessary to accurately catalogue emission factors and activity data to the level required to reduce the uncertainties discussed in this manuscript. Perhaps this sentence could be broadened to something like: "In addition to further development of institutional air quality management infrastructure, comprehensive air quality campaigns..."

5. Lines 104-106: Suggest rewriting: "Biogenic SOA (BSOA) in urban areas typically results from advection of regional background concentrations rather than processing of locally emitted biogenic VOCs."

6. Lines 116-118: Seltzer et al. (2020) is currently finalizing discussion in ACPD and presents a detailed VCP emission inventory for the U.S. Based on this, the authors may want to update this sentence to include that step forward, but it's their choice.

7. Line 119: "uncertainty on the (burden of -or- emissions of) ASOA precursors..."

8. Eq. 3: Recommend adding an exp subscript to [OH] here to make it clear that it is calculated from Eq. 2.

9. Table S10. Please indicate which slope is being shown here (delta_SOA/R_BTEX)

10. Line 201: Many of the BTEX values are modeled with equation 3 right? Please make this clear.

11. Line 213: C* range is not consistent with how IVOCs are usually defined.

12. Line 209-210: Based on the reference, it appears the authors are specifically referring to underestimation of IVOCs in the ambient. Please make that more clear in the sentence.

13. Line 214-216: It's unclear to me how the IVOCs and unspeciated SOA precursors relate to each other here. Are the authors saying they used SOA yields from Jathar et al (2014) to define the IVOC SOA yields uniformly for all C* bins? Please clarify.

[Figure]

For example, a clearer way of making that point might be, "SOA yields from IVOC oxidation were parameterized with data from n-tridecane for gasoline engines and n-pentadecane for diesel engines (Jathar et al., 2014)."

14. Lines 216-218: Should VOCs be IVOCs here? Again, aren't all the IVOCs in this study unspeciated? If so, why make the distinction?

15. Line 224: Why was the Huffman et al. (2009) distribution not used for the cooking VBS distribution?

16. Table S9: What is the HOA and Other POA mass normalized to? Shouldn't these also be normalized to CO, or is POA a separate variable in the inventories? POA is never mentioned in the SI in the discussion of the inventory development.

17. SI Line 70: The emission ratios are small, or the range is small?

18. Lines 250-273: Is the TSI parameterization acting simultaneously with the Ma parameterization or are they different cases that are explored? It seems like Ma et al. (2017) is used for IVOC SOA yields instead of Jathar et al. (2014). There are a lot of parameterizations, precursors classes and products in this model approach. I strongly recommend adding a table(s) explicitly specifying all of the SOA yields and the corresponding precursors used in this study.

19. Line 273: Recommend rephrasing "increase in mass of 0.99" to "change in mass of 0.99" or "decrease in mass of 1%".

20. Line 276-281: This opening sentence is overly dense and meandering. What is the point of the appositive, "for ASOA apportionment (Fig. 1)"? It seems redundant. Should the second "apportionment" be "attribution"? The last portion of the sentence, after the GEOS-Chem reference should be broken off into its own sentence.

21. Line 335: Recommend presenting Eq. 4 as the summation of premature deaths among all considered causes.

22. Lines 363-376: This discussion belongs before the introduction of the premature death and RR parameterizations. The reader will follow along better if the discussion of GEOS-Chem ends with how the hybrid PM2.5 will be estimated so that it can be fed into the health eqns.

23. Line 448-449: Is BTEX meant to be here twice?

24. Throughout paper: USOA (urban SOA?) should be replaced with ASOA. It especially shows up in figures.

25. Line 536: Are the bottom-up ASOA predictions from GEOS-Chem just replaced with the SIMPLE post-processed predictions? If so, please state this clearer. What happens to the nonvolatile POA from GEOS-Chem that I think is included in the PM2.5 (line 289)? I presume the authors consider absorptive partitioning artifacts to be less important than other uncertainties in this study?

26. Fig. 9: Australia is white in panel a. Does that indicate less than 1 death per year due to ASOA?

27. Section 4 title is incomplete

References

Khare, Peeyush, et al. "Asphalt-related emissions are a major missing nontraditional source of secondary organic aerosol precursors." Science advances 6.36 (2020): eabb9785.

Lu, Quanyang, Yunliang Zhao, and Allen L. Robinson. "Comprehensive organic emission profiles for gasoline, diesel, and gas-turbine engines including intermediate and semi-volatile organic compound emissions." Atmospheric Chemistry & Physics 18.23 (2018).

May, Andrew A., et al. "Gas‐particle partitioning of primary organic aerosol emissions: 3. Biomass burning." Journal of Geophysical Research: Atmospheres 118.19

(2013): 11-327.

Morino, Yu, et al. "Contributions of condensable particulate matter to atmospheric organic aerosol over Japan." Environmental science & technology 52.15 (2018): 8456-8466.

Seltzer, Karl M., et al. "Reactive Organic Carbon Emissions from Volatile Chemical Products." Atmospheric Chemistry and Physics Discussions (2020): 1-33.

---

## Referee Comment (RC2) · Anonymous Referee #3 · 3 Feb 2021

The study represents an attempt to estimate the premature mortality linked to Anthropogenic Secondary Organic Aerosols. Using 11 urban areas on three continents and specific volatile organic compounds emission ratios were estimated and a budget for ASOA is attempted. With the studied dataset the SIMPLE parameterization for ASOA in the GEOS-Chem model is updated to reproduce observed ASOA. Finally an attribution of ASOA PM2.5 premature deaths is attempted.

General comment:

My greatest concern for the specific study is the overall omission of solid fuel combustion in all calculations, both for ASOA production (emissions and subsequent processing/oxidation/ageing) as well as its contribution to premature mortality. Not only biomass burning for heating purposes but also forest fires, burning of crops etc. This leads to unaccounted emissions from urban areas such as Europe/US during winter from household heating but also from forested areas such as the Amazon, Canada, Siberia, Southeast Asia.

Specific comments:

1) Line 110-114: Isn't solid fuel combustion/biomass burning aged SOA considered as ASOA? According to Kodros et al. 2020 in active fire regions bbOOA increases by more than 50-60% from fast oxidation processes even in the dark. Significant contribution of primary BBOA oxidation to the oxygenated OA have also been identified in large urban centers such as Paris (Petit et al., 2014) and Athens (Stavroulas et al., 2019).

2) Line 119-132: Isn't the current study also under-predicting ASOA by ignoring bbOOA? Furthermore, there is also the additive effect of the different pollutants when considering premature mortality. For example, Kodros et al. 2017 estimate joint exposure from household solid fuel use and ambient PM2.5 pollution and find 18% more deaths than by separating household and ambient mortality calculations. Which shows that solid fuel combustions is important for mortality as well, not only for ASOA calculations.

3) Fig.5a and Line 174-180, Fig. 6 and line 423-428: Authors only mention the uncertainties in x- and y-axis values. Does really by removing just one point increases the slope that much? The y-axis has an upper value of 140 compared to x-axis of 6! Why only ∼25% of the observed ASOA can be associated with BTEX? What about the rest? Isn't this a solid proof that solid fuel combustion (BBOA) should definitely be taken into account?

4) Section 2.5.2 Once more, by not including solid fuel combustion in ASOA all the respective chemistry and oxidation is missing, losing 50-60% of SOA from fast oxidation of BBOA, even in the dark (NO3 radicals) (Kodros et al., 2020).

5) Line 578-579: How is the "model constrained to atmospheric observations for a more accurate contribution of ASOA" when an important source of ASOA such as solid fuel combustion is omitted?

Technical corrections:

Fig. 7 & 8: USOA? Should it be ASOA?

References

John K. Kodros, Dimitrios K. Papanastasiou, Marco Paglione, Mauro Masiol, Stefania Squizzato, Kalliopi Florou, Ksakousti Skyllakou, Christos Kaltsonoudis, Athanasios Nenes, Spyros N. Pandis: Rapid dark aging of biomass burning as an overlooked source of oxidized organic aerosol, Proceedings of the National Academy of Sciences Dec 2020, 117 (52) 33028-33033; DOI: 10.1073/pnas.2010365117

Stavroulas, I., Bougiatioti, A., Grivas, G., Paraskevopoulou, D., Tsagkaraki, M., Zarmpas, P., Liakakou, E., Gerasopoulos, E., and Mihalopoulos, N.: Sources and processes that control the submicron organic aerosol composition in an urban Mediterranean environment (Athens): a high temporal-resolution chemical composition measurement study, Atmos. Chem. Phys., 19, 901–919, https://doi.org/10.5194/acp-19-901-2019, 2019.

Petit, J.-E., Favez, O., Sciare, J., Canonaco, F., Croteau, P., Močnik, G., Jayne, J., Worsnop, D., and Leoz-Garziandia, E.: Submicron aerosol source apportionment of wintertime pollution in Paris, France by double positive matrix factorization (PMF2) using an aerosol chemical speciation monitor (ACSM) and a multi-wavelength Aethalometer, Atmos. Chem. Phys., 14, 13773–13787, https://doi.org/10.5194/acp-14-13773-2014, 2014.

Kodros, J. K., Carter, E., Brauer, M., Volckens, J., Bilsback, K. R., L'Orange, C., . . . Pierce, J. R. (2018). Quantifying the contribution to uncertainty in mortality attributed to household, ambient, and joint exposure to PM2.5 from residential solid fuel use.

GeoHealth, 2, 25– 39. https://doi.org/10.1002/2017GH000115

---

## Author Comment (AC1) · 30 Apr 2021

Please see attached.

Please also note the supplement to this comment:
https://acp.copernicus.org/preprints/acp-2020-914/acp-2020-914-AC1-supplement.pdf

---

## Author Response (AR1)

**Response to reviewers' comments on the paper "Anthropogenic Secondary Organic Aerosols Contribute Substantially to Air Pollution Mortality"**

We would like to thank the reviewers for their time and for their useful comments that have helped to improve and clarify our paper. For ease, comments from reviewers are in black, responses in blue, and new text added to paper in **bold blue**.

*Reviewer #1*

1.0 This manuscript leverages an impressive breadth and diversity of data to shed light on a critical public health and environmental policy question: how many premature deaths can be avoided annually with reductions in emissions of organic compounds that humans have direct control over? The methods and approaches that are developed are generally sound, although far from perfect, for this kind of high-level endeavor. My main issue with the paper in its current form are the confusing organization and presentation of ideas, some slight misplaced focus on particular organic aerosol model updates, and the decision to ignore solid fuel burning in the model formulation. These and other issues below should be addressed before publication.

We thank the reviewer for the overall positive review and the detailed input. We have replied to all the specific points below.

General Comments:

1.1 I think the main ideas in the paper are quite compelling: a) reconstruct measured SOA from in situ campaigns using correlations with likely predictors, b) incrementally improve a streamlined parameterization for SOA prediction (SIMPLE), c) integrate the SIMPLE predictions into a full-science CTM prediction of PM2.5 and use satellite data to further refine the predictions, d) feed those predictions to a premature death parameterization to quantify human health impact, and e) investigate key sensitivities. I would reorganize the entire paper so that the methods, results and discussion each flow in that order. Currently, the introduction gives little clue about how the pieces will fit together or the goals of the paper, beyond showing that ASOA is important. Much of (b) above is discussed inappropriately in the current results section 4. Manuscript sections 3, 4, and 5 contain quite a lot of methods discussion that should be moved out to section 2. For example, equations rarely belong in a results section. I could even suggest that most of section 2.2 and 2.4 be moved to SI. The details of the chemical mechanism used in 2.4 are a bit irrelevant once the SOA/R_BTEX enhancement ratio is confirmed for use in SIMPLE. If the mechanism were more sophisticated (e.g. HOM formation, carbon-conserving fragmentation to lower MW products, oligomerization, etc) then I think there would be more cause for focusing on it, but the schemes used here are relatively close to the SIMPLE approach in terms of one-way generation of SOA.

Perhaps some confusion arises from the fact that the paper has two sets of major results. The explanation of the parameters controlling the variability of ASOA at major locations is an important result by itself. The application of state-of-the-art methods (that apply the improved quantification of ASOA) to provide the first realistic estimate of the mortality associated specifically with ASOA is a second important result. Perhaps they could have been reported in two separate papers, but we decided to report them together. Therefore the structure of the paper does make sense, because (a) and (d) are both key results, and (b) and (c) are needed to connect those results.

We have nevertheless made an effort to streamline the structure of the paper to reduce possible confusion for some readers. We have added more discussion in the introduction to better frame the uncertainty in ASOA production impacting both models and the ability to apportion emissions to reduce premature mortality. We have both expanded the methods section (e.g., moving some sections, such as 4.1, into methods) and moved some methods into the SI (those that seemed less important for understanding the entirety of the study, e.g., Error Analysis, Box Model, and GEOS-Chem Description). Further, we have expanded the discussion in Sect. 4 concerning the discussion of implementation and improvement of the SIMPLE model compared to what is currently used in GEOS-Chem.

The text added is:

In introduction:

[revised manuscript text omitted]

Other updates for the introduction can be found in other specific comments (e.g., R1.6, R1.7, R1.8, and R1.9).

For Sect. 3.1, added information can be found in R1.3, and for Sect. 3.2, added information can be found in R1.2.

In Sect. 4, about the SIMPLE model improvements, the following has been added:

**"The "improved" SIMPLE shows higher ASOA compared to the default VBS GEOS-Chem (Fig. 6a,b). In areas strongly impacted by urban emissions (e.g., Europe, East Asia, India, east and west coast US, and regions impacted by Santiago, Chile, Buenos Aires, Argentina, Sao Paulo, Brazil, Durban and Cape Town, South Africa, and Melbourne and Sydney, Australia), the "improved" SIMPLE model predicts up to 14 µg m$^{-3}$ more ASOA, or ~30 to 60 times more ASOA than the default scheme (Fig. 6c,d). As shown in Fig. 1, during intensive measurements, the ASOA composed 17-39% of PM$_1$, with an average**

**contribution of ~25%. The default ASOA scheme in GEOS-Chem greatly underestimates the fractional contribution of ASOA to total PM$_{2.5}$ (<2%; Fig. 6e). The "improved" SIMPLE model greatly improves the predicted fractional contribution, showing that ASOA in the urban regions ranges from 15-30%, with an average of ~15% for the grid cells corresponding to the urban areas investigated here (Fig. 6f). Thus, the "improved" SIMPLE predicts the fractional contribution of ASOA to total PM$_{2.5}$ far more realistically, compared to observations. As discussed in Sect. 2.3 and Eq. 11, having the model accurately predict the fractional contribution of ASOA to the total PM is very important, as the total PM$_{2.5}$ is derived from satellite-based estimates (van Donkelaar et al., 2015), and the model fractions are then applied to those total PM$_{2.5}$ estimates. The ability for the "improved" SIMPLE model to better represent the ASOA composition provides confidence attributing the ASOA contribution to premature mortality."**

The figures that have been added to the paper to accompany the text above are:

[Figure]

**Figure 6. (a) Annual average modeled ASOA using the default VBS. (b) Annual average modeled ASOA using the updated SIMPLE model. (c) Difference between annual average modeled updated SIMPLE and default VBS. (d) Ratio between annual average modeled updated SIMPLE and default VBS. (e) Percent contribution of annual average modeled ASOA using default VBS to total modelled PM$_{2.5}$. (f) Percent contribution of annual average modeled ASOA using updated SIMPLE to total modelled PM$_{2.5}$.**

Finally, added text for Sect. 5 can be found in R1.3.

1.2 The authors repeatedly compare their updated approach including semivolatile POA to previous efforts to assess ASOA impacts on human health using nonvolatile POA assumptions.

The implication here is that treating POA as semivolatile might be as important to getting ASOA correct as the dramatic increase in ASOA precursors. But one look at Fig. 6 shows that in most cases it's the way BTEX, IVOCs and Other Aromatics are treated (emissions, SOA yields, aging) that is really driving the ASOA formation. So while I think the update to semivolatile POA is a good one and it gives room for greater ASOA production, I think the authors focus on it too much in this study (see specific comment #3 below and lines 574-581 for examples that should be addressed). Instead there should be much more focus in the budget section on how the parameters have chosen for their VCP emissions and SOA chemistry scheme are driving larger IVOC and VOC contributions and how confident they are in those parameters. For the SIMPLE and GEOS-Chem sections, the key sensitivity is the SOA/R_BTEX ratio (see specific comment #7) and there should be more discussion on its impact. Also, the SIMPLE rate constant k is parameterized from the CalNex data, correct? Why was that not revisited and optimized for performance among all the measurement campaigns?

Perhaps there is some misunderstanding of Fig. 6 from the ACPD submission. So first, we would like to clarify Fig. 6. In Fig. 6, the two panels are not two different ways the VOCs are treated, but two different ways to apportion the same VOCs: on the left by type of species, on the right by the source of the species.

We have added the following text at the beginning of Sect. 3.2 to clarify this point:

**"To investigate the correlation between ASOA and $R_{BTEX}$, a box model using the emission ratios from BTEX (Table S5), other aromatics (Table S8), IVOCs (Sect. S1), and SVOCs (Sect. S1) was run for five urban areas:  New York City, 2002, Los Angeles, Beijing, London, and New York City, 2015 (see Sect. S1 and S3 for more information). The differences in the results shown in Fig. 4 are due to differences in the emissions for each city."**

We believe the point made in lines 573 - 575 is  very important, as many models may get the total OA approximately correct while getting SOA vs POA incorrect (e.g., Hodzic et al., 2020). This in turn can mean that focus of emission controls may be misplaced on reducing POA while neglecting the emissions that lead to the observed ASOA concentrations (e.g., IVOCs from traditional and non-traditional sources). Though POA and IVOC emissions may sometimes originate from similar sources, e.g., diesel (Zhao et al., 2014), the IVOCs will also be emitted from sources that do not include POA, e.g., VCPs (McDonald et al., 2018).

We agree that more emphasis by the community on VCP and IVOC emissions and their SOA production is important. The present paper can also be viewed as a follow-up study to McDonald et al. (2018) that shows the applicability of VCP emissions outside the United States.

Further, we have added the following text to further address these points:

First, about SVOCs:

**"Note, the emissions investigated here ignore any oxygenated VOC emissions not associated with IVOCs and SVOCs due to the challenge in estimating the emission ratios for these compounds (de Gouw et al., 2018). Further, SVOC emission ratios are estimated from the average POA observed by the AMS during the specific campaign and scaled by profiles in literature for a given average temperature and average OA (Robinson et al., 2007; Worton et al., 2014; Lu et al., 2018). As most of the campaigns had an average OA between 1 and 10 μg m$^{-3}$ and temperature of ~298 K, this led to the majority of the estimated emitted SVOC gases in the highest SVOC bin. However, this does not lead to SVOCs dominating the predicted ASOA due to taking into account the fragmentation and overall yield from the photooxidation of SVOC to ASOA."**

We agree the key parameter is SOA/$R_{BTEX}$, and the purpose of Sect. 3.2 is to explore this ratio. As shown in Fig. 6 of the original manuscript, assuming a constant ratio for SOA/$R_{BTEX}$ (the slope from Fig. 5), we are able to explain most of the observed ASOA with the box model and emission inventories. We have added the following text to clarify this point:

**"This investigation shows that the bottom-up calculated ASOA agrees with observed top-down ASOA within 15%. As highlighted above, this ratio is explained by the co-emissions of IVOCs with BTEX from traditional sources (diesel, gasoline, and other fossil fuel emissions) and VCPs (Fig. 5) along with similar rate constants for these ASOA precursors (Table S12). Thus, the ASOA/$R_{BTEX}$ ratio obtained from Fig. 2 results in accurate predictions of ASOA for the urban areas evaluated here, and this value can be used to better estimate ASOA with chemical transport models (Sect. 4)."**

The rate constant of the SIMPLE model, as stated in Line 513 - 520 in the original manuscript, was originally parameterized to the observations from both Mexico City and Los Angeles. It is also generally consistent with observations of ASOA formation with a time scale of 1 day in other studies (e.g., de Gouw et al., 2005; DeCarlo et al., 2010; Nault et al., 2018; Schroder et al., 2018).

The following has been added to the text to also reflect this point:

**"This rate constant is also consistent with observed ASOA formation time scale of ~1 day that has been observed across numerous studies (e.g., de Gouw et al., 2005; DeCarlo et al., 2010; Hayes et al., 2013; Nault et al., 2018; Schroder et al., 2018)."**

1.3 The authors spend some time in the discussion addressing the fact that solid fuel combustion emissions are missing from this study. I'm still very concerned that much of the global results they show are corrupted by this emission, not just in southern Asia and Africa. Large regions of Northern/Western Europe and North America will also be affected by residential wood fuel burning, especially in the winter. The authors should at least justify their omission of solid fuels for the measurement campaigns citing tracer analyses, for example. To address global comparisons, can you add a reference to SOA/CO ratios for wood combustion and comment on their similarity or difference from what has gone into SIMPLE for this study?

This is not quite correct. Unfortunately, we did not emphasize enough that two studies used to constrain the $\Delta SOA/\Delta CO$ vs $R_{BTEX}/\Delta CO$ slope shown in Fig. 5a, and thus constrains the updated SIMPLE model, are from campaigns that include large contributions from solid fuel combustion. These include a wintertime campaign in the Northeast US (Schroder et al., 2018) and a late winter, early spring campaign in China (Hu et al., 2013). Both of these studies were strongly impacted by solid fuel combustion, as highlighted in Table S9 in the "Other POA" category (for NYC as we do not have reliable emissions inventory for the observations from Hu et al. (2013)).

Importantly, as we discuss with the updated analysis on the influence of any one point for the slope shown in Fig. 5a (see response to R1.12), the data from these two studies are very close to the slope and do not influence the results. Thus, within the limitations of the available datasets, solid fuels are included and do not result in deviations for the parameterization derived in this study. Clearly it is useful to investigate this point further using data from future campaigns, as we are not aware of any other past campaigns with complete enough data to perform these analyses.

We have added the following text to the revised paper to explain this point in more detail:

**"An important aspect of this study is that most of these observations occurred during spring and summer, when solid fuel emissions are expected to be lower (e.g., Chafe et al., 2015; Lam et al., 2017; Hu et al., 2020). Further, the most important observations used here are during the afternoon, investigating specifically the photochemically produced ASOA. These results here might partially miss any ASOA produced through nighttime aqueous chemistry or oxidation by nitrate radical (Kodros et al., 2020). However, two of the studies included in our analysis, Chinese Outflow (CAPTAIN, 2011) and New York City (WINTER, 2015), occurred in late winter/early spring, when solid fuel emissions were important (Hu et al., 2013; Schroder et al., 2018). We find that these observations lie within the uncertainty in the slope between ASOA and $R_{BTEX}$ (Fig. 2a). Their photochemically produced ASOA observed under strong impact from solid fuel emissions shows similar behavior as the ASOA observed during spring and summer time. Thus, given the limited datasets currently available, photochemically produced ASOA is expected to follow the relationship shown in Fig. 2a and is expected to also follow this relationship for regions**

impacted by solid fuel burning. Future comprehensive studies in regions strongly impacted by solid fuel burning are needed to further investigate photochemical ASOA production under those conditions."

In addition, we have also added the following text to section 5 to address the potential uncertainties:

"Solid fuels are used for residential heating and cooking, which impact the outdoor air quality as well (Hu et al., 2013, 2016; Lacey et al., 2017; Stewart et al., 2020), and which also lead to SOA (Heringa et al., 2011). As discussed in Sect. 3.1, though the majority of the studies evaluated here occurred in spring to summer time, when solid fuel emissions are decreased, two studies occurred during the winter/early spring time, where solid fuel emissions were important (Hu et al., 2013; Schroder et al., 2018). These studies still follow the same relationship between ASOA and $R_{BTEX}$ as the studies that focused on spring/summer time photochemistry. Thus, the limited datasets available indicate that photochemically produced ASOA from solid fuels follow a similar relationship to that from other ASOA sources.

Also, solid fuel sources are included in the inventories used in our modeling. For the HTaP emission inventory used here (Janssens-Maenhout et al., 2015), small-scale combustion, which includes heating and cooking (e.g., solid-fuel use), is included in the residential emission sector. Both CO and BTEX are included in this source, and can account for a large fraction of the total emissions where solid-fuel use may be important (Fig. S15). Thus, as CO and BTEX are used in the updated SIMPLE model, and campaigns that observed solid-fuel emissions fall within the trend for all urban areas, the solid-fuel contribution to photochemically-produced ASOA is accounted for (as accurately as allowed by current datasets) in the estimation of ASOA for the attribution to premature mortality.

Note that recent work has observed potential nighttime aqueous chemistry and/or oxidation by nitrate radical from solid fuel emissions to produce ASOA (Kodros et al., 2020). Thus, missing this source of ASOA may lead to an underestimation of total ASOA versus the photochemically-produced ASOA we discuss here, leading to a potential underestimation in the attribution of ASOA to premature mortality. From the studies that investigated "night-time aging" of solid-fuel emissions to form SOA, we predict that the total ASOA may be underestimated by 1 to 3 µg m$^{-3}$ (Kodros et al., 2020). This potential underestimation, though, is less than the current underestimation in ASOA in GEOS-Chem (default versus "Updated" SIMPLE)."

Have also added the following figure in SI to go with the text above:

[Figure]

[Figure]

**Figure S15. (top) Fractional contribution of CO emissions from residential sources to total emission sources from HTAP. (bottom) Fractional contribution of BTEX emissions from residential sources to total emission sources from HTAP. Residential sources include small-scale combustion, such as heating and cooking, which may include solid-fuel emissions.**

1.4. Why does SIMPLE model now rely on BTEX alone? It seems to be doing better overall than when it just relied on CO, but why not use more than one variable with BTEX to develop a multilinear fit for the SOAP emissions? For example, Seoul is likely a problematic point in the SOA vs. BTEX regression (see specific comment #7). It's driving down the ratio and thus probably leading to under-representing impacts in the Northeast US and LA. Taken together,

Figs. 3 and 4 suggest that in Seoul there are SOA sources associated with CO emissions that are not as highly associated with BTEX. Using too many independent variables would surely end up overfitting, but why not add 1 or 2 key variables (like CO and POA) since you have a good idea that the relative contributions of sources (e.g. vehicles, VCPs, and solid fuel use) vary from city to city?

As we discuss in response to 1.12, Seoul is not driving the relationship and thus is not a problematic point.

Also, there is perhaps some confusion. The updated version of SIMPLE does not rely on BTEX alone, but rather on both BTEX and CO emissions (e.g., eq. 7 in the ACPD version) as well as OH concentrations within the model. This is an improvement from the original SIMPLE model, in which the parameterization only depended on CO and the model OH fields.

We do not see a reason for a more complex parameterization, since the available data are well-fit with the updated parameterization proposed in the paper. Of course more complex parameterizations could be devised, but they would be underconstrained by the observations. Indeed, Fig. 2 in the ACPD version shows that BTEX is co-emitted in both "traditional" and "non-traditional" sources (fossil fuel versus VCP), and both these sources account for the majority of the predicted ASOA (Fig. 6 of ACPD version). Finally, most emission inventories have BTEX, providing a more straightforward method to implement this parameterization into chemical transport models.

We have added the following text to address this point:

**"The $R_{aromatics}/\Delta CO$ allows a dynamic calculation of the $E(VOC)/E(CO) = SOA/\Delta CO$. Hodzic and Jimenez (2011) and Hayes et al. (2015) used a constant value of 0.069 g g$^{-1}$, which worked well for the two cities investigated, but does not for the expanded dataset studied here. Thus, both the aromatic emissions and CO emissions are used in this study to better represent the variable emissions of ASOA precursors (Fig. S5)."**

The following figure has been added to address the comment as well:

[Figure]

**Figure S5. (a) Annually average CO emissions from HTAP. (b) Annually average benzene, toluene, and xylenes (BTX) emissions, weighted by their OH reaction rate**

$$\left(E_{weight} = N\frac{\sum_i E_i k_{OH,i}}{\sum_i k_{OH,i}}, \; i = B, T, X \; ; \; N=3\right).$$

1.5. Line 455-457: The observation about SVOCs is difficult to believe based on existing NMOG and POA profiles in literature. I have yet to see a volatility profile for any source where the SVOC accounted for half of the total ASOA precursor, let alone 88%. Is there something unexpected going on with the CO normalization of POA vs. VOCs here? I confess this one catches me completely by surprise, and likewise the large influence of SVOCs in Fig. 3 looks strange as well. The authors have made an emphatic case for the dominant and growing role of VCPs. Wouldn't these be overwhelmingly VOCs and IVOCs? Cooking emissions are used to explain this to some degree, but if the Robinson et al. (2007) profile is used for cooking emissions, I would expect a lot more 10^3 and 10^4 C* compounds. Regardless, SVOC should probably be included on Fig. 2 as a separate series like BTEX and IVOC. And I recommend adding more description about how SVOC smission ratios are derived.

There are multiple prior publications, where ASOA formed from SVOCs were accounted for by several tens of percent of the total SOA (e.g., Dzepina et al., 2009; Ma et al., 2017). As stated in that section "Beijing has the highest contribution of SVOCs to ASOA precursors due to the use of solid fuels and cooking emissions (Hu et al., 2016)." Thus these results are not so surprising. They are correct and are based on the detailed inventories reported here.

We have moved Fig. 3 from the main paper to the SI, as it is not a finding but more a tool that was used to estimate ASOA. Part of the reason is that for the emissions with C* greater than $10^6$ μg m$^{-3}$ only include the VOCs reported in the SI, as the campaigns used here either had missing oxygenated VOCs and/or the challenge of estimating oxygenated VOC emission ratios (e.g., de Gouw et al., 2018; McDonald et al., 2018). Not including OVOCs may lead to an underestimation of the emission ratios at high volatility.

We have added the following text to the SI to describe how the SVOC emission profile was determined:

**"To estimate the SVOC mass concentration in equilibrium with the POA (Table S9) in each bin, the POA mass concentration is first multiplied by the fraction of POA measured in each bin from literature. This yields the concentration of POA for that specific volatility bin. Then the total POA + SVOC concentration for that bin is obtained divided by the amount of material found in the particle phase for that bin for the average temperature (~298 K) and OA mass concentration (~10 μg m$^{-3}$). Then, the gas-phase SVOC concentration is calculated by multiplying the total concentration by the gas-phase fraction. Thus, e.g., SVOC in the C* = 100 μg m$^{-3}$ bin, ~91% of the SVOC mass will be found in the gas-phase."**

Specific Comments:

1.6 Line 95-98: These generalities about IVOCs and SVOCs are perhaps useful for an introduction for those who may be unfamiliar but based on more current understanding of emissions sampling and speciation, they may be more confusing than helpful. Lu et al. (2018) show in their Fig. 1 that most of the IVOC would have missed the filters for the vehicles they studied, but much of the SVOC is expected to be captured by the filters. Even more SVOC would presumably be captured for stationary sources at conditions relevant for "condensable particulate matter" measurements (i.e. low dilution, cooled temperatures); see Morino et al. (2018). As for IVOCs, VBS profiles for biomass burning sources like those in May et al. (2013)

show that IVOCs are probably included in many if not most PM emission factors measured for those sources if the emissions are not diluted enough. The authors here are not focused on wildfires, but certainly cooking/residential wood-burning PM emission factors may include these IVOCs. Admittedly, the problem is even more complicated by the fact that many countries report wood-burning PM emission factors at high temperature conditions, so they may not actually be capturing the IVOCs. Still, it's highly uncertain to what extent they are already measured. I urge the authors to update their discussion of these classes of compounds to better reflect some of the nuances we now understand better.

We have removed that line and have expanded the discussion of S/IVOCs in the introduction to read:

"Many of these prior studies generally investigated AVOC with high volatility, where volatility here is defined as the saturation concentration, C*, in µg m$^{-3}$ (de Gouw et al., 2005; Volkamer et al., 2006; Dzepina et al., 2009; Freney et al., 2014; Woody et al., 2016). More recent studies have identified lower volatility compounds in transportation-related emissions (e.g., Zhao et al., 2014, 2016b; Lu et al., 2018). These compounds have been broadly identified as intermediate-volatile organic compounds (IVOCs) and semi-volatile organic compounds (SVOCs). IVOCs have a C* generally of $10^3$ to $10^6$ µg m$^{-3}$ while SVOCs have a C* generally of 1 to $10^2$ µg m$^{-3}$. Due to their lower volatility and functional groups, these classes of compounds generally form ASOA more efficiently than traditional, higher volatile AVOCs; however, S/IVOCs have also been more difficult to measure (e.g., Zhao et al., 2014; Pagonis et al., 2017; Deming et al., 2018). IVOCs generally have been the more difficult of the two classes to measure and identify as these compounds cannot be collected onto filters to be sampled off-line (Lu et al., 2018) and generally show up as unresolved complex mixture for in-situ measurements using gas-chromatography (GC) (Zhao et al., 2014). SVOCs, on the other hand, can be more readily collected onto filters and sampled off-line due to their lower volatility (Lu et al., 2018). Another potential issue has been an under-estimation of the S/IVOC aerosol production, as well as an under-estimation in the contribution of photochemically produced S/IVOC from photooxidized "traditional" VOCs, due to partitioning of these low volatile compounds to chamber walls and tubing (Krechmer et al., 2016; Ye et al., 2016; Liu et al., 2019). Accounting for this under-estimation increases the predicted ASOA (Ma et al., 2017). The inclusion of these classes of compounds have led to improvement in some urban SOA budget closure; however, many studies still have indicated a general short-fall in ASOA budget even when including these compounds from transportation-related emissions. (Dzepina et al., 2009; Tsimpidi et al., 2010; Hayes et al., 2015; Cappa et al., 2016; Ma et al., 2017; McDonald et al., 2018)."

1.7 Line 99-118: I encourage the authors to add residential wood burning/cookstoves to their list, and possibly also the recent work on asphalt emissions (Khare et al., 2020).

We have added the following:

**". . . as well as cooking emissions (Hayes et al., 2015), asphalt emissions (Khare et al., 2020), and solid fuel emissions from residential wood burning and/or cookstoves (e.g., Hu et al., 2013, 2020; Schroder et al., 2018). . ."**

1.8 Line 119-132: I think the authors get somewhat stuck on the SVOC portion of the ASOA problem in this paragraph and would do well to keep the broad focus on both IVOC-SOA and SVOC-SOA they have been introducing so far. For one thing, I'm not sure how important revising the (terrible) assumption of POA nonvolatility is for connecting urban PM to health impacts in the context of annual mean guidelines. Of course it's important to know how much of the PM started as an SVOC vapor for the purposes of control. But meanwhile, if we think that a portion of the SOA mass was emitted in the particle phase and then evaporated, oxidized and recondensed after dilution, then how does updating our conceptual picture to consider that portion volatile necessarily help us control it better - we could still control it with particulate filters. To me, the important reasons to update the conceptual model form nonvolatile POA to semivolatile are to 1) better track composition of the OA because maybe it has different toxicity or efficiencies for losses as it is oxidized, 2) sensitivity to temperature and concentration swings might have an impact on urban scale versus suburban or rural exposure or diurnal timing of concentration peaks and thus impacts on human exposure. Adding in the SVOC and IVOC vapors helps us achieve a total mass balance on the amount of carbon with potential to make SOA and this is really a separate point. In short, the authors could make it more clear in this paragraph, at least qualitatively, which sources of uncertainty they are most concerned about in previous estimations of PM mortality. Is it a) poor traditional POA models, b) undersampled SVOC and IVOC emissions from known sources, c) underestimated yields (i.e. vapor wall-losses, etc), d) missing or unacknowledged sources of vapor precursors or e) something else. Right now, it seems like (a) is their chief concern.

We have addressed some of these concerns ((b) and (c)) in response to 1.6. We have softened this section to be less focused on SVOC, and instead the under estimation of SOA most likely due to IVOC and "non-traditional" sources. We have changed it to clarify:

**"Due to the uncertainty on the emissions of ASOA precursors and on the amount of ASOA formed from them, the number of premature deaths associated with urban organic emissions is largely unknown. Since numerous studies have shown the importance of VCPs and other non-traditional VOC emission sources, efforts have been made to try to improve the representation and emissions of VCPs (Seltzer et al., 2021), which can reduce the**

**uncertainty in ASOA precursors and the associated premature deaths estimations. Currently, most studies have not included ASOA realistically (e.g., Lelieveld et al., 2015; Silva et al., 2016; Ridley et al., 2018) in source apportionment of the premature deaths associated with long-term exposure of $PM_{2.5}$. These models represented total OA as non-volatile POA and "traditional" ASOA precursors (transportation-based VOCs), which largely under-predict ASOA (Ensberg et al., 2014; Hayes et al., 2015; Nault et al., 2018; Schroder et al., 2018) while over-predicting POA (e.g., Hodzic et al., 2010; Zhao et al., 2016a; Jathar et al., 2017). This does not reflect the current understanding that POA is volatile and contributes to ASOA mass concentration (e.g., Grieshop et al., 2009; Lu et al., 2018). Though the models are estimating total OA correctly (Ridley et al., 2018; Hodzic et al., 2020; Pai et al., 2020), the attribution of premature deaths to POA instead of SOA formed from "traditional" and "non-traditional" sources, including IVOCs from both sources, could lead to regulations that may not target the emissions that would reduce OA in urban areas. As $PM_1$ and SOA mass are highest in urban areas (Fig. 1), also shown in Jimenez et al. (2009), it is necessary to quantify the amount and identify the sources of ASOA to target future emission standards that will optimally improve air quality and the associated health impacts. As these emissions are from human activities, they will contribute to SOA mass outside urban regions and to potential health impacts outside urban regions as well."'**

1.9 Line 141: A complete introduction or general description of the modeling approach is needed to begin the methods section. Before the authors get into the extreme details (e.g. how data were averaged), we reader would do well to learn what the basic idea of the study is going to be (i.e. parameterize ASOA in cities using campaign data, replace ASOA in GEOS-Chem with these results, plug new PM2.5 into relative risk and premature death parameterizations, assess the impact, and explore some key sensitivities). For example, on line 142, I'm not sure what 'values' are being discussed, how they are measured, or how they will be used.

We have added the following to introduce everything discussed in Sect. 2:

**"Here, we introduce the ambient observations from various campaigns used to constrain ASOA production (Sect. 2.1), description of the simplified model used in CTMs to better predict ASOA (Sect. 2.2), and description of how premature mortality was estimated for this study (Sect. 2.3). In the SI, the following can be found: description of the emissions used to calculate the ASOA budget for five different locations (Sect. S1), description of how the ASOA budget was calculated for the five different locations (Sect. S2), description of the CTM (GEOS-Chem) used in this study (Sect. S3 - S4), and error analysis for the observations (Sect. S5)."**

1.10 Figures are introduced out of order in the methods section.

We have removed reference to figures in Sect. 3 to instead be references to the sections themselves.

1.11 It looks to me like the emission ratio in Tables S5-S8 that were calculated with Eq. 3 are in most (though not all) cases well outside the range of measured emission ratios from other campaigns. For example, o-xylene in Table S5 is all as high or higher than the maximum observations, propene in Table S7 as well. The values for London in Table S8 are either below the minimum observed or above the maximum, depending on the species. Are these predictions expected by the authors? Can they be explained by variations among cities? I recommend calculating and reporting the performance of the Eq. 3 model in reproducing the observed values in Table S5-S8. Also, what values of t were used to calculate the emission ratios in Eq. 3? I assume many values used and then all averaged together? Or were the values for each daily averaged and then a campaign average derived from that? What is the spread in the intermediate emission ratio values? I think this paragraph (lines 150-161) could be written more clearly to better describe the multiple levels of averaging and error analysis taking place here.

Prior studies have shown very large variability across different cities for the same compound, e.g., Bon et al. (2011) showed an order of magnitude difference in ethane emission ratios across three different studies and a factor of ~20 difference in propane across three different studies. Further, as shown in Bon et al. (2011) and Apel et al. (2010) and highlighted in the table made below, there can be large differences, especially for the alkanes, for the same location, depending on how the emission ratio was determined. Thus, there can be large variability across cities as well as potential uncertainty, which most prominent for the longer lived compounds that minimally contribute to ASOA production.

We have added the following text in the SI to discuss and clarify this point:

**"A further potential source of uncertainty in this analysis is the calculated VOC emission ratios for the studies that did not have ratios published previously (Houston 2000, London, Houston 2013, and Seoul). To investigate how well Eq. 3 does in estimating the VOC emission ratios, a comparison of the estimated VOC emission ratios versus previously published ratios for two different cities, Mexico City (Apel et al., 2010; Bon et al., 2011) and Los Angeles (de Gouw et al., 2017) was made (Table S10). Also, for Mexico City, two locations, an urban and a suburban site, were compared both against each other (Apel et al., 2010; Bon et al., 2011) and the calculated values from Eq. 3.**

**First, as shown in Table S10, even for the same location (suburban Mexico City), different values in the emission ratio, especially for the alkanes, can be observed, by as much as a factor of 7. This can be partially explained by differences in how the emission ratios were**

determined. For both Apel et al. (2010) and Bon et al. (2011), the authors took the slope of VOCs versus CO and used different regression techniques and different time periods. Comparing their technique with ours, we generally estimate VOC emission ratios within 50% of the reported values, and the estimation improves for shorter lived compounds (e.g., aromatics). However, de Gouw et al. (2017) more carefully took chemistry into consideration for any potential losses of the VOCs prior to observation to determine emission ratios, similar to this study. We believe the comparison with de Gouw et al. (2017) provides a more useful comparison in the method presented here. We find, at most, a 30% difference in the emission ratios, with an average difference of 4±15% for all compounds. Thus, from this analysis, we conclude that (1) there is large variability in VOC emission ratios across urban areas around the world, which has been highlighted in other studies (Warneke et al., 2007), and (2) the method that considers losses of VOCs is the more accurate procedure to estimate VOC emissions and leads to the best reproducibility across studies and lowest uncertainty (< 30%, ~4% on average).”

The following table has been added to the SI:

**Table S10. Comparison of estimated VOC emission ratios from two studies from Mexico City (Apel et al., 2010; Bon et al., 2011), one study from Los Angeles (de Gouw et al., 2017), and this study.**

| VOC Ratio | Apel et al. (2010) Downtown MC | This Study | Apel et al. (2010) Suburbs MC | Bon et al. (2011) Outskirt MC | This Study | de Gouw et al. (2017) LA | This Study |
|---|---|---|---|---|---|---|---|
| Ethane | 7.4 | 8.2 | 3.0 | 21.5 | 8.2 | 16.5 | 18.9 |
| Propane | 41.5 | 36.9 | 49.3 | 61.7 | 38.4 | 13.4 | 14.0 |
| n-Butane | 15.1 | 14.9 | 15.3 | 21.7 | 14.1 | 5.0 | 5.7 |
| i-Butane | 4.8 | 4.8 | 5.3 | 7.2 | 4.9 | 3.2 | 3.5 |
| n-Pentane | 2.1 | 2.9 | 2.1 | 2.5 | 2.1 | 3.4 | 3.4 |
| i-Pentane | 2.7 | 3.6 | 3.2 | 3.3 | 3.1 | 8.7 | 7.8 |
| n-Hexane | 1.5 | 1.9 | 1.3 | 1.5 | 1.2 | 1.4 | 1.7 |
| Ethene | 8.4 | 6.1 | 7.9 | 7.0 | 7.1 | 11.2 | 9.6 |
| Propene | 2.6 | 1.3 | 2.9 | 3.0 | 1.6 | 4.1 | 3.9 |
| Benzene | 0.9 | 1.0 | 1.2 | 1.2 | 1.3 | 1.3 | 1.4 |
| Toluene | 7.5 | 9.2 | 5.2 | 4.2 | 4.1 | 3.4 | 3.0 |

| | | | | | | | |
|---|---|---|---|---|---|---|---|
| **Ethylbenzene** | 0.9 | 0.8 | 0.4 | 4.3* | 0.4 | 0.6 | 0.6 |
| **m+p-Xylene** | 1.1 | 0.7 | 0.5 | No Data | 0.4 | 2.1 | 1.9 |
| **o-Xylene** | 0.4 | 0.2 | 0.2 | No Data | 0.2 | 0.8 | 0.7 |
| **Trimethylbenzenes** | No Data | No Data | No Data | No Data | No Data | 1.6 | 1.1 |
| **Ethyltoluenes** | No Data | No Data | No Data | No Data | No Data | 0.6 | 0.4 |
| **Propylbenzene** | No Data | No Data | No Data | No Data | No Data | 0.1 | 0.1 |

*In Bon et al. (2011), they reported the sum of C8 aromatics, which is the sum of ethylbenzene and xylenes

1.12 Lines 181-188: I appreciate the spirit of the leave-one-out sensitivity study and the results presented in Table S10. However, I do not think it accomplishes what the authors intended, which is to justify the regressed slope of 24.8. The reference to 95% confidence intervals seems misleading, perhaps because a clear null hypothesis is not stated. I'm not sure I've seen confidence intervals used to prove two slopes are statistically similar before, but I'd be interested to learn if the authors can show their work. A conventional leave-one-out would calculate the error in predicting the removed point and then average the errors across all trials. I'm not sure how knowing this error statistic would be helpful either though, except to perhaps compare among similar leave-one-out analyses for the other slopes in Fig. 5. In my opinion, a better analysis would involve an assessment of the degree to which the Seoul data point is influencing the slope parameter. For example, the Cook's distance is commonly used in regression approaches to flag highly influential data points. If the point is determined to be influential, then the authors need to discuss what impact the change in slope from 24.8 to 34.0 has on the conclusion of the paper.

The equation we had used to investigate statistical difference in slopes was:

$$t = \frac{b_1 - b_2}{\sqrt{s_{b_1}^2 + s_{b_2}^2}}$$

Where $b_i$ is the slope and $s_i$ is the standard deviation about the slope.

In addition, we have also conducted the Cook's distance test, of which we were not aware. We appreciate the reviewer bringing this statistical tool to our knowledge. We have found that the T-test, Cook's distance test, and the difference in fits test all show that the one point from Seoul is not an outlier. We have added the following table and text to the paper:

**Table S11. Statistical analysis of the data used in Fig. 2 to determine if any point is influencing the slope, using the T-test, Cook's Distance test, and Difference in Fits test. For**

**the T-test, the point is influential if the t value is < 0.05 while for the Cook's Distance and Difference in Fits test, the point is influential if the value is > 1.**

| Campaign | T-test | Cook's Distance | Difference in Fits |
|---|---|---|---|
| NE US Ship | 0.63 | 0.06 | -0.29 |
| NE US Aircraft | 0.12 | 0.27 | 0.73 |
| Mexico City | 0.39 | 0.06 | 0.33 |
| Los Angeles | 0.32 | 0.08 | 0.38 |
| Changdao Island, China | 0.41 | 0.09 | -0.38 |
| Beijing | 0.42 | 0.06 | -0.32 |
| London | 0.31 | 0.13 | -0.48 |
| NYC | 0.90 | 0.00 | -0.05 |
| Seoul | 0.99 | 0.00 | 0.01 |

We have updated the text to say:

**"Statistical analysis for the influence of the data from Seoul on the figure was conducted, including a T-test, Cook's Distance test, and Difference in Fits test (Table S11). All three statistical tests show that the data from Seoul (and all the data in general) is not overly influencing the reported slope."**

1.13 The SIMPLE model relies on having an accurate BTEX field for input. So how consistent were the HTAPv2 emission inputs with each of the measurement campaigns, allowing the expected deviations for year to year trends?

We have added the following text in the SI:

**"Analysis of the HTAP emissions, compared to other emission inventories, generally showed the highest correlation with observations ($R^2 = 0.54$), versus the other inventories (CEDS $R^2 = 0.26$, MACCity $R^2 = 0.00$, and RETROv2 $R^2 = 0.04$), leading to the selection of this emission inventory."**

1.14 Why not add a supplemental figure showing the average spatial distribution of CO and R_btex emissions so readers can get a sense for which is driving the SIMPLE predictions in the various countries? I recommend at least plotting this as country average, if not both country averages and grid cells.

Please see response to comment 1.4.

1.15 Consider adding to the conclusions the ASOA-associated premature death estimates you are most confident in.

We respectfully disagree on this point, as prior studies have discussed $PM_{2.5}$-associated premature death estimates with less investigation into how well the models predicted the composition of the total $PM_{2.5}$ while still attributing premature deaths to different sources (e.g., Lelieveld et al., 2015; Silva et al., 2016; Ridley et al., 2018).

Minor comments:

1.16 Line 60: Rewrite "anthropogenic reactivity of specific organic compounds" to "reactivity of specific anthropogenic volatile organic compounds"?

Completed.

1.17 Line 66: "results in up to . . ."

Completed.

1.18 Line 67: "extrapolation" of what data specifically? Is it more informative to say "extrapolation from regions where detailed emission inventory data are available to other regions where uncertainties in emissions are larger."?

We have updated the line to say:

**"A limitation of this study is the extrapolation from cities with detailed studies and regions where detailed emission inventories are available to other regions where uncertainties in emissions are larger."**

1.19 Line 68: I agree that comprehensive air quality campaigns are certainly helpful and possibly necessary, but it seems that robust national-scale institutions (government, academic, or private) are absolutely necessary to accurately catalogue emission factors and activity data to the level required to reduce the uncertainties discussed in this manuscript. Perhaps this sentence could be broadened to something like: "In addition to further development of institutional air quality management infrastructure, comprehensive air quality campaigns . . ."

We have updated the line to say:

**"In addition to further development of institutional air quality management infrastructure, . . ."**

1.20 Lines 104 - 106:  Suggest rewriting: "Biogenic SOA (BSOA) in urban areas typically results from advection of regional background concentrations rather than processing of locally emitted biogenic VOCs."

Updated to this.

1.21 Lines 116 - 118:  Seltzer et al. (2021) is currently finalizing discussion in ACPD and presents a detailed VCP emission inventory for the U.S. Based on this the authors may want to update this sentence to include that step forward, but it's their choice.

We have added the following to address this:

**"Since numerous studies have shown the importance of VCPs and other non-typical VOC emission sources, efforts have been made to try to improve the representation and emissions of VCPs (Seltzer et al., 2021), which can reduce the uncertainty in ASOA precursors and the associated premature deaths estimation."**

1.22 Line 119:  "uncertainty on the (burden of -or- emissions of) ASOA precursors. . ."

Updated the text to say:

**". . . uncertainty on the emissions of ASOA precursors . . ."**

1.23 Eq. 3:  Recommend adding an exp subscript to [OH] here to make it clear that it is calculated from Eq. 2.

Updated.

1.24 Table S10. Please indicate which slope is being shown here (delta_SOA/R_BTEX)

We have removed this Table due to response 1.12 and have replaced it with the table shown there.

1.25 Line 201: Many of the BTEX values are modeled with equation 3 right? Please make this clear.

For the IVOCs used in this analysis, only 1 city had BTEX was calculated with Eq. 3 (London) while the rest of the BTEX are used from studies (NE USA, LA, Beijing, and NYC).

We have added the following to clarify:

**"The IVOC:BTEX emission ratio from inventories are multiplied with the observed BTEX, either the reported value from studies (NE US aircraft (Warneke et al., 2007), Los Angeles (de Gouw et al., 2017), Beijing (Wang et al., 2014), and New York City (Warneke et al., 2007)) or estimated from Eq. 3 (London), . . ."**

1.26 Line 213: C* range is not consistent with how IVOCs are usually defined.

We have updated the values to $10^3 < C^* < 10^6$ μg m$^{-3}$.

1.27 Line 209 - 210: Based on the reference, it appears the authors are specifically referring to underestimation of IVOCs in the ambient. Please make that more clear in the sentence.

We have rephrased the sentences to clarify this point:

**"Additionally, we rely on inventories for estimating atmospheric abundances of IVOCs because it has been challenging to measure the full range of IVOC precursors that are emitted into urban air due to many of the IVOCs from VCPs being oxygenated VOCs. These compounds are challenging to measure using traditional instrumentation (e.g., gas chromatography-mass spectrometry), leading to potential underestimation of the IVOC emission ratios (Zhao et al., 2014, 2017; Lu et al., 2018)."**

1.28 Line 214 - 216: It's unclear to me how the IVOCs and unspeciated SOA precursors relate to each other here. Are the authors saying they used SOA yields from Jather et al. (2014) to define the IVOC SOA yields uniformly for all C* bins? Please clarify. For example, a clearer way of making that point might be, "SOA yields from IVOC oxidation were parameterized with data from n-tridecane for gasoline engines and n-pentadecane for diesel engines (Jathar et al., 2014)."

We have updated to the text to say:

**"The ASOA yields and rate constants for IVOC oxidation were parameterized with data from n-tridecane and n-pentadecane for gasoline and diesel emissions, respectively (Jathar**

**et al., 2014), and for VCPs, the yields and rate constants for IVOC oxidation were parameterized with data from n-tetradecane (McDonald et al., 2018)."**

1.29 Lines 216 - 218: Should VOCs be IVOCs here? Again, aren't all the IVOCs in this study unspeciated? If so, why make the distinction?

We have added the I before the VOCs here. Also, we made the distinction for unspeciated specifically for VCPs as the IVOCs are unspeciated; however, BTEX, which can be in VCPs (Fig. 2), is speciated.

1.30 Line 224: Why was the Huffman et al. (2009) distribution not used for the cooking VBS distribution?

To be consistent with Ma et al. (2017), we used the same profiles as those authors used in their analysis.

We have added the following to clarify:

**"These profiles were selected to be consistent with Ma et al. (2017)."**

1.31 Table S9: What is the HOA and Other POA mass normalized to? Shouldn't these also be normalized to CO, or is POA a separate variable in the inventories? POA is never mentioned in the SI in the discussion of the inventory development.

We have updated the table to include the CO term; thus, the units are $\mu g \ sm^{-3} \ ppmv^{-1}$. As described in line 217 - 218, the SVOC emission ratios were estimated relative to the POA mass concentrations.

1.32 SI Line 70: The emission ratios are small, or the range is small?

The range is small and has been updated.

1.33 Lines 250 - 273: Is the TSI parameterization with the Ma parameterization or are they different cases that are explored? It seems like Ma et al. (2017) is used for IVOC SOA yields instead of Jather et al. (2014). There are a lot of parameterizations, precursors classes and products in this model approach. I strongly recommend adding a table(s) explicitly specifying all of the SOA yields and the corresponding precursors used in this study.

No, the TSI parameterization was not used, but the "WOR + ROB + MA" case from Ma et al. (2017). We have added the following table, from Ma et al. (2017), as the compounds used and their rate constants were already included:

**Table 13. Parameters for VOC, IVOC, and SVOC aerosol yields. The yields are taken from Ma et al. (2017).**

| Compound | Stoichiometric SOA yield High-NOx, 298 K ($\mu g\ m^{-3}$) | | | | |
|---|---|---|---|---|---|
| | 0.1 | 1 | 10 | 100 | 1000 |
| Benzene | N/A | 0.276 | 0.002 | 0.431 | 0.202 |
| Toluene | | | | | |
| Ethyltoluene | | | | | |
| Propylbenzenes | | | | | |
| Xylenes | N/A | 0.310 | 0.000 | 0.420 | 0.209 |
| Trimethylbenzenes | | | | | |
| IVOC C* = 6 | 0.007 | 0.090 | 0.206 | 0.350 | 0.00 |
| IVOC C* = 5 | 0.0498 | 0.0814 | 0.456 | 0.278 | 0.00 |
| IVOC C* = 4 | 0.053 | 0.103 | 0.464 | 0.266 | 0.00 |
| IVOC C* = 3 | 0.064 | 0.0914 | 0.562 | 0.209 | 0.00 |
| HOA C* = 2 | N/A | N/A | 0.28 | N/A | N/A |
| HOA C* = 1 | N/A | 0.18 | N/A | N/A | N/A |
| HOA C* = 0 | 0.12 | N/A | N/A | N/A | N/A |
| COA C* = 2 | N/A | N/A | 0.1881 | N/A | N/A |
| COA C* = 1 | N/A | 0.1188 | N/A | N/A | N/A |
| COA C* = 0 | 0.0594 | N/A | N/A | N/A | N/A |

We have also updated the rate constant table to include the rate constants for IVOCs and SVOCs.

1.34 Line 273: Recommend rephrasing "increase in mass of 0.99" to "change in mass of 0.99" or "decrease in mass of 1%".

We have updated the text to say:

**". . . a decrease in mass of 1%. . ."**

1.35 Line 276 - 281: This opening sentence is overly dense and meandering. What is the point of the appositive, "for ASOA apportionment (Fig. 1)"? It seems redundant. Should the second "apportionment" be "attribution"? The last portion of the sentence, after the GEOS-Chem reference should be broken off into its own sentence.

We have changed this text to say:

**"The model used in this study is GEOS-Chem v12.0.0 (Bey et al., 2001; The International GEOS-Chem User Community, 2018). This model is used for the following calculations: (1) ASOA apportionment (Fig. 1), (2) apportionment of ASOA to total PM2.5 for premature mortality calculations (Sect. 5), and (3) sensitivity analysis for ASOA production and emissions on premature mortality calculations. GEOS-Chem is operated at 2°×2.5° horizontal resolution."**

1.36 Line 335: Recommend presenting Eq. 4 as the summation of premature deaths among all considered causes.

We disagree, as this is how the equation is typically presented in epidemiology papers (e.g., Burnett et al., 2018), and we stated in line 333 that the equation varies according to both the particular disease category and geographic region. The combination of these two dependencies would make writing the summation harder to understand.

*Reviewer #2*

2.0 The study represents an attempt to estimate the premature mortality linked to Anthropogenic Secondary Organic Aerosols. Using 11 urban areas on three continents and specific volatile organic compounds emission ratios were estimated and a budget for ASOA is attempted. With the studied dataset the SIMPLE parameterization for ASOA in the GEOS-Chem model is updated to reproduce observed ASOA. Finally an attribution of ASOA PM2.5 premature deaths is attempted.

General comment:

2.1 My greatest concern for the specific study is the overall omission of solid fuel combustion in all calculations, both for ASOA production (emissions and subsequent processing/oxidation/ageing) as well as its contribution to premature mortality. Not only biomass burning for heating purposes but also forest forest, burning of crops etc. This leads to unaccounted emissions from urban areas such as Europe/US during winter from household heating but also from forested areas such as the Amazon, Canada, Siberia, Southeast Asia.

The purpose of this paper was to investigate the role of photochemically produced anthropogenic SOA. We provide further discussion and clarification of this point in response to 1.3.

We have also added the following line in the introduction to clarify this point:

**"Note, for the rest of the paper, unless explicitly stated otherwise, ASOA refers to SOA produced from the photooxidation of AVOCs, as there are potentially other relevant paths for the production of SOA in urban environments (e.g., Petit et al., 2014; Kodros et al., 2018, 2020; Stavroulas et al., 2019)."**

Specific Comments:

2.2 Line 110 - 114:  Isn't solid fuel combustion/biomass burning aged SOA considered as ASOA? According to Kodros et al. (2020) in active fire regions bbOOA increases by more than 50-60% from fast oxidation processes even in the dark. Significant contribution of primary BBOA oxidation to the oxygenated OA have also been identified in large urban centers such as Paris (2014) and Athens (2019).

We have added the following:

**". . . and solid fuel emissions from residential wood burning and/or cookstoves (e.g., Hu et al., 2013, 2020; Schroder et al., 2018), . . ."**

Further, as discussed in R1.3, we further emphasize two studies that did have important impacts from solid fuel emissions. The results for these studies fall within the trend of the photochemically-produced ASOA.

However, we do not have the ability to potentially constrain or include "dark-aging" of bbOA into bbOOA. Thus, as we noted in R1.3, we suggest that the ASOA concentrations in this study may be an underestimate for this reason.

2.3 Line 119 - 132:  Isn't the current study also under-predicting ASO by ignoring bbOOA? Furthermore, there is also the additive effect of the different pollutants when considering premature mortality. For example, Kodros et al. (2018) estimate joint exposure from household solid fuel use and ambient PM2.5 pollution and find 18% more deaths than by separating household and ambient mortality calculations. Which shows that solid fuel combustions is important for mortality as well, not only for ASOA calculations.

The Kodros et al. (2018) study investigated indoor and outdoor exposure; whereas, the purpose of this study is to only investigate the role of exposure to outdoor $PM_{2.5}$. We agree that this source of indoor pollution (among others, e.g., HOMEChem and other references) could be important additional sources of $PM_{2.5}$ exposures, and thus contributors to premature mortality, Kodros et al.

(2018)

acknowledged a large source of uncertainty associated with the indoor estimation of solid-fuel use and thus associated premature mortality.

To clarify this point, we have added the following:

**"Though there are potentially other important exposure pathways to PM that may increase premature mortality, such as exposure to solid-fuel emissions indoors (e.g., Kodros et al., 2018), the focus of this paper is on exposure to outdoor ASOA and its associated impacts to premature mortality."**

2.4 Fig.5a and Line 174-180, Fig. 6 and line 423-428: Authors only mention the uncertainties in x- and y-axis values. Does really by removing just one point increases the slope that much? The y-axis has an upper value of 140 compared to x-axis of 6! Why only 25% of the observed ASOA be associated with BTEX? What about the rest? Isn't this a solid proof that solid fuel combustion (BBOA) should definitely be taken into account?

See response to comment 1.12 concerning a more robust statistical analysis to determine if any one point could be driving the slope or not.

We have updated the text (see R1.3) to reflect that we have two studies (Chinese Outflow 2011 and New York City 2015) that had major impacts of solid fuel emissions (coal combustion for Chinese Outflow and biomass burning from New York City). As discussed in response to 1.12, these two points are not outliers and do not individually overly influence the slope; thus, the update we propose here for the SIMPLE model that is used in GEOS-Chem appears to capture and not underestimate the *photochemical* ASOA production from those sources as well. See response to 1.3 for updated text and clarification on this point.

In regards to the 25% of observed ASOA being explained by BTEX, as stated on pg 18, lines 410 - 413 in the original manuscript:

"However, BTEX alone cannot account for much of the ASOA formation (see budget closure discussion below), and instead, BTEX may be better thought of as both partial contributors and also as indicators for the co-emissions of other (unmeasured) organic precursors that are also efficient at forming ASOA."

BTEX only explaining 25±6% of the observed ASOA is not shocking, as has been highlighted in prior studies (e.g., Dzepina et al., 2009; Ensberg et al., 2014; Hayes et al., 2015; Ma et al., 2017; Nault et al., 2018), which led to the McDonald et al. (2018) study and the importance for the finding of VCPs potentially explaining a large fraction of the missing *photochemically* produced ASOA. To further clarify this point, we have added the following:

**"BTEX only explaining 25% of the observed ASOA is similar to prior studies that have done budget analysis of precursor gases and observed SOA (e.g., Dzepina et al., 2009; Ensberg et al., 2014; Hayes et al., 2015; Ma et al., 2017; Nault et al., 2018)."**

As explained throughout Section 3.2 and with Fig. 6, the remaining budget of the observed *photochemically* produced ASOA is explained by other aromatic compounds, IVOCs, and SVOCs. This is the important finding, as we have expanded the work from McDonald et al. (2018) to show the importance of IVOCs and VCPs in the production of *photochemically* produced ASOA. Specifically, we find (pg 21, line 464 of original submission) 85±12% of the observed ASOA for the five different cities to be explained by BTEX, aromatic compounds, IVOCs, and SVOCs.

2.5 Section 2.5.2 Once more, by not including solid fuel combustion in ASOA all the respective chemistry and oxidation is missing, losing 50-60% of SOA from fast oxidation of BBOA, even in the dark (NO3 radicals) (Kodros et al., 2020).

As explained in response to comments 1.3 and 2.4, solid fuel combustion is included in both the experimental studies and the model. We do not have a way to include the dark oxidation of BBOA, and we have acknowledged that this may lead to an underestimate of the concentrations and thus the health effects of ASOA (see response to R1.3).

2.6 Line 578 - 579: How is the "model constrained to atmospheric observations for a more accurate contribution of SOA" when an important source of ASOA such as solid fuel combustion is omitted?

As summarized in response comments 1.3, 2.4, and 2.5, this is a misunderstanding. Solid fuel combustion is included. We have updated this text to say:

**"Using a model constrained to day-time atmospheric observations (Fig. 2 and Fig. 4, see Sect. 4) leads to more accurate estimation of the contribution of photochemically-produced ASOA to PM$_{2.5}$ associated premature mortality that has not been possible in prior studies."**

Technical corrections:

2.7 Fig. 7 & 8: USOA? Should it be ASOA?

Updated:

[Figure]

**Figure 7. Flowchart describing how observed ASOA production was used to calculate ASOA in GEOS-Chem, and how the satellite-based PM$_{2.5}$ estimates and GEOS-Chem PM$_{2.5}$ speciation was used to estimate the premature mortality and attribution of premature mortality by ASOA. See Sect. 2 for further information about the details in the figure. SIMPLE is described in Eq. 9 and by Hodzic and Jimenez (2011) and Hayes et al. (2015). The one of two methods mentioned include either the Integrated Exposure-Response (IER) (Burnett et al., 2014) with Global Burden of Disease (GBD) dataset (IHME, 2016) or the new Global Exposure Mortality Model (GEMM) (Burnett et al., 2018) methods. For both IER and GEMM, the marginal method (Silva et al., 2016) or attributable fraction method (Anenberg et al., 2019) are used.**

[Figure]

**Figure 8. Five-year average (a) estimated reduction in PM$_{2.5}$-associated premature deaths, by country, upon removing ASOA from total PM$_{2.5}$, and (b) fractional reduction (reduction PM$_{2.5}$ premature deaths / total PM$_{2.5}$ premature deaths) in PM$_{2.5}$-associated premature deaths, by country, upon removing ASOA from GEOS-Chem. The IER methods are used here. See Fig. S9 and Fig. S12 for results using GEMM. See Fig. S10 for 10×10 km$^2$ area results in comparison with country-level results.**

We also noticed an error in the labels in Fig. 6 and have updated and include the update below:

[Figure]

**Figure 4. (a) Budget analysis for the contribution of the observed $\Delta$SOA/$R_{BTEX}$ (Fig. 25) for cities with known emissions inventories for different volatility classes (see SI and Fig. 2 and Fig. S6). (b) Same as (a), but for sources of emissions. For (a) and (b), SVOC is the contribution from both vehicle and other (cooking, etc.) sources. See Sect. 2 and SI for information about the emissions, ASOA precursor contribution, error analysis, and discussion about sensitivity of emission inventory IVOC/BTEX ratios for different cities and years in the US.**

Similar to IVOCs, the ability to measure the full range of SVOCs emitted into urban air is challenging. Therefore, we estimate SVOC emission ratios relative to POA mass concentrations (Table S9), as described by Ma et al. (2017). For the hydrocarbon-like portion, we used the volatility distribution from Worton et al. (2014) to estimate SVOC, as this is associated with fossil fuel emissions from transportation (Zhang et al., 2005). For the other POA, we used the volatility distribution from Robinson et al. (2007), as this POA is typically cooking primary aerosol. These profiles were selected to be consistent with Ma et al. (2017).

To estimate the SVOC mass concentration in equilibrium with the POA (Table S9) in each bin, the POA mass concentration is first multiplied by the fraction of POA measured in each bin from literature. This yields the concentration of POA for that specific volatility bin.

Then the total POA + SVOC concentration for that bin is obtained divided by the amount of material found in the particle phase for that bin for the average temperature (~298 K) and OA

mass concentration (~10 µg m$^{-3}$). Then, the gas-phase SVOC concentration is calculated by multiplying the total concentration by the gas-phase fraction. Thus, e.g., SVOC in the $C^* = 100$ µg m$^{-3}$ bin, ~91% of the SVOC mass will be found in the gas-phase.

Fig. S6 shows the calculated emission ratio versus saturation concentration ($c^*$) for the cities with emission inventories. The saturation concentration for SVOC was determined as part of the estimation procedure discussed above. For IVOC, the emission ratios for the different sources (gasoline, diesel, other fossil fuel sources, and VCP emissions) were split into the volatility bins, as in McDonald et al. (2018). Finally, for BTEX and non-BTEX aromatics, and other VOC emission ratios (see Fig. S6 for references for the other VOC emission ratios), CRC (Rumble, 2019) or SIMPOL.1 (Pankow and Asher, 2008) (for estimating vapor pressures not in CRC) was used to estimate the saturation concentrations.

**.1 US Emission Inventories**

*Anthropogenic VOC emissions*

The US emissions of VOCs is based on a mass balance estimate of the petrochemical industry reported by McDonald et al. (2018). Briefly, fuel sales and chemical product use are estimated from publicly available reports on energy use, chemical production, economic surveys, and freight shipments. Mobile source emission factors are from prior work quantifying both on-road and off-road engines (McDonald et al., 2013, 2015). Evaporative sources of transportation fuels are considered in addition to tailpipe exhaust (Pierson et al., 1999). VCP emission factors are based on literature values, including from the indoor environment, and reported in McDonald et al. (2018). Other fossil energy sources of VOCs, such as from oil refineries and industry, are taken from official inventories reported by the California Air

Resources Board (CARB, 2013) or US Environmental Protection Agency (NEI, 2015).

McDonald et al. (2018) reported fossil-VOC emissions for the Los Angeles basin in the year

2010.

*Speciation of VOC emissions*

The total VOC emissions are speciated to estimate BTEX and IVOC emissions from petrochemical VOC sources. Briefly, gasoline and diesel exhaust, gasoline fuel, and headspace vapors are based on profiles reported in the literature from the Caldecott Tunnel (Gentner et al.,

2012, 2013). Speciation profiles of VCPs are based on California Air Resources Board surveys of architectural coatings (Davis, 2007) and consumer products (CCPR, 2015). Other industrial solvent uses and point/area source emissions are from the EPA SPECIATE (v4.4) database (EPA,

2014).

*Extrapolating IVOC/BTEX ratios from 2010 Los Angeles to other field campaigns*

In the ASOA mass closure estimation, three separate field campaigns are utilized from the US: NEAQS 2002 (Boston/New York City), CalNex 2010 (Los Angeles), and WINTER

2015 (New York City outflow). These field campaigns span two megacities (Los Angeles and

New York City), ~one decade, and two seasons (summer versus winter). Here, we discuss how each of these variables could affect the IVOC/BTEX emissions ratio. We focus the discussion on mobile sources and VCPs because these are the dominant contributors to BTEX and IVOCs.

The IVOC/BTEX emissions ratio could be affected by the population density of a city. It is well-established that per capita transportation fuel use decreases with increasing population density (Gately et al., 2015), whereas VCP usage is expected to scale with population. Relative to Los Angeles, the per capita fuel use in New York City is ~2 times lower (Gately et al., 2015), resulting in lower on-road transportation VOC emissions relative to VCPs. Because aromatics are mainly found in gasoline, whereas the IVOCs have a higher contribution from VCPs, the

IVOC/BTEX ratio is expected to be higher in New York City than Los Angeles.

To assess impacts of annual trends on the IVOC/BTEX ratio, we utilize long-term trend analyses of mobile source VOC emissions in Los Angeles (McDonald et al., 2013, 2015; Hassler et al., 2016). The main effect is that on-road gasoline emissions have decreased with time, both from the tailpipe of vehicles (McDonald et al., 2013) and of gasoline-related VOCs in ambient air measurements (Warneke et al., 2012). We utilize the EPA Trends Report to scale VOC

emissions for other anthropogenic sectors, including VCPs and industrial sources (https://www.epa.gov/air-emissions-inventories/air-pollutant-emissions-trends-data). The EPA

Trends Report suggests that VCP (or solvent) emissions decreased by ~30% between 2002 and

2010, including efforts to reduce the VOC content of architectural coatings (Matheson, 2002).

After 2010, the emissions have been slightly increasing, likely due to population growth.

Because both mobile sources and VCP emissions are decreasing with time, the IVOC/BTEX

emissions ratio is not significantly altered.

Lastly, the effects of seasonality influence on-road transportation emissions through: (i)

increased VOC emissions in winter relative to summer from cold-starting engines, and (ii) lower evaporative emissions due to colder ambient temperatures. We estimate that exhaust emissions from passenger vehicles increases by ~50% due to higher cold-start emissions in winter relative to summer based on the EPA MOVES model (MOVES, 2015). Evaporated gasoline and headspace vapors are known to exhibit a temperature-dependence (Rubin et al., 2006), and estimated to be ~20% and ~80% lower, respectively, based on typical wintertime temperatures of New York City relative to summertime Los Angeles. Due to compensating factors between cold-start engines and evaporated fuels, the IVOC/BTEX emissions are not significantly affected by seasonality.

Overall, when taking into account differences in population density between Los Angeles and New York City, trends of mobile source and VCP emissions over time, and seasonality, the IVOC/BTEX emission ratios range between ~2.3 to 2.7, which is a relatively small range. This sensitivity analysis helps explain why the enhancement observed in SOA scales with BTEX levels in the urban atmosphere.

**S1.2 Beijing Emission Inventory**

*Anthropogenic VOC emissions*

The total VOC emissions of Beijing were developed following the bottom-up framework of the Multi-resolution Emission Inventory for China (MEIC) model (available at http://www.meicmodel.org), based on a technology-based methodology. The details of activity rates, emission factors, technology distribution, and control measures configured in the MEIC model are summarized in a series of papers (Zhang et al., 2009; Zheng et al., 2014, 2018; Liu et al., 2015; Li et al., 2017, 2019).

In the MEIC model, a detailed four-level source classification system, representing sector, fuel/product, technology/solvent type, and end-of-pipe pollutant abatement facilities, was established by including over 700 emitting sources for each province. All anthropogenic sources, including power plants, industrial sources, volatile chemical products, fossil fuel burning in residential stoves, transportation were all considered.

Power plants are treated as point sources in the MEIC model. The VOC emissions were derived from the China coal-fired Power Plant Emissions Database (CPED, (Liu et al., 2015)), which is developed based on information of each unit on fuel type, fuel quality, combustion technology, etc.

Volatile chemical products are comprised of solvent use applied for architecture, vehicles, wood, and other industrial purposes, glue use, printing, pesticide use, and domestic solvent use.

The market share of waterborne and solvent-based paint is further taken into account for each source category. For the on-road transportation sector, the improved emissions developed by

Zheng et al. (2014) were integrated into the framework of MEIC, which estimated the vehicle population and emission factors at a county level. Both the VOC emissions in running mode and evaporation were considered. Emission standards covering pre-Euro I and Euro I to Euro V in

Beijing were applied for each vehicle type (Zheng et al., 2018; Li et al., 2019). Regarding oxygenated volatile organic compounds (OVOCs), the emission factors for on-road vehicles were corrected, as current emission factors are only for non-methane hydrocarbons (NMHC).

Correction ratios of 1.32, 1.08, 1.10, and 1.06 were applied for heavy-duty and light-duty diesel vehicles, and heavy-duty and light-duty gasoline vehicles, respectively, to the original values to comply with the follow-up speciation for the total VOC, following the method of Li et al. (2014,

2019).

*Speciation of VOC emissions*

Emissions by individual chemical species were developed based on the profile-assignment approach (Li et al., 2014, 2019). First, a "composite" profile database for

China was established by integrating the local profiles and supplementing it with the SPECIATE

v4.5 database for absent sources ((Simon et al., 2010), available at:

https://www.epa.gov/air-emissions-modeling/speciate-version-45-through-40). The detailed procedure for developing the composite profile database is illustrated in Li et al. (2014). In brief, for sources where there are significant differences in technology or legislation between China and western countries, only local profiles are used; otherwise, all candidate profiles are included for further compilation in the composite profile database. Local profiles covering most of the important sources were gathered and reviewed, including biofuel combustion, coal combustion, asphalt production, oil production, refinery, paint use, gasoline evaporation, gasoline vehicle exhaust, diesel vehicles, and so on, as detailed illustrated in Li et al. (2019).

Then, profiles for all combustion-related sources, including fossil fuel combustion in power plants, industry, residential, and transportation sectors were reviewed, and incomplete profiles that were absent from the OVOC fractions were corrected by appending the component of "OVOC" with fractions derived from the "complete" profiles for the same source. After

OVOC correction, all "candidate" profiles were averaged by species to establish the composite profile database. Finally, the composite profile to each source was assigned by setting up the source linkage between the profile database and the inventory. Emissions by individual chemical species for each source were then further developed.

**1.3 London/United Kingdom Emission Inventory**

*Anthropogenic VOC emissions*

The National Atmospheric Emissions Inventory (NAEI) estimates UK emissions of VOCs from anthropogenic sources following methods in the EMEP/EEA Emissions Inventory Guidebook (EMEP/EEA, 2016) for submission under the revised EU Directive 2016/2284/EU on National Emissions Ceilings (NECD), available at: https://eur-lex.europa.eu/legal-content/EN/TXT/PDF/?uri=CELEX:32016L2284&from=EN, and the United Nations Economic Commission for Europe (UNECE) Convention on Long-Range Transboundary Air Pollution (CLRTAP), available at: http://www.ceip.at/ms/ceip_home1/ceip_home/reporting_instructions/reporting_programme/. The NECD and CLRTAP define those VOC sources to be included and excluded from the national inventory (for example, emissions of NMVOCs from biogenic sources are not included). The Guidebook provides estimation methodologies and default emission factors for each source category, although countries can use country-specific emission factors where these are deemed relevant. The NAEI currently covers organic emissions from around 400 individual source categories, with a large contribution from a diverse range of industrial processes and solvents, but with very few individually dominant sources.  The inventory then speciates emissions into ~650 individual compounds, or groups of compounds. Groupings of organics, for example, expressed as 'sum of all C14 compounds,' make up a substantial fraction of IVOC emissions, rather than being reported as individual compounds.

Emissions from the use of solvents and other volatile chemicals in industry and in consumer products, fuel production and distribution, food and drink manufacture and other non-combustion industrial processes accounted for 72% of all UK NMVOC emissions in 2017, according to the NAEI.  Both the solvent and industrial process sectors cover a diverse range of emission source categories:  the NAEI identifies 136 separate categories across the two sectors

  For the road transport sector, the NAEI reports exhaust emissions of NMVOCs and its emissions from evaporative losses of fuel vapor from petrol vehicles.  Emissions from re-fueling at filling stations are reported separately under the fugitive emissions from the fuel distribution sector. The method used for road transport in the NAEI follows the method in the European

COPERT 5 model and described in the EMEP/EEA Emissions Inventory Guidebook.  The method uses average speed-related emission factors for hot exhaust emissions of total hydrocarbons for detailed vehicle categories (vehicle type, weight and/or engine size) and Euro standards for petrol cars, diesel cars, petrol and diesel light goods vehicles, rigid and articulated

HGVs, buses and coaches, and mopeds and motorcycles, and combines these with detailed traffic and fleet activity data derived from information provided by DfT. Separate estimates are made of methane emissions for each vehicle type and subtracted from the THC emissions to derive the

NMVOC emissions.

  Evaporative emissions from vehicles are estimated in the NAEI, using the Guidebook method for three different processes: diurnal losses, hot soak, and running losses. Emissions are dependent on ambient temperature and fuel vapor pressure and different factors are provided for vehicles with and without carbon canisters for evaporative emission controls. All vehicles from

Euro 1 onwards are fitted with these devices; so, evaporative emission have been decreasing from the early 1990s with the penetration of these vehicles in the fleet. The method also takes into account the reduction in Reid Vapour Petrol of petrol sold in the UK since 2000, as required for compliance with the EU Fuel Quality Directive 98/70/EC, amended by Directive

2009/30/EC.

*Speciation of VOC emissions*

The NAEI is considered to adequately reflect annual real world emissions of BTEX (see, for example, eddy covariance flux comparisons in London by Langford et al. (2010) and Vaughn et al. (2017)); so, those values are taken directly from the NAEI and used here. IVOCs, and particularly long chain hydrocarbons, are included in many cases in the inventory as groups, but their emissions are known to be significantly underestimated when compared against field observations. We use the observations of Dunmore et al. (2015), made in wintertime central

London in 2012, as guide to uprate NAEI emissions for IVOC species based on the estimated discrepancies between inventory and field observation reported for each carbon number above

C10. This leads to some significant multipliers being applied to the inventory values, sometimes of the order 60 to 70. We assume that the same multipliers apply to all sources, since field data does not provide any means to attribute different factors to road transport IVOCs compared with

IVOCs from VCP sources.

Since the NAEI represents a reporting of emissions for the purposes of compliance with international treaties, some fraction of those emissions are not released on the mainland UK. For this paper, offshore BTEX and IVOC emissions, arising for example from offshore oil and gas activity, aircraft in cruise, or shipping and emissions associated with overseas Crown

Dependencies are removed from the UK total, since they play no part in determining the chemical environment of London. The annual NAEI totals are then divided equally to give a daily national emission.

**S2 ASOA Budget Analysis of Ambient Observations**

To calculate the ASOA budget, we used the observed BTEX (Table S5) and non-BTEX

aromatic (Table S8) emission ratios, the emission inventories for IVOC (see above), and estimated SVOCs from the primary OA emissions (see above). The methods to calculate ASOA

from emissions have been described in detail elsewhere (Hayes et al., 2015; Ma et al., 2017;

Schroder et al., 2018), and are briefly described here. All calculations described were conducted with the KinSim v4.02 chemical kinetics simulator (Peng and Jimenez, 2019) within Igor Pro 7

(Lake Oswego, Oregon), and are summarized in Fig. S7. A typical average particle diameter for urban environments of ~200 nm (Seinfeld and Pandis, 2006) is used to estimate the condensational sink term for the partitioning of gas-to-particle, although condensation is always fast compared to the experiment timescales. Further, we assume an average 250 g mol$^{-1}$ molar mass for OA and an average SOA density of 1.4 g cm$^{-3}$ (Vaden et al., 2011; Kuwata et al., 2012).

Finally, all models are initialized with the campaign specific OA background (typically ~2 µg sm$^{-3}$) and POA (Table S9) for partitioning of gases to the particle phase, and ran at the average temperature for the campaign.

For the modeled VOCs (BTEX and non-BTEX aromatics), each species undergoes temperature-dependent OH oxidation (Table S12), forming four SVOCs that partition between gas- and particle-phase, using updated SOA yields that account for wall loss (Ma et al., 2017).

For IVOCs, the emission weighted SOA yields and rate constants from the "Zhao" option (Zhao et al., 2014) of Ma et al. (2017) are used, and the products are apportioned into three SVOC bins and one low-volatility organic compound (LVOC) bin (Fig. S7). Finally, SVOCs undergo photooxidation at a rate of $4\times10^{-11}$ cm$^3$ molecules$^{-1}$ s$^{-1}$ (Dzepina et al., 2009; Hodzic et al.,

2010b; Tsimpidi et al., 2010; Hodzic and Jimenez, 2011; Hayes et al., 2015; Ma et al., 2017;

Schroder et al., 2018), producing one product per oxidation step, with yields from Robinson et al.

(2007) for cooking and other SVOCs and yields from Worton et al. (2014) for fossil fuel related

SVOCs, as recommended by Ma et al. (2017). The products from SVOC and IVOC oxidation are allowed to further oxidize, as highlighted in Fig. S7 and described in prior studies (Hayes et al.,

2015; Ma et al., 2017; Schroder et al., 2018). Generally, each product reacts at a rate of $4\times10^{-11}$

cm$^3$ molecules$^{-1}$ s$^{-1}$ to produce some product at one volatility bin lower, adding one oxygen to the compound for each oxidation (Dzepina et al., 2009; Tsimpidi et al., 2010; Hodzic and Jimenez,

2011; Hayes et al., 2015; Ma et al., 2017; Schroder et al., 2018). An update includes fragmentation for a fraction of the molecules that are oxidized, as described in Schroder et al.

(2018) and Koo et al. (2014). As shown in Fig. S7, fragmentation of the compound occurs as it is oxidized and goes down one volatility bin. For further oxidation of SVOCs from the oxidation of primary IVOCs, one oxygen is added and 0.25 carbon is removed per step, leading to an increase in mass of 1.03 (instead of 1.07) per oxidation step (Koo et al., 2014; Schroder et al., 2018). For further oxidation of products from primary SVOC emissions, one oxygen is added and 0.5

carbon is removed per step, leading to a decrease in mass of 1% (instead of 1.07) per oxidation step (Koo et al. 2014; Schroder et al. 2018).

**S3 GEOS-Chem Modeling**

The model used in this study is GEOS-Chem v12.0.0 (Bey et al., 2001; The International

GEOS-Chem User Community, 2018). This model is used for the following calculations: (1)

ASOA apportionment (Fig. 1), (2) apportionment of ASOA to total PM2.5 for premature mortality calculations (Worldwide Premature Deaths Due to ASOA), and (3) sensitivity analysis for ASOA production and emissions on premature mortality calculations. GEOS-Chem is operated at 2°×2.5° horizontal resolution. Goddard Earth Observing System – Forward

[revised manuscript text omitted]

¶

**S4 Ozone Sensitivity to ASOA Simulations**

A potential issue in the attribution of premature mortality to AOSA is that reducing emissions that lead to ASOA is that this may impact ozone concentrations. A sensitivity analysis was conducted, where the ASOA emissions were reduced by 20% (Fig. S14). In general, there is a less than 1% reduction in total ozone concentration in the boundary layer. This is due to the fact that the most important AVOCs that contribute to ozone formation are light alkenes (e.g., ethylene and propylene, Fig. 2), which are not ASOA precursors. Though the reaction rate constant of the ASOA precursors is generally high (Table S12), the concentration of the precursors is low and they thus account for a low percentage of the total ozone production potential (Table S5 through Table S9). For example, the measured OH reactivity (Sect. 3) for two different urban regions was between 15 to 25 $s^{-1}$ (Griffith et al., 2016; Whalley et al., 2016)

while the OH reactivity for the ASOA precursors for the same region was between 2 to 4 $s^{-1}$. The small contribution to the OH reactivity is in line to the minimal impact to the ozone concentration observed in Fig. S14.

**S5 Error Analysis of Observations**

The errors that will be discussed here are in reference to  Fig. 2 and Fig. 4 and Table S4

either come from the 1σ uncertainty in the slopes (the SOA versus $O_x$, HCHO, or PAN values) or propagation of uncertainty in observations. For SOA, we estimate the 1σ uncertainty of ~15%, which is lower than the typical 1σ uncertainty of the AMS (Bahreini et al., 2009) due to the careful calibrations and excellent intercomparisons in the various campaigns (see Table 1 for references for the AMS comparisons). For ΔCO, the largest uncertainty is associated with the

CO background (Hayes et al., 2013; Nault et al., 2018), and is estimated to be ~10% at 0.5

photochemical equivalent days (Hayes et al., 2013). The uncertainty in the emission ratios is

~10% (Wang et al., 2014; de Gouw et al., 2017); though, it may be higher for the values calculated here (see above) due to the uncertainty in CO background, rate constants, and photochemical age. Therefore, for Fig. 2a, the uncertainty in the y-values is 18% and the uncertainty in the x-values is 10%. For Fig. 4, the uncertainty in the measurement is 21%.

Another potential source of uncertainty may stem from the fit of the data in Fig. 2a, as the data point from Seoul (KORUS-AQ) could be impacting the fit due to the difference in its value compared to the other locations. Statistical analysis for the influence of the data from Seoul on the figure was conducted, including a T-test, Cook's Distance test, and Difference in Fits test (Table S11). All three statistical tests show that the data from Seoul (and all the data in general)

is not overly influencing the reported slope.

A further potential source of uncertainty in this analysis is the calculated VOC emission ratios for the studies that did not have ratios published previously (Houston 2000, London,

Houston 2013, and Seoul). To investigate how well Eq. 3 does in estimating the VOC emission ratios, a comparison of the estimated VOC emission ratios versus previously published ratios for two different cities, Mexico City (Apel et al., 2010; Bon et al., 2011) and Los Angeles (de Gouw et al., 2017) was made (Table S10). Also, for Mexico City, two locations, an urban and a suburban site, were compared both against each other (Apel et al., 2010; Bon et al., 2011) and the calculated values from Eq. 3.

First, as shown in Table S10, even for the same location (suburban Mexico City), different values in the emission ratio, especially for the alkanes, can be observed, by as much as a factor of 7. This can be partially explained by differences in how the emission ratios were determined. For both Apel et al. (2010) and Bon et al. (2011), the authors took the slope of

VOCs versus CO and used different regression techniques and different time periods. Comparing their technique with ours, we generally estimate VOC emission ratios within 50% of the reported values, and the estimation improves for shorter lived compounds (e.g., aromatics). However, de

Gouw et al. (2017) more carefully took chemistry into consideration for any potential losses of the VOCs prior to observation to determine emission ratios, similar to this study. We believe the comparison with de Gouw et al. (2017) provides a more useful comparison in the method presented here. We find, at most, a 30% difference in the emission ratios, with an average difference of 4±15% for all compounds. Thus, from this analysis, we conclude that (1) there is large variability in VOC emission ratios across urban areas around the world, which has been highlighted in other studies (Warneke et al., 2007), and (2) the method that considers losses of

VOCs is the more accurate procedure to estimate VOC emissions and leads to the best reproducibility across studies and lowest uncertainty ($< 30\%$, ~4% on average).

**Supporting Information Tables**

**Table S1.** List of instruments whose observations are used in this study. In some cases $\Delta SOA/\Delta CO$ (Table S4), SOA versus $O_x$ slope (Table S4), or VOC emission ratios (Table S5 through Table S8) had already been reported, and, in those cases, we use the previous literature reports in our analyses.

| Location | SOA | $O_x$ | HCHO | PAN | VOCs | CO |
|---|---|---|---|---|---|---|
| Houston, TX, USA (2000) | Q-AMS[a] | CL & UV Absorpion[b] | DOAS[c] | GC-ECD[d] | GC-FID, GC-MS[e] | Infrared Absortion[f] |
| Mexico City, Mexico (2006) | HR-ToF-AMS[g] | CL[h] | TDLAS[i] | CIMS[j] | WAS[k] | UV RF[l] |
| Los Angeles, CA, USA (2010) | HR-ToF-AMS[g] | CL & UV Absorption[m] | Average of DOAS[c] & Hantzsch Reaction[n] | GC-ECD[d] | GC-MS[o] | UV RF[l] |
| Beijing, China (2011) | HR-ToF-AMS[g] | CL & UV Absorption[p] | PTR-MS[q] | GC-ECD[r] | GC-FID[s] | IR Absorption[p] |
| London, UK (2012) | C-ToF-AMS[t] | CL & UV Absorption[u] | Hantzsch Reaction[n] | GC-ECD[v] | GC-FID & GC×GC-FID[w] | UV RF[l] |
| Houston, TX, USA (2013) | HR-ToF-AMS[g] | CL[x] | Average of LIF[y] & CAMS[z] | CIMS[j] | WAS[k] | DACOM[aa] |
| Seoul, South Korea (2016) | HR-ToF-AMS[g] | CL[h] | CAMS[z] | CIMS[j] | WAS[k] | DACOM[aa] |

[a]Quadrupole Aerosol Mass Spectrometer (Q-AMS) (Jayne et al., 2000)
[b]Chemiluminescence (CL) and UV Absorption (Williams et al., 1997)
[c]Differential Optical Absorption Spectrometry (DOAS) (Stutz and Platt, 1996, 1997)
[d]Gas chromatography-electron capture detector (GC-ECD) (Williams et al., 2000; Roberts et al., 2002)
[e]Gas chromatography-flame ionization detector (GC-FID) and gas chromatography mass spectrometer (Roberts et al., 2001)
[f]TECO Model 48s IR gas-filter
[g]High Resolution Time-of-Flight Aerosol Mass Spectrometer (HR-ToF-AMS) (DeCarlo et al., 2006)
[h]Chemiluminescence (CL) and UV Absorption (Weinheimer et al., 1994)
[i]Tunable diode laser absorption spectroscopic (TDLAS) measurements (Fried et al., 2003)
[j]Chemical ionization mass spectrometer (CIMS) (Huey L Tanner D Slusher D Dibb J Arimoto R Chen G Davis D Buhr M Nowak J Mauldin R Eisele F, 2004; Slusher et al., 2004; Kim et al., 2007)
[k]Whole air sample, followed by analysis with GC-FID and/or GC-MS (Blake et al., 2003)
[l]UV Resonance Fluorescence (RF) (Gerbig et al., 1999)

[m]Chemiluminescence (CL) and UV Absorption (Hayes et al., 2013)
[n]Hantzsch reaction (Cárdenas et al., 2000)
[o]Gas chromatograph mass spectrometer (Gilman et al., 2010)
[p]Chemiluminescence (CL), UV Absorption, and IR Absorption (Hu et al., 2016)
[q]Proton transfer reaction mass spectrometer (PTR-MS) (Warneke et al., 2011)
[r]Gas chromatography electron capture detector (GC-ECD) (Zhang et al., 2017)
[s]Gas chromatography flame ionization detector (GC-FID) (Wang et al., 2014)
[t]Compact Time-of-Flight Aerosol Mass Spectrometer (C-ToF-AMS) (Drewnick et al., 2005)
[u]Chemiluminescence (CL) and UV Absorption (Whalley et al., 2016)
[v]Gas chromatography electron capture detector (GC-ECD) (Whalley et al., 2016)
[w]Gas chromatography flame ionization detector (GC-FID) (Dunmore et al., 2015)
[x]Chemiluminescence (CL) (Ryerson et al., 1999; Pollack et al., 2010)
[y]Laser induced fluorescence (LIF) (Cazorla et al., 2015)
[z]Compact Atmospheric Multi-species Spectrometer (CAMS) difference frequency absorption
spectrometer (Weibring et al., 2010)
[aa]Tunable diode laser absorption spectroscopy (Sachse et al., 1987)

**Table S2.** Concentrations of $PM_1$ components shown in Fig. 1. References for the measurements
can be found in Table 1.

| Dataset Location | Average Concentration ($\mu g\ sm^{-3}$) of submicron aerosol under standard temperature and pressure | | | | |
|---|---|---|---|---|---|
| | **SOA** | **HOA** | **SO$_4$** | **NO$_3$** | **NH$_4$** |
| Houston, TX, USA (2000) | 2.7 | 0.7 | 4.9 | 0.4 | 1.5 |
| Northeast USA (2002) | 4.9 | 0.5 | 2.0 | 0.3 | 0.7 |
| Tokyo, Japan (2004) | 6.0 | 1.5 | 4.4 | 0.9 | 4.0 |
| Mexico City, Mexico (2006) | 11.2 | 4.8 | 1.9 | 6.0 | 2.5 |
| Paris, France (2009) | 1.9 | 1.1 | 1.2 | 0.5 | 0.6 |
| Los Angeles, CA, USA (2010) | 5.0 | 2.0 | 2.9 | 3.6 | 2.1 |
| Changdao Island, China (2011) | 9.4 | 4.4 | 8.3 | 12.2 | 6.5 |
| Beijing, China (2011) | 17.1 | 8.9 | 22.0 | 16.8 | 13.7 |
| London, UK (2012) | 2.7 | 1.6 | 1.4 | 2.7 | 1.3 |
| Houston, TX, USA (2013) | 3.7 | 0.0 | 2.7 | 0.1 | 0.6 |
| New York City, NY, USA (2015) | 0.8 | 0.7 | 1.2 | 1.4 | 0.4 |
| Seoul, South Korea (2016) | 11.9 | 1.3 | 5.0 | 7.9 | 4.4 |

**Table S3.** Table summarizing the results of recent GEOS-Chem performance evaluations for modeling BSOA.

| Study | Observed Data | Species | Details |
|---|---|---|---|
| Fisher et al. (2016)[a] | SEAC[4]RS, below 1 km (spatial pattern), below 500 m (bias) | Isoprene | Spatial patterns well captured, and biases are +34% for isoprene and +3% for monoterpenes |
| | | Monoterpene | |
| | | Organic Nitrates from Isoprene | Spatial patterns well captured, and biases are -0.6% for first- and -35% for second-generation isoprene nitrates |
| | SEAC[4]RS, 0 - 4 km vertical profiles | Isoprene | Agreed well but GEOS-Chem somewhat overestimated observed concentrations near 1km |
| | | Monoterpene | |
| | | HCHO | |
| | | Organic Nitrates from Isoprene | Agreed within measurement uncertainties |
| | SOAS, at the surface | Isoprene | Underestimated isoprene and monoterpenes (-28% and -54%), but overestimated first- and second- generation isoprene nitrates (+85% and +43%) |
| | | Monoterpene | |
| | | HCHO | |
| | | Organic Nitrates from Isoprene | |
| Travis et al. (2016) | SEAC[4]RS, 0 - 12 km | First Generation from Isoprene Nitrates | Good agreement for ISOPOOH and ISOPN, underestimation of HPALDs by a factor of two |
| | | ISOPOOH | |
| | | HPALDS | |
| Marais et al. (2016) | SOAS, at the surface | IEPOX-SOA | Good agreement for isoprene derived aerosols, mean concentrations were almost the same |
| | | ISOPOOH-SOA | |
| | SEAC[4]RS, below 2 km (spatial pattern) | IEPOX-SOA | Spatial patterns well captured |

[a]This study decreased isoprene emissions by 15% and doubled monoterpene emissions of MEGANv2.1.

 **Table S3 cont.**

| Study | Observed Data | Species | Details |
|---|---|---|---|
| Kaiser et al. (2018)[a] | SEAC⁴RS | Isoprene | All were overestimated, except for first generation isoprene nitrates |
| | | HCHO | |
| | | ISOPOOH | |
| | | MVK + MACR | |
| | | First Generation Isoprene Nitrates | |
| Pai et al. (2020) | 15 airborne campaigns (SEAC⁴RS, GoAmazon, SENEX, OP3, etc.) | OA under biognic dominant conditions | Slight overestimation, but generally very similar in magnitude |

[a]NEI $NO_x$ emissions other than power plants decreased by 60%, soil $NO_x$ emissions were
reduced by 50% across the Midwestern US. With the decrease of $NO_x$ emissions, ISOPOOH
concentrations were increased in GEOS-Chem.

**Table S4**. Dilution-corrected SOA concentrations at 0.5 equivalent days and slopes of SOA versus $O_x$, HCHO, and PAN used in Fig. 2 and Fig. 3. References for the values can be found either in Table 1 or found in Fig. S2 through Fig. S4. Uncertainty is 1σ, and either represents propagation in uncertainty in measurements (see Sect. S5) for ΔSOA/ΔCO or uncertainty in slopes for SOA versus the three photochemical species.

| Dataset Location | ΔSOA/ΔCO at 0.5 eq. days | SOA vs. $O_x$ Slopes | SOA vs. HCHO Slopes | SOA vs. PAN Slopes |
|---|---|---|---|---|
| Houston, TX, USA (2000) | | 0.04±0.01[a] | 0.32±0.08 | 1.41±0.46 |
| Northeast USA (2002) | 16±3[b]
48±9[c] | | | |
| Mexico City, Mexico (2003) | | 0.14±0.01[a] | | |
| Tokyo, Japan (2004) | | 0.19±0.01[a] | | |
| Mexico City, Mexico (2006) | 58±10 | 0.16±0.01 | 1.60±0.06 | 5.60±0.30 |
| Paris, France (2009) | | 0.14±0.01[a] | | |
| Pasadena, CA, USA (2010) | 59±11 | 0.16±0.01 | 1.93±0.02 | 5.41±0.12 |
| Changdao Island, China (2011) | 23±4 | | | |
| Beijing, China (2011) | 31±6 | 0.21±0.01 | 3.90±0.15 | 7.42±0.46 |
| London, UK (2012) | 54±10 | 0.13±0.01 | 0.36±0.02 | 3.37±0.41 |
| Houston, TX, USA (2013) | | 0.16±0.01 | 1.52±0.13 | 6.92±0.58 |
| New York City, NY, USA (2015) | 33±6 | | | |
| Seoul, South Korea (2016) | 107±19 | 0.29±0.02 | 3.73±0.26 | 10.13±0.52 |

[a]Missing reported uncertainty; therefore, assuming ±0.01, as that is typical for other campaigns
[b]From de Gouw et al. (2005). [c]From Kleinman et al. (2007).

 **Table S5**. Emission ratios of BTEX aromatics used in this study. If no reference is listed, then
the emission ratio was calculated using Eq. 3.

| Dataset Location | Emission Ratios (ppbv aromatic/ppmv CO) | | | | | References |
| --- | --- | --- | --- | --- | --- | --- |
| | Benzene | Toluene | Ethylbenzene | m+p-xylene | o-xylene | |
| Houston, TX, USA (2000) | 2.6 | 3.5 | 0.6 | 2.8 | 0.8 | |
| NE USA, Ship (2002) | 0.9 | 2.0 | 0.2 | 0.6 | 0.3 | Baker et al. (2008) |
| NE USA, Aircraft (2002) | 0.8 | 2.9 | 0.4 | 1.2 | 0.5 | Warneke et al. (2007) |
| Mexico City, Mexico (2006) | 0.9 | 7.5 | 0.9 | 1.1 | 0.4 | Apel et al. (2010) |
| Los Angeles, CA, USA (2010) | 1.3 | 3.4 | 0.6 | 2.1 | 0.8 | de Gouw et al. (2017) |
| Changdao Island, China (2011) | 2.3 | 1.9 | 0.5 | 1.3 | 0.4 | Yuan et al. (2013) |
| Beijing, China (2011) | 1.2 | 2.4 | 1.0 | 1.6 | 0.6 | Wang et al. (2014) |
| London, UK (2012) | 1.8 | 6.3 | 1.2 | 2.2 | 1.1 | |
| Houston, TX, USA (2013) | 2.3 | 3.0 | 0.6 | 3.9 | 1.2 | |
| New York City, NY, USA (2015) | 0.8 | 2.9 | 0.4 | 1.2 | 0.5 | Warneke et al. (2007)[a] |
| Seoul, South Korea (2016) | 1.1 | 13.1 | 2.4 | 3.3 | 2.3 | |

[a]Using the emissions from Warneke et al. (2007) instead of Schroder et al. (2018) as Schroder et al. found significant uncertainty in the emissions calculated from observations.

**Table S6**. Emission ratios of alkanes used in this study. If no reference is listed, then the emission ratio was calculated using Eq. 3.

| Dataset Location | Emission Ratios (ppbv alkane/ppmv CO) | | | | | | | References |
|---|---|---|---|---|---|---|---|---|
| | Ethane | Propane | n-Butane | i-Butane | n-Pentane | i-Pentane | n-Hexane | |
| Houston, TX, USA (2000) | 40.9 | 24.3 | 9.0 | 14.7 | 3.1 | 10.0 | 3.1 | |
| NE USA, Ship (2002) | 8.3 | 2.3 | 1.8 | 1.3 | 1.0 | 2.8 | 0.9 | Baker et al. (2008) |
| NE USA, Aircraft (2002) | 9.9 | 9.0 | 2.4 | 1.3 | 2.0 | 5.4 | 0.6 | Warneke et al. (2007) |
| Mexico City, Mexico (2006) | 7.4 | 41.5 | 15.1 | 4.8 | 2.1 | 2.7 | 1.5 | Apel et al. (2010) |
| Los Angeles, CA, USA (2010) | 16.5 | 13.4 | 5.0 | 3.2 | 3.4 | 8.7 | 1.4 | de Gouw et al. (2017) |
| Changdao Island, China (2011) | 7.7 | 4.5 | 2.5 | 1.2 | 1.0 | 1.5 | 0.5 | Yuan et al. (2013) |
| Beijing, China (2011) | 4.3 | 3.9 | 2.5 | 2.5 | 1.2 | 2.0 | 0.6 | Wang et al. (2014) |
| London, UK (2012) | 33.0 | 17.8 | 17.3 | 8.4 | 4.6 | 11.3 | 1.3 | |
| Houston, TX, USA (2013) | 86.5 | 37.3 | 14.6 | 10.6 | 7.0 | 10.5 | 3.0 | |
| Seoul, South Korea (2016) | 16.1 | 0.4 | 6.0 | 3.4 | 3.1 | 3.7 | 1.7 | |

 **Table S7**. Emission ratios of alkenes used in this study. If no reference is listed, then the
 emission ratio was calculated using Eq. 3.

| Dataset Location | Emission Ratios (ppbv alkene/ppmv CO) | | References |
| --- | --- | --- | --- |
| | Ethene | Propene | |
| Houston, TX, USA (2000) | 24.4 | 28.4 | |
| NE USA, Ship (2002) | 4.4 | 1.1 | Baker et al. (2008) |
| NE USA, Aircraft (2002) | 4.9 | 1.4 | Warneke et al. (2007) |
| Mexico City, Mexico (2006) | 8.4 | 2.6 | Apel et al. (2010) |
| Los Angeles, CA, USA (2010) | 11.2 | 4.1 | de Gouw et al. (2017) |
| Changdao Island, China (2011) | 5.3 | 1.4 | Yuan et al. (2013) |
| Beijing, China (2011) | 4.4 | 1.4 | Wang et al. (2014) |
| London, UK ()2012) | 10.3 | 6.2 | |
| Houston, TX, USA (2013) | 12.0 | 15.8 | |
| Seoul, South Korea (2016) | 5.4 | 2.1 | |

**Table S8**. Emission ratios of non-BTEX aromatics used in this study. If no reference is listed, then the emission ratio was calculated using Eq. 3.

| Dataset Location | Emission Ratios (ppbv aromatic/ppmv CO) | | | References |
| --- | --- | --- | --- | --- |
| | Trimethylbenzenes | Ethyltoluenes | Propylbenzene | |
| NE USA, Aircraft (2002) | 0.71 | 0.58 | 0.14 | Warneke et al. (2007) |
| Los Angeles, CA, USA (2010) | 1.47 | 0.56 | 0.13 | de Gouw et al.(2017) |
| Beijing, China (2011) | 0.57 | 0.41 | 0.09 | Wang et al. (2014) |
| London, UK (2012) | 0.49 | 0.23 | 0.58 | |
| New York City, NY, USA (2015) | 0.71 | 0.58 | 0.14 | Warneke et al. (2007) |

**Table S9**. Normalized mass concentration of primary organic aerosol (POA/CO) measured in various campaigns, used to determine SVOC emission ratios.

| Dataset Location | Normalized Mass Concentration ($\mu g\ sm^{-3}\ ppmv^{-1}$) | | References |
| --- | --- | --- | --- |
| | HOA/CO | Other POA/CO | |
| NE USA (2002) | 12.2 | - | de Gouw et al. (2005) |
| Los Angeles, CA, USA (2010) | 5.3 | 7.7 | Hayes et al. (2013) |
| Beijing, China (2011) | 6.1 | 9.9 | Hu et al. (2016) |
| London, UK (2012) | 17.9 | 14.1 | Young et al. (2015) |
| New York City, NY, USA (2015) | 5.6 | 14.4 | Schroder et al. (2018) |

**Table S10.** Comparison of estimated VOC emission ratios from two studies from Mexico City (Apel et al. 2010; Bon et al. 2011), one study from Los Angeles (de Gouw et al. 2017), and this study.

| VOC Ratio | Apel et al. (2010) Downtown MC | This Study | Apel et al. (2010) Suburbs MC | Bon et al. (2011) Outskirt MC | This Study | de Gouw et al. (2017) LA | This Study |
|---|---|---|---|---|---|---|---|
| Ethane | 7.4 | 8.2 | 3.0 | 21.5 | 8.2 | 16.5 | 18.9 |
| Propane | 41.5 | 36.9 | 49.3 | 61.7 | 38.4 | 13.4 | 14.0 |
| n-Butane | 15.1 | 14.9 | 15.3 | 21.7 | 14.1 | 5.0 | 5.7 |
| i-Butane | 4.8 | 4.8 | 5.3 | 7.2 | 4.9 | 3.2 | 3.5 |
| n-Pentane | 2.1 | 2.9 | 2.1 | 2.5 | 2.1 | 3.4 | 3.4 |
| i-Pentane | 2.7 | 3.6 | 3.2 | 3.3 | 3.1 | 8.7 | 7.8 |
| n-Hexane | 1.5 | 1.9 | 1.3 | 1.5 | 1.2 | 1.4 | 1.7 |
| Ethene | 8.4 | 6.1 | 7.9 | 7.0 | 7.1 | 11.2 | 9.6 |
| Propene | 2.6 | 1.3 | 2.9 | 3.0 | 1.6 | 4.1 | 3.9 |
| Benzene | 0.9 | 1.0 | 1.2 | 1.2 | 1.3 | 1.3 | 1.4 |
| Toluene | 7.5 | 9.2 | 5.2 | 4.2 | 4.1 | 3.4 | 3.0 |
| Ethylbenzene | 0.9 | 0.8 | 0.4 | 4.3* | 0.4 | 0.6 | 0.6 |
| m+p-Xylene | 1.1 | 0.7 | 0.5 | No Data | 0.4 | 2.1 | 1.9 |
| o-Xylene | 0.4 | 0.2 | 0.2 | No Data | 0.2 | 0.8 | 0.7 |
| Trimethylbenzenes | No Data | No Data | No Data | No Data | No Data | 1.6 | 1.1 |
| Ethyltoluenes | No Data | No Data | No Data | No Data | No Data | 0.6 | 0.4 |
| Propylbenzene | No Data | No Data | No Data | No Data | No Data | 0.1 | 0.1 |

*In Bon et al. (2011), they reported the sum of C8 aromatics, which is the sum of ethylbenzene and xylenes

**Table S11**. Statistical analysis of the data used in Fig. 2 to determine if any point is influencing the slope, using the T-test, Cook's Distance test, and Difference in Fits test. For the T-test, the point is influential if the t value is < 0.05 while for the Cook's Distance and Difference in Fits test, the point is influential if the value is > 1.

|  |  |  |
|---|---|---|
|  |  |  |
|  |  |  |
|  |  |  |
|  |  |  |
|  |  |  |
|  |  |  |
|  |  |  |
|  |  |  |
|  |  |  |
|  |  |  |

| Campaign | T-test | Cook's Distance | Difference in Fits |
|---|---|---|---|
| NE US Ship | 0.63 | 0.06 | -0.29 |
| NE US Aircraft | 0.12 | 0.27 | 0.73 |
| Mexico City | 0.39 | 0.06 | 0.33 |
| Los Angeles | 0.32 | 0.08 | 0.38 |
| Changdao Island, China | 0.41 | 0.09 | -0.38 |
| Beijing | 0.42 | 0.06 | -0.32 |
| London | 0.31 | 0.13 | -0.48 |
| NYC | 0.90 | 0.00 | -0.05 |
| Seoul | 0.99 | 0.00 | 0.01 |

**Table S112**. Rate constants used throughout this study.

| Compound | Rate Constant (cm$^3$ molec.$^{-1}$ s$^{-1}$) | References |
|---|---|---|
| *Alkanes* | | |
| Ethane | $6.9 \times 10^{-12} \times \exp(-1000/T)$ | Atkinson et al. (2006) |
| Propane | $7.6 \times 10^{-12} \times \exp(-585/T)$ | Atkinson et al. (2006) |
| n-Butane | $9.8 \times 10^{-12} \times \exp(-425/T)$ | Atkinson et al. (2006) |
| i-Butane | $1.17 \times 10^{-17} \times T^2 \times \exp(213/T)$ | Atkinson and Arey (2003) |
| n-Pentane | $2.52 \times 10^{-17} \times T^2 \times \exp(158/T)$ | Atkinson and Arey (2003) |
| i-Pentane | $3.6 \times 10^{-12}$ | Atkinson and Arey (2003) |
| n-Hexane | $2.54 \times 10^{-14} \times T \times \exp(-112/T)$ | Atkinson and Arey (2003) |
| *Alkenes* | | |
| Ethene | $7.84 \times 10^{-12,a}$ | Atkinson et al. (2006) |
| Propene | $2.86 \times 10^{-11,a}$ | Atkinson et al. (2006) |
| *Aromatics* | | |
| Benzene | $2.3 \times 10^{-12} \times \exp(-190/T)$ | Atkinson et al. (2006) |
| Toluene | $1.8 \times 10^{-12} \times \exp(340/T)$ | Atkinson et al. (2006) |
| Ethylbenzene | $7 \times 10^{-12}$ | Atkinson and Arey (2003) |
| m+p-xylene | $1.87 \times 10^{-11,b}$ | Atkinson and Arey (2003) |
| o-xylene | $1.36 \times 10^{-11}$ | Atkinson and Arey (2003) |
| Trimethylbenzenes | $2.73 \times 10^{-12} \times \exp(730/T)$ | Bohn and Zetzsch (2012) |
| Ethyltoluenes | $1.2 \times 10^{-11}$ | Atkinson and Arey (2003) |
| Propylbenzene | $5.8 \times 10^{-12}$ | Atkinson and Arey (2003) |
| *S/IVOCs* | | |
| IVOCs C* = 4 - 6 | $2 \times 10^{-11}$ | Jathar et al. (2014) |
| IVOCs C* = 3 | $3 \times 10^{-11}$ | McDonald et al. (2018) |
| SVOCs & "aging" | $4 \times 10^{-11}$ | Tsimpidi et al. (2010) |
| *NO$_x$/NO$_y$* | | |
| OH + NO$_2$ | $1.23 \times 10^{-11,a}$ | Mollner et al. (2010) |

[a]Showing the rate constant at 298 K, 1013 hPa. However, for this study, we used the temperature
and pressure dependent formulation listed in each respective reference.
[b]This is the average of m-xylene and p-xylene rate constants.

**Table 13.** Parameters for VOC, IVOC, and SVOC aerosol yields. The yields are taken from Ma et al. (2017).

| Compound | Stoichiometric SOA yield High-NOx, 298 K ($\mu g\ m^{-3}$) | | | | |
|---|---|---|---|---|---|
| | 0.1 | 1 | 10 | 100 | 1000 |
| Benzene | | | | | |
| Toluene | N/A | 0.276 | 0.002 | 0.431 | 0.202 |
| Ethyltoluene | | | | | |
| Propylbenzenes | | | | | |
| Xylenes | N/A | 0.310 | 0.000 | 0.420 | 0.209 |
| Trimethylbenzenes | | | | | |
| IVOC C* = 6 | 0.007 | 0.090 | 0.206 | 0.350 | 0.00 |
| IVOC C* = 5 | 0.0498 | 0.0814 | 0.456 | 0.278 | 0.00 |
| IVOC C* = 4 | 0.053 | 0.103 | 0.464 | 0.266 | 0.00 |
| IVOC C* = 3 | 0.064 | 0.0914 | 0.562 | 0.209 | 0.00 |
| HOA C* = 2 | N/A | N/A | 0.28 | N/A | N/A |
| HOA C* = 1 | N/A | 0.18 | N/A | N/A | N/A |
| HOA C* = 0 | 0.12 | N/A | N/A | N/A | N/A |
| COA C* = 2 | N/A | N/A | 0.1881 | N/A | N/A |
| COA C* = 1 | N/A | 0.1188 | N/A | N/A | N/A |
| COA C* = 0 | 0.0594 | N/A | N/A | N/A | N/A |

**Table S14**. Table of GBD parameters, which is the mean of the draw values (see associated file) from the IHME website: http://ghdx.healthdata.org/record/global-burden-disease-study-2010-gbd-2010-ambient-air-pollution-risk-model-1990-2010.

| Parameter | IHD | Stroke | COPD | LC | ALRI |
|---|---|---|---|---|---|
| $\alpha$ | 1.4273 | 1.2641 | 15.224 | 114.74 | 2.2023 |
| $\beta$ | 0.04764 | 0.00722 | 0.00095 | 0.000141 | 0.000284 |
| $\rho$ | 0.376 | 1.314 | 0.684 | 0.741 | 1.183 |
| $PM_{2.5,Threshold}$ | 7.462 | 7.387 | 7.374 | 7.380 | 7.283 |

**Table S15**. Table of GEMM parameters. The GEMM parameters are from Burnett et al. (2018),
with the Chinese male cohort.

| Cause of Death | Age Range (years) | ϴ | Standard Error ϴ | α | μ | π |
|---|---|---|---|---|---|---|
| NCD + LRI | >25 | 0.1430 | 0.01807 | 1.6 | 15.5 | 36.8 |
| | 27.5 | 0.1585 | 0.01477 | 1.6 | 15.5 | 36.8 |
| | 32.5 | 0.1577 | 0.01470 | 1.6 | 15.5 | 36.8 |
| | 37.5 | 0.1570 | 0.01463 | 1.6 | 15.5 | 36.8 |
| | 42.5 | 0.1558 | 0.01450 | 1.6 | 15.5 | 36.8 |
| | 47.5 | 0.1532 | 0.01425 | 1.6 | 15.5 | 36.8 |
| | 52.5 | 0.1499 | 0.01394 | 1.6 | 15.5 | 36.8 |
| | 57.5 | 0.1462 | 0.01361 | 1.6 | 15.5 | 36.8 |
| | 62.5 | 0.1421 | 0.01325 | 1.6 | 15.5 | 36.8 |
| | 67.5 | 0.1374 | 0.01284 | 1.6 | 15.5 | 36.8 |
| | 72.5 | 0.1319 | 0.01234 | 1.6 | 15.5 | 36.8 |
| | 77.5 | 0.1253 | 0.01174 | 1.6 | 15.5 | 36.8 |
| | 85 | 0.1141 | 0.01071 | 1.6 | 15.5 | 36.8 |
| IHD | >25 | 0.2969 | 0.01787 | 1.9 | 12 | 40.2 |
| | 27.5 | 0.5070 | 0.02458 | 1.9 | 12 | 40.2 |
| | 32.5 | 0.4762 | 0.02309 | 1.9 | 12 | 40.2 |
| | 37.5 | 0.4455 | 0.02160 | 1.9 | 12 | 40.2 |
| | 42.5 | 0.4148 | 0.02011 | 1.9 | 12 | 40.2 |
| | 47.5 | 0.3841 | 0.01862 | 1.9 | 12 | 40.2 |
| | 52.5 | 0.3533 | 0.01713 | 1.9 | 12 | 40.2 |
| | 57.5 | 0.3226 | 0.01564 | 1.9 | 12 | 40.2 |
| | 62.5 | 0.2919 | 0.01415 | 1.9 | 12 | 40.2 |

 **Table 153 cont.**

| Cause of Death | Age Range (years) | θ | Standard Error θ | α | μ | π |
|---|---|---|---|---|---|---|
| IHD | 67.5 | 0.2612 | 0.01266 | 1.9 | 12 | 40.2 |
| | 72.5 | 0.2304 | 0.01117 | 1.9 | 12 | 40.2 |
| | 77.5 | 0.1997 | 0.00968 | 1.9 | 12 | 40.2 |
| | 85 | 0.1536 | 0.00745 | 1.9 | 12 | 40.2 |
| Stroke | >25 | 0.2720 | 0.07697 | 6.2 | 16.7 | 23.7 |
| | 27.5 | 0.4513 | 0.11919 | 6.2 | 16.7 | 23.7 |
| | 32.5 | 0.4240 | 0.11197 | 6.2 | 16.7 | 23.7 |
| | 37.5 | 0.3966 | 0.10475 | 6.2 | 16.7 | 23.7 |
| | 42.5 | 0.3693 | 0.09752 | 6.2 | 16.7 | 23.7 |
| | 47.5 | 0.3419 | 0.09030 | 6.2 | 16.7 | 23.7 |
| | 52.5 | 0.3146 | 0.08307 | 6.2 | 16.7 | 23.7 |
| | 57.5 | 0.2872 | 0.07585 | 6.2 | 16.7 | 23.7 |
| | 62.5 | 0.2598 | 0.06863 | 6.2 | 16.7 | 23.7 |
| | 67.5 | 0.2325 | 0.06190 | 6.2 | 16.7 | 23.7 |
| | 72.5 | 0.2051 | 0.05418 | 6.2 | 16.7 | 23.7 |
| | 77.5 | 0.1778 | 0.04695 | 6.2 | 16.7 | 23.7 |
| | 85 | 0.1368 | 0.03611 | 6.2 | 16.7 | 23.7 |
| COPD | >25 | 0.2510 | 0.06762 | 6.5 | 2.5 | 3.2 |
| Lung Cancer | >25 | 0.2942 | 0.06147 | 6.2 | 9.3 | 29.8 |
| LRI | >25 | 0.4468 | 0.11735 | 6.4 | 5.7 | 8.4 |

**Table S164**. Calculated premature mortality from PM with all aerosol (base mortality) and removing ASOA, using the IER method.

| Location[a] | Base Mortality | Mortality reduced due to removing ASOA | Percent mortality reduced due to removing ASOA |
|---|---|---|---|
| North America | 43,408 | 18,479 | 43% |
| Central America | 11,808 | 3,395 | 29% |
| South America | 31,214 | 10,100 | 32% |
| Africa | 258,294 | 14,869 | 6% |
| Western Europe | 305,754 | 31,880 | 10% |
| Eastern Europe | 195,749 | 16,003 | 8% |
| South Asia | 938,967 | 75,085 | 8% |
| Southeastern Asia | 135,433 | 31,886 | 24% |
| East Asia | 1,315,720 | 122,190 | 9% |
| Oceania | 95 | 27 | 28% |
| Rest of the World | 72,385 | 13,337 | 18% |
| Total | 3,308,957 | 337,224 | 10% |

[a]Locations defined by:

http://themasites.pbl.nl/tridion/en/themasites/_disabled_image/background/regions/index-2.html

**Table S175**. Calculated premature mortality from PM with all aerosol (base mortality) and removing ASOA, using the GEMM method.

| Location[a] | Base Mortality | Mortality reduced due to removing ASOA | Percent mortality reduced due to removing ASOA |
|---|---|---|---|
| North America | 178,793 | 24,892 | 14% |
| Central America | 58,516 | 7,298 | 12% |
| South America | 145,395 | 22,372 | 15% |
| Africa | 765,946 | 34,528 | 5% |
| Western Europe | 768,991 | 50,427 | 7% |
| Eastern Europe | 465,341 | 25,552 | 5% |
| South Asia | 2,285,903 | 166,228 | 7% |
| Southeastern Asia | 347,191 | 50,802 | 15% |
| East Asia | 2,487,349 | 220,264 | 9% |
| Oceania | 3,375 | 428 | 13% |
| Rest of the World | 269,769 | 35,051 | 13% |
| Total | 7,776,570 | 638,219 | 8% |

[a]Locations defined by:

http://themasites.pbl.nl/tridion/en/themasites/_disabled_image/background/regions/index-2.html

**Table S186**. List of total final consumption, in millions of tonnes of oil equivalent, of oil products and oil, for each organization. Total final consumption includes imports, and does not include exports (IEA, 2019).

| Organization | Industry | Transportation | Non-Energy |
|---|---|---|---|
| World | 307 | 2533 | 645 |
| OECD | 89 | 1147 | 326 |
| Africa | 18.4 | 115.4 | 7.9 |
| Non-OECD | 28.3 | 135 | 20 |
| Middle East | 33.5 | 126.3 | 47.5 |
| Non-OECD Europe and Eurasia | 35 | 101 | 53 |

**Supplemental figures for this study**

[Figure]

**Figure S1**. Comparison of HCHO measured by the DOAS (Stutz and Platt, 1996, 1997) and
Hantzsch reaction (Cárdenas et al., 2000) methods during the CalNex 2010 study in Pasadena,
CA, ground site (Ryerson et al., 2013).

[Figure]

**Figure S12.** Regression plot of SOA versus HCHO from different campaigns around the world that have not been previously published. Note, for (c), HCHO is 1.24×Hantzsch HCHO, to account for the differences between the two HCHO measurements during CalNex. Note, for (a), SOA is 0.5×OA, estimated from Young et al. (2015), and for (f), SOA is 0.8×OA, estimated from DeCarlo et al. (2010).

[Figure]

[Figure]

**Figure S23**. Regression plot of SOA versus PAN from different campaigns around the world that
have not been previously published. Note, for (a), SOA is 0.5×OA, estimated from Young et al.
(2015), and for (f), SOA is 0.8×OA, estimated from DeCarlo et al. (2010).

[Figure]

**Figure S34**. Regression plot of SOA versus Ox from different campaigns around the world that
have not been previously published. Note, for (a), SOA is 0.5×OA, estimated from Young et al.
(2015).

[Figure]

 **Figure S4**. Comparison of HCHO measured by the DOAS (Stutz and Platt, 1996, 1997) and
Hantzsch reaction (Cárdenas et al., 2000) methods during the CalNex 2010 study in Pasadena,
CA, ground site (Ryerson et al., 2013).

[Figure]

**Figure S5.** (a) Annually average CO emissions from HTaP. (b) Annually average benzene,
toluene, and xylenes (BTX) emissions, weighted by their OH reaction rate

$(E_{weight} = N \dfrac{\sum_i E_i k_{OH,i}}{\sum_i k_{OH,i}}$, $i = B, T, X$; N=3).

[Figure]

**Figure S6.** Emission ratio versus saturation concentration ($\log_{10}(c^*)$) for (a) Los Angeles, (b) NE US, aircraft, (c) Beijing, and (d) London. The emission ratios for VOCs ($\log_{10}(c^*) \geq 7$) were taken from de Gouw et al. (2017) and Ma et al. (2017) for Los Angeles, Warneke et al. (2007) for NE US, aircraft, and Wang et al. (2014) for Beijing while the VOC emission ratio for London is from Table S6 to Table S8. For VOCs between $\log_{10}(c^*)$ of 3 and 6 (IVOCs), the volatility distribution from McDonald et al. (2018), along with the ratio of IVOC to BTEX from Figure SI-6 and the emission ratio of BTEX (Table S6), were used to determine the emission ratio versus saturation concentration. Finally, for VOCs between $\log_{10}(c^*)$ 0 and 2 (SVOCs), the volatility distributions from Robinson et al. (2007) for non-fossil fuel POA and from Worton et al. (2014) for fossil fuel POA were used to convert the normalized POA mass concentration (Table S9) to VOC emission ratios. Note, the emission ratio versus saturation concentration for New York City, 2015, was similar to (b), as the emissions were similar (Fig. 5) and the BTEX for New York City is the same as NE US (Table S5).

[Figure]

**Figure S75.** 2-D VBS space defined by oxygen to carbon (O:C) ratio and saturation concentration [$\log_{10}(c^*)$] for different oxidation mechanisms and primary sources of OA precursors. Dashed boxes represent primary emissions, while the full boxes represent the secondary oxidation products. (A) and (B) represent different parameterizations for treating traditional anthropogenic and biogenic sources of SOA. Both parameterizations depict the oxidation of an 8-carbon precursor VOC. (A) represents the TSI, or aging, parameterization; (B) represents the MA, or wall-loss corrected, parameterization. (C) Represents the initial oxidation and aging pathway of P-IVOCs following the ZHAO parameterization. It should be noted that the carbon number corresponds to first generation aging and subsequent oxidation results in a 0.25 reduction in carbon number. (D) Represents the decadal aging of SVOCs by hydroxyl radicals. In (D), the full aging pathway of only the C21 species is depicted as an example, though all primary species are allowed to age until the $\log_{10}(c^*)$ = -2 bin. All emitted P-SVOC species undergo the same decadal aging scheme which begins from the saturation concentration bin of the emitted species.

[Figure]

**Figure S86**. CO emissions for the cities investigated here from HTAP (Janssens-Maenhout et al.,
2015).

[Figure]

**Figure S97**. (top) Total deaths associated to PM$_{2.5}$ (left) per 10×10 km$^2$ area and (right) summed
up for each country, using the Integrated Exposure-Response (IER) method (Burnett et al.,
2014). These values are derived from satellite. (bottom) Same as above, but using the Global
Exposure Mortality Model (GEMM) (Burnett et al., 2018) for PM$_{2.5}$ per 10×10 km$^2$ area (left)
and summed up for each country (right). Premature mortality was determined with PM$_{2.5}$ derived
by the methods described in van Donkelaar (2015), which includes satellite and ground-based
observations of aerosol.

[Figure]

**Figure S108**. Same as Fig. 8, where top are the results per 10×10 km² area for the attribution of
premature mortality to ASOA (people yr$^{-1}$, left) and fractional attribution of premature mortality
to ASOA for one year (right) by the IER method. See Fig. 8 for per country comparison.

[Figure]

**Figure S119**. Comparison of satellite retrieved PM$_{2.5}$ (upper left) versus modeled PM$_{2.5}$ (upper right). (Bottom) Fractional contribution of ASOA to total modeled PM$_{2.5}$.

[Figure]

**Figure S12**. Same as Fig. S10, but using the GEMM from Burnett et al. (2018). (top). (Left) Attribution of premature mortality to ASOA per $10 \times 10$ km$^2$ area (people yr$^{-1}$) and (Right) fractional attribution of preamture mortality to ASOA per $10 \times 10^2$ km for one year.

[Figure]

**Figure S13**. Same as Fig. S12 but summed up for each country for the (left) attribution of
premature mortality to ASOA (people yr$^{-1}$) and (right) the fractional attribution of premature
mortality to ASOA for one year.

———————¶

[Figure]

**Figure S142**. Comparison for surface level ozone upon reducing SOA precursors by 20%.

[Figure]

**Figure S15.** (top) Fractional contribution of CO emissions from residential sources to total emission sources from HTaP. (bottom) Fractional contribution of BTEX emissions from residential sources to total emission sources from HTaP. Residential sources include small-scale combustion, such as heating and cooking, which may include solid-fuel emissions.

---

## Author Response (AR2)

**Response to reviewers' comments on the paper "Anthropogenic Secondary Organic Aerosols Contribute Substantially to Air Pollution Mortality"**

We would like to thank the reviewers for their time and for their useful comments that have helped to improve and clarify our paper. For ease, comments from reviewers are in black, responses in blue, and new text added to paper in **bold blue**.

**Reviewer Comments**

R1.0 The authors have taken into consideration all comments provided by both reviewers and all issues raised are not addressed. Furthermore now it is clear that authors focus on the anthropogenic SOA from the oxidation of VOCs and why the solid fuel-laden anthropogenic SOA is not considered. Nevertheless I would suggest also to add this clarification in the title of the publication in order to include the fact that SOA is from AVOCs. Other than that the flow of the text is significantly ameliorated and all necessary clarifications that were needed are now added. Therefore I do not see any reason why not to proceed with the publications.

Thank you for the positive re-evaluation and support for publication.

However, it is not correct to say that "solid fuel-laden anthropogenic SOA is not considered." As we had explained in the responses, the impact of solid-fuel precursors and SOA is included in both of the two field studies (that do not show a different trend than the other studies) and the model inventories.

We have updated the title though to reflect that we have focused on SOA from anthropogenic VOCs:

**"Secondary Organic Aerosols from Anthropogenic Volatile Organic Compounds Contribute Substantially to Air Pollution Mortality."**

**Editor Comments**

E1.0 Thank you very much for the careful and significant revision of your manuscript, which has improved its readability and clarity. However, there are a few addition corrections to be made as follows.

E1.1 Consider revising the title of the manuscript as suggested by the reviewer and also perform the following corrections:

Please see response to comment R1.0.

E1.2 According to figure's 6 caption, figure 6d, shows the ratio of Figure 6c/Fig6a. I wonder why the values are provided at areas (for instance over the oceans) where Fig 6a shows ASOA equal to zero (white color).

We have added the following for clarification in the caption:

**"(d) Ratio between annual average modeled updated SIMPLE (b) and default VBS (a)."**

And this:

**"Note, for (a) - (b), values less than 0.05 µg m$^{-3}$ are white, and for (c), values less than 0.02 µg m$^{-3}$ are white."**

With this, not all values are 0; therefore, a ratio can occur.

E1.3 Line 654: I will not say that the 'new predictions of ASOA are accurate'. I would just say they are 'more accurate than earlier'. Please correct accordingly.

Added "more accurate than earlier".

E1.4 Line 198-202: Replace: 'included ASOA realistically' by 'treated explicitly ASOA'

Added "treated ASOA explicitly"

E1.5 Line 200: replace 'These models' by 'Most models'

Changed

E1.6 Line 202: replace 'over-redirecting POA' by 'over-predicting POA'

Corrected

E1.7 Supplement, lines 47 to 49. This is very confusing. Please explain better how the calculations are done and where the fractions or concentrations come from. What is divided by the amount of total PM measured in the respective size bin ? and also divided by the OA mass in the bin ?

We have added the following text to further clarify this point:

"To estimate the SVOC mass concentration in equilibrium with POA (Table S9), in each bin (e.g., $C* = 0, 1, 2$), the normalized POA mass concentration is first multiplied by the fraction of POA measured in each bin from literature. For other POA, which includes biomass burning and cooking OA, the fraction of POA found in $\log_{10}C* = 0, 1,$ and 2 are 0.22, 0.34, and 0.44, respectively (Robinson et al. 2007), and for vehicular POA, the fraction of POA found in $\log_{10}C* = 0, 1,$ and 2 are 0.42, 0.40, and 0.18, respectively (Worton et al. 2014). So, for example, for NE US, this would correspond to normalized POA mass concentrations (POA/$\Delta$CO) of 5.1, 4.9, and 2.2 µg sm$^{-3}$ ppmv$^{-1}$ for $\log_{10}C* = 0, 1,$ and 2, respectively. Then the total POA + SVOC normalized mass concentration for that bin is obtained by dividing the amount of material found in the particle-phase for that bin at the average temperature (~298 K) and OA mass concentration (~10 µg sm$^{-3}$). So, taking NE US as an example, for $\log_{10}C* = 0, 1,$ and 2, 9%, 50%, and 91% of the material, respectively, will be in the gas-phase versus the aerosol-phase, leading to the normalized mass concentration of SVOC as inputs into the model of 0.39, 3.8, and 17.1 µg sm$^{-3}$ ppmv$^{-1}$. The values of 9%, 50%, and 91% were used for NE US, Los Angeles, London, and Beijing, as the ambient temperatures were ~298 K. For New York City, as the study took place during winter, values of 3%, 22%, and 74% were used as the ambient temperature was ~273 K."

[revised manuscript text omitted]
, either reported value from studies (NE US aircraft (Warneke et al., 2007), Los Angeles (de Gouw et al., 2017), Beijing (Wang et al., 2014), and New York City (Warneke et al., 2007)) or estimated from Eq. 3 (London), to estimate IVOCs emitted in each region (Table S5). This ensures IVOC emissions used in our calculations properly reflect differences in mixtures of emission sources (e.g., mobile sources versus VCPs) that vary by continent for each field campaign. Additionally, we rely on inventories for estimating atmospheric abundances of IVOCs because it has been challenging to measure the full range of IVOC precursors that are emitted into urban air due to many of the IVOCs from VCPs being oxygenated VOCs. These compounds are challenging to measure using traditional instrumentation (e.g., gas chromatography-mass spectrometry), leading to potential underestimation of the IVOC emission ratios (Zhao et al., 2014, 2017; Lu et al., 2018). The bottom-up IVOC:BTEX ratios for the US, Beijing, and UK are described in greater detail below.

IVOC emissions are classified based on their vapor pressure (effective saturation concentration:

$10^3 < C^* < 10^6$ µg m$^{-3}$), with the vapor pressure estimated by the SIMPOL.1 model (Pankow and

Asher, 2008). The ASOA yields and rate constants for IVOC oxidation were parameterized with data from n-tridecane and n-pentadecane for gasoline and diesel emissions, respectively (Jathar et al., 2014), and for VCPs, the yields and rate constants for IVOC oxidation were parameterized with data from n-tetradecane (McDonald et al., 2018).

Similar to IVOCs, the ability to measure the full range of SVOCs emitted into urban air is challenging. Therefore, we estimate SVOC emission ratios relative to POA mass concentrations (Table S9), as described by Ma et al. (2017). For the hydrocarbon-like portion, we used the volatility distribution from Worton et al. (2014) to estimate SVOC, as this is associated with fossil fuel emissions from transportation (Zhang et al., 2005). For the other POA, we used the volatility distribution from Robinson et al. (2007), as this POA is typically cooking primary aerosol. These profiles were selected to be consistent with Ma et al. (2017).

To estimate the SVOC mass concentration in equilibrium with POA (Table S9), in each bin (e.g., $C^* = 0, 1, 2$), the normalized POA mass concentration is first multiplied by the fraction of POA measured in each bin from literature. For other POA, which includes biomass burning and cooking OA, the fraction of POA found in $\log_{10}C^* = 0, 1,$ and 2 are 0.22, 0.34, and 0.44, respectively (Robinson et al. 2007), and for vehicular POA, the fraction of POA found in $\log_{10}C^*$

$= 0, 1,$ and 2 are 0.42, 0.40, and 0.18, respectively (Worton et al. 2014). So, for example, for NE

US, this would correspond to normalized POA mass concentrations (POA/$\Delta$CO) of 5.1, 4.9, and

2.2 µg sm$^{-3}$ ppmv$^{-1}$ for $\log_{10}C^* = 0, 1,$ and 2, respectively. Then the total POA + SVOC

normalized mass concentration for that bin is obtained by dividing the amount of material found in the particle-phase for that bin at the average temperature (~298 K) and OA mass concentration (~10 µg sm$^{-3}$). So, taking NE US as an example, for $\log_{10}C^* = 0$, 1, and 2, 9%, 50%, and 91% of the material, respectively, will be in the gas-phase versus the aerosol-phase, leading to the normalized mass concentration of SVOC as inputs into the model of 0.39, 3.8, and 17.1 µg sm$^{-3}$ ppmv$^{-1}$. The values of 9%, 50%, and 91% were used for NE US, Los Angeles, London, and Beijing, as the ambient temperatures were ~298 K. For New York City, as the study took place during winter, values of 3%, 22%, and 74% were used as the ambient temperature was ~273 K. ~~To estimate the SVOC mass concentration in equilibrium with the POA (Table S9) in each bin, the POA mass concentration is first multiplied by the fraction of POA measured in each bin from literature. This yields the concentration of POA for that specific volatility bin. Then the total POA + SVOC concentration for that bin is obtained divided by the amount of material found in the particle-phase for that bin for the average temperature (~298 K) and OA mass concentration (~10 µg m-3). Then, the gas-phase SVOC concentration is calculated by multiplying the total concentration by the gas-phase fraction. Thus, e.g., SVOC in the C\* = 100 µg m-3 bin, ~91% of the SVOC mass will be found in the gas-phase.~~

Fig. S6 shows the calculated emission ratio versus saturation concentration (c\*) for the cities with emission inventories. The saturation concentration for SVOC was determined as part of the estimation procedure discussed above. For IVOC, the emission ratios for the different sources (gasoline, diesel, other fossil fuel sources, and VCP emissions) were split into the volatility bins, as in McDonald et al. (2018). Finally, for BTEX and non-BTEX aromatics, and other VOC emission ratios (see Fig. S6 for references for the other VOC emission ratios), CRC

(Rumble, 2019) or SIMPOL.1 (Pankow and Asher, 2008) (for estimating vapor pressures not in

CRC) was used to estimate the saturation concentrations.

**S1.1 US Emission Inventories**

*Anthropogenic VOC emissions*

The US emissions of VOCs is based on a mass balance estimate of the petrochemical industry reported by McDonald et al. (2018). Briefly, fuel sales and chemical product use are estimated from publicly available reports on energy use, chemical production, economic surveys, and freight shipments. Mobile source emission factors are from prior work quantifying both on-road and off-road engines (McDonald et al., 2013, 2015). Evaporative sources of transportation fuels are considered in addition to tailpipe exhaust (Pierson et al., 1999). VCP

emission factors are based on literature values, including from the indoor environment, and reported in McDonald et al. (2018). Other fossil energy sources of VOCs, such as from oil refineries and industry, are taken from official inventories reported by the California Air

Resources Board (CARB, 2013) or US Environmental Protection Agency (NEI, 2015).

McDonald et al. (2018) reported fossil-VOC emissions for the Los Angeles basin in the year

2010.

*Speciation of VOC emissions*

The total VOC emissions are speciated to estimate BTEX and IVOC emissions from petrochemical VOC sources. Briefly, gasoline and diesel exhaust, gasoline fuel, and headspace vapors are based on profiles reported in the literature from the Caldecott Tunnel (Gentner et al.,

2012, 2013). Speciation profiles of VCPs are based on California Air Resources Board surveys of architectural coatings (Davis, 2007) and consumer products (CCPR, 2015). Other industrial solvent uses and point/area source emissions are from the EPA SPECIATE (v4.4) database (EPA,

2014).

*Extrapolating IVOC/BTEX ratios from 2010 Los Angeles to other field campaigns*

In the ASOA mass closure estimation, three separate field campaigns are utilized from the US: NEAQS 2002 (Boston/New York City), CalNex 2010 (Los Angeles), and WINTER

2015 (New York City outflow). These field campaigns span two megacities (Los Angeles and

New York City), ~one decade, and two seasons (summer versus winter). Here, we discuss how each of these variables could affect the IVOC/BTEX emissions ratio. We focus the discussion on mobile sources and VCPs because these are the dominant contributors to BTEX and IVOCs.

The IVOC/BTEX emissions ratio could be affected by the population density of a city. It is well-established that per capita transportation fuel use decreases with increasing population density (Gately et al., 2015), whereas VCP usage is expected to scale with population. Relative to Los Angeles, the per capita fuel use in New York City is ~2 times lower (Gately et al., 2015), resulting in lower on-road transportation VOC emissions relative to VCPs. Because aromatics are mainly found in gasoline, whereas the IVOCs have a higher contribution from VCPs, the

IVOC/BTEX ratio is expected to be higher in New York City than Los Angeles.

To assess impacts of annual trends on the IVOC/BTEX ratio, we utilize long-term trend analyses of mobile source VOC emissions in Los Angeles (McDonald et al., 2013, 2015; Hassler et al., 2016). The main effect is that on-road gasoline emissions have decreased with time, both from the tailpipe of vehicles (McDonald et al., 2013) and of gasoline-related VOCs in ambient air measurements (Warneke et al., 2012). We utilize the EPA Trends Report to scale VOC

emissions for other anthropogenic sectors, including VCPs and industrial sources (https://www.epa.gov/air-emissions-inventories/air-pollutant-emissions-trends-data). The EPA

Trends Report suggests that VCP (or solvent) emissions decreased by ~30% between 2002 and

2010, including efforts to reduce the VOC content of architectural coatings (Matheson, 2002).

After 2010, the emissions have been slightly increasing, likely due to population growth.

Because both mobile sources and VCP emissions are decreasing with time, the IVOC/BTEX

emissions ratio is not significantly altered.

Lastly, the effects of seasonality influence on-road transportation emissions through: (i)

increased VOC emissions in winter relative to summer from cold-starting engines, and (ii) lower evaporative emissions due to colder ambient temperatures. We estimate that exhaust emissions from passenger vehicles increases by ~50% due to higher cold-start emissions in winter relative to summer based on the EPA MOVES model (MOVES, 2015). Evaporated gasoline and headspace vapors are known to exhibit a temperature-dependence (Rubin et al., 2006), and estimated to be ~20% and ~80% lower, respectively, based on typical wintertime temperatures of

New York City relative to summertime Los Angeles. Due to compensating factors between cold-start engines and evaporated fuels, the IVOC/BTEX emissions are not significantly affected by seasonality.

Overall, when taking into account differences in population density between Los Angeles and New York City, trends of mobile source and VCP emissions over time, and seasonality, the

IVOC/BTEX emission ratios range between ~2.3 to 2.7, which is a relatively small range. This sensitivity analysis helps explain why the enhancement observed in SOA scales with BTEX

levels in the urban atmosphere.

**S1.2 Beijing Emission Inventory**

*Anthropogenic VOC emissions*

The total VOC emissions of Beijing were developed following the bottom-up framework of the Multi-resolution Emission Inventory for China (MEIC) model (available at http://www.meicmodel.org), based on a technology-based methodology. The details of activity rates, emission factors, technology distribution, and control measures configured in the MEIC

model are summarized in a series of papers (Zhang et al., 2009; Zheng et al., 2014, 2018; Liu et al., 2015; Li et al., 2017, 2019).

In the MEIC model, a detailed four-level source classification system, representing sector, fuel/product, technology/solvent type, and end-of-pipe pollutant abatement facilities, was established by including over 700 emitting sources for each province. All anthropogenic sources, including power plants, industrial sources, volatile chemical products, fossil fuel burning in residential stoves, transportation were all considered.

Power plants are treated as point sources in the MEIC model. The VOC emissions were derived from the China coal-fired Power Plant Emissions Database (CPED, (Liu et al., 2015)), which is developed based on information of each unit on fuel type, fuel quality, combustion technology, etc.

Volatile chemical products are comprised of solvent use applied for architecture, vehicles, wood, and other industrial purposes, glue use, printing, pesticide use, and domestic solvent use.

The market share of waterborne and solvent-based paint is further taken into account for each source category. For the on-road transportation sector, the improved emissions developed by

Zheng et al. (2014) were integrated into the framework of MEIC, which estimated the vehicle population and emission factors at a county level. Both the VOC emissions in running mode and evaporation were considered. Emission standards covering pre-Euro I and Euro I to Euro V in

Beijing were applied for each vehicle type (Zheng et al., 2018; Li et al., 2019). Regarding oxygenated volatile organic compounds (OVOCs), the emission factors for on-road vehicles were corrected, as current emission factors are only for non-methane hydrocarbons (NMHC).

Correction ratios of 1.32, 1.08, 1.10, and 1.06 were applied for heavy-duty and light-duty diesel vehicles, and heavy-duty and light-duty gasoline vehicles, respectively, to the original values to comply with the follow-up speciation for the total VOC, following the method of Li et al. (2014,

2019).

*Speciation of VOC emissions*

Emissions by individual chemical species were developed based on the profile-assignment approach (Li et al., 2014, 2019). First, a "composite" profile database for

China was established by integrating the local profiles and supplementing it with the SPECIATE

v4.5 database for absent sources ((Simon et al., 2010), available at:

https://www.epa.gov/air-emissions-modeling/speciate-version-45-through-40). The detailed procedure for developing the composite profile database is illustrated in Li et al. (2014). In brief, for sources where there are significant differences in technology or legislation between China and western countries, only local profiles are used; otherwise, all candidate profiles are included for further compilation in the composite profile database. Local profiles covering most of the important sources were gathered and reviewed, including biofuel combustion, coal combustion, asphalt production, oil production, refinery, paint use, gasoline evaporation, gasoline vehicle exhaust, diesel vehicles, and so on, as detailed illustrated in Li et al. (2019).

Then, profiles for all combustion-related sources, including fossil fuel combustion in power plants, industry, residential, and transportation sectors were reviewed, and incomplete profiles that were absent from the OVOC fractions were corrected by appending the component of "OVOC" with fractions derived from the "complete" profiles for the same source. After OVOC correction, all "candidate" profiles were averaged by species to establish the composite profile database. Finally, the composite profile to each source was assigned by setting up the source linkage between the profile database and the inventory. Emissions by individual chemical species for each source were then further developed.

**S1.3 London/United Kingdom Emission Inventory**

*Anthropogenic VOC emissions*

The National Atmospheric Emissions Inventory (NAEI) estimates UK emissions of VOCs from anthropogenic sources following methods in the EMEP/EEA Emissions Inventory Guidebook (EMEP/EEA, 2016) for submission under the revised EU Directive 2016/2284/EU on National Emissions Ceilings (NECD), available at: https://eur-lex.europa.eu/legal-content/EN/TXT/PDF/?uri=CELEX:32016L2284&from=EN, and the United Nations Economic Commission for Europe (UNECE) Convention on Long-Range Transboundary Air Pollution (CLRTAP), available at:

http://www.ceip.at/ms/ceip_home1/ceip_home/reporting_instructions/reporting_programme/.

The NECD and CLRTAP define those VOC sources to be included and excluded from the national inventory (for example, emissions of NMVOCs from biogenic sources are not included).

The Guidebook provides estimation methodologies and default emission factors for each source category, although countries can use country-specific emission factors where these are deemed relevant. The NAEI currently covers organic emissions from around 400 individual source categories, with a large contribution from a diverse range of industrial processes and solvents, but with very few individually dominant sources.  The inventory then speciates emissions into

~650 individual compounds, or groups of compounds. Groupings of organics, for example, expressed as 'sum of all C14 compounds,' make up a substantial fraction of IVOC emissions, rather than being reported as individual compounds.

Emissions from the use of solvents and other volatile chemicals in industry and in consumer products, fuel production and distribution, food and drink manufacture and other non-combustion industrial processes accounted for 72% of all UK NMVOC emissions in 2017, according to the NAEI.  Both the solvent and industrial process sectors cover a diverse range of emission source categories:  the NAEI identifies 136 separate categories across the two sectors

For the road transport sector, the NAEI reports exhaust emissions of NMVOCs and its emissions from evaporative losses of fuel vapor from petrol vehicles.  Emissions from re-fueling at filling stations are reported separately under the fugitive emissions from the fuel distribution sector. The method used for road transport in the NAEI follows the method in the European

COPERT 5 model and described in the EMEP/EEA Emissions Inventory Guidebook.  The method uses average speed-related emission factors for hot exhaust emissions of total hydrocarbons for detailed vehicle categories (vehicle type, weight and/or engine size) and Euro standards for petrol cars, diesel cars, petrol and diesel light goods vehicles, rigid and articulated

HGVs, buses and coaches, and mopeds and motorcycles, and combines these with detailed traffic and fleet activity data derived from information provided by DfT. Separate estimates are made of methane emissions for each vehicle type and subtracted from the THC emissions to derive the

NMVOC emissions.

Evaporative emissions from vehicles are estimated in the NAEI, using the Guidebook method for three different processes: diurnal losses, hot soak, and running losses. Emissions are dependent on ambient temperature and fuel vapor pressure and different factors are provided for vehicles with and without carbon canisters for evaporative emission controls. All vehicles from

Euro 1 onwards are fitted with these devices; so, evaporative emission have been decreasing from the early 1990s with the penetration of these vehicles in the fleet. The method also takes into account the reduction in Reid Vapour Petrol of petrol sold in the UK since 2000, as required for compliance with the EU Fuel Quality Directive 98/70/EC, amended by Directive

2009/30/EC.

*Speciation of VOC emissions*

The NAEI is considered to adequately reflect annual real world emissions of BTEX (see, for example, eddy covariance flux comparisons in London by Langford et al. (2010) and Vaughn et al. (2017)); so, those values are taken directly from the NAEI and used here. IVOCs, and particularly long chain hydrocarbons, are included in many cases in the inventory as groups, but their emissions are known to be significantly underestimated when compared against field observations. We use the observations of Dunmore et al. (2015), made in wintertime central London in 2012, as guide to uprate NAEI emissions for IVOC species based on the estimated discrepancies between inventory and field observation reported for each carbon number above C10. This leads to some significant multipliers being applied to the inventory values, sometimes of the order 60 to 70. We assume that the same multipliers apply to all sources, since field data does not provide any means to attribute different factors to road transport IVOCs compared with IVOCs from VCP sources.

Since the NAEI represents a reporting of emissions for the purposes of compliance with international treaties, some fraction of those emissions are not released on the mainland UK. For this paper, offshore BTEX and IVOC emissions, arising for example from offshore oil and gas activity, aircraft in cruise, or shipping and emissions associated with overseas Crown Dependencies are removed from the UK total, since they play no part in determining the chemical environment of London. The annual NAEI totals are then divided equally to give a daily national emission.

**S2 ASOA Budget Analysis of Ambient Observations**

To calculate the ASOA budget, we used the observed BTEX (Table S5) and non-BTEX aromatic (Table S8) emission ratios, the emission inventories for IVOC (see above), and estimated SVOCs from the primary OA emissions (see above). The methods to calculate ASOA from emissions have been described in detail elsewhere (Hayes et al., 2015; Ma et al., 2017; Schroder et al., 2018), and are briefly described here. All calculations described were conducted with the KinSim v4.02 chemical kinetics simulator (Peng and Jimenez, 2019) within Igor Pro 7

(Lake Oswego, Oregon), and are summarized in Fig. S7. A typical average particle diameter for urban environments of ~200 nm (Seinfeld and Pandis, 2006) is used to estimate the condensational sink term for the partitioning of gas-to-particle, although condensation is always fast compared to the experiment timescales. Further, we assume an average 250 g mol$^{-1}$ molar mass for OA and an average SOA density of 1.4 g cm$^{-3}$ (Vaden et al., 2011; Kuwata et al., 2012).

Finally, all models are initialized with the campaign specific OA background (typically ~2 μg sm$^{-3}$) and POA (Table S9) for partitioning of gases to the particle phase, and ran at the average temperature for the campaign.

For the modeled VOCs (BTEX and non-BTEX aromatics), each species undergoes temperature-dependent OH oxidation (Table S12), forming four SVOCs that partition between gas- and particle-phase, using updated SOA yields that account for wall loss (Ma et al., 2017).

For IVOCs, the emission weighted SOA yields and rate constants from the "Zhao" option (Zhao et al., 2014) of Ma et al. (2017) are used, and the products are apportioned into three SVOC bins and one low-volatility organic compound (LVOC) bin (Fig. S7). Finally, SVOCs undergo photooxidation at a rate of $4 \times 10^{-11}$ cm$^3$ molecules$^{-1}$ s$^{-1}$ (Dzepina et al., 2009; Hodzic et al., 2010;

Tsimpidi et al., 2010; Hodzic and Jimenez, 2011; Hayes et al., 2015; Ma et al., 2017; Schroder et al., 2018), producing one product per oxidation step, with yields from Robinson et al. (2007) for cooking and other SVOCs and yields from Worton et al. (2014) for fossil fuel related SVOCs, as recommended by Ma et al. (2017). The products from SVOC and IVOC oxidation are allowed to further oxidize, as highlighted in Fig. S7 and described in prior studies (Hayes et al., 2015; Ma et al., 2017; Schroder et al., 2018). Generally, each product reacts at a rate of $4 \times 10^{-11}$ cm$^3$

molecules$^{-1}$ s$^{-1}$ to produce some product at one volatility bin lower, adding one oxygen to the compound for each oxidation (Dzepina et al., 2009; Tsimpidi et al., 2010; Hodzic and Jimenez,

2011; Hayes et al., 2015; Ma et al., 2017; Schroder et al., 2018). An update includes fragmentation for a fraction of the molecules that are oxidized, as described in Schroder et al.

(2018) and Koo et al. (2014). As shown in Fig. S7, fragmentation of the compound occurs as it is oxidized and goes down one volatility bin. For further oxidation of SVOCs from the oxidation of primary IVOCs, one oxygen is added and 0.25 carbon is removed per step, leading to an increase in mass of 1.03 (instead of 1.07) per oxidation step (Koo et al., 2014; Schroder et al., 2018). For further oxidation of products from primary SVOC emissions, one oxygen is added and 0.5

carbon is removed per step, leading to a decrease in mass of 1% (instead of 1.07) per oxidation step (Koo et al., 2014; Schroder et al., 2018).

**S3 GEOS-Chem Modeling**

The model used in this study is GEOS-Chem v12.0.0 (Bey et al., 2001; The International

GEOS-Chem User Community, 2018). This model is used for the following calculations: (1)

ASOA apportionment (Fig. 1), (2) apportionment of ASOA to total PM2.5 for premature mortality calculations (Sect. 5), and (3) sensitivity analysis for ASOA production and emissions on premature mortality calculations. GEOS-Chem is operated at 2°×2.5° horizontal resolution.

Goddard Earth Observing System – Forward Processing (GEOS-FP) assimilated data from the

NASA Global Modeling and Assimilation Office (GMAO) were used for input meteorological fields. The model was run for 2013 to 2018 to take into account interannual variability of meteorological impacts onto $PM_{2.5}$ (therefore, averaging $PM_{2.5}$ over variations in meteorology).

However, the HTAPv2 emission inventory, which was used for anthropogenic emissions (Janssens-Maenhout et al., 2015), was kept constant for the 5 years. Analysis of the HTAP

emissions, compared to other emission inventories, generally showed the highest correlation with observations ($R^2 = 0.54$), versus the other inventories (CEDS $R^2 = 0.26$, MACCity $R^2 = 0.00$, and

RETROv2 $R^2 = 0.04$), leading to the selection of this emission inventory. GEOS-Chem simulates gas and aerosol chemistry with ~700 chemical reactions. GEOS-Chem calculates the following

$PM_{2.5}$ species: sulfate, ammonium, nitrate (Park et al., 2006); black carbon and POA (Park et al.,

2005); SOA (Pye and Seinfeld, 2010; Marais et al., 2016); sea salt (accumulation mode only (Jaeglé et al., 2011)); and, dust (Duncan Fairlie et al., 2007).

**S3.1 Biogenic SOA**

For monoterpene and sesquiterpene SOAs, we used the default complex SOA scheme (without semi-volatile POA) using the two-product model framework (Pye and Seinfeld, 2010).

This scheme calculates initial oxidation of VOCs with OH, $O_3$, and $NO_3$, and resulting products are assigned to four different gas-phase semi-volatile species (TSOA0–3) based on volatilities (c* = 0.1, 1, 10, 100 µg m$^{-3}$). Aerosol and gas species fractions are calculated online using the partitioning theory, and all are removed by dry and wet deposition processes.

For isoprene SOA, we used the explicit isoprene chemistry developed by Marais et al.

(2016). All the isoprene-derived gas-phase products, including isoprene peroxy radical,

ISOPOOH, IEPOX, glyoxal, and methylglyoxal, are explicitly simulated. Irreversible heterogeneous uptake of precursors to aqueous aerosols are further calculated using online aerosol pH and surface area.

GEOS-Chem was used to estimate the relative fractions of the measured SOA in our studies between anthropogenic and biogenic (isoprene and monoterpene) sources (Fig. 1).

Extensive research has been conducted to evaluate and improve the models performance in predicting BSOA, as summarized in Table S3. Though these evaluations mainly occurred in the southeast US, a recent study has also included more global observations to compare with

GEOS-Chem (Pai et al., 2020). Generally, GEOS-Chem appears to overestimate biogenically derived SOA; however, the model predicted SOA is typically within the uncertainty of the AMS

(Table S3). The overestimation, though, would suggest that the fraction of urban SOA may be under-predicted by this method, whereas the BSOA may be over-predicted. Therefore, in urban regions, the amount of SOA from biogenic sources may be lower, especially after the rapid SOA

enhancements (within 12 to 24 equivalent photochemical hours that have been observed around the world (Nault et al., 2018)). Typically the BSOA is present as a regional background and subtracted for the analyses used in this work, which focus on strong urban plumes on top of that background (Hayes et al., 2013, 2015).

**S3.2 Default GEOS-Chem Sensitivity to ASOA Simulations**

For the sensitivity calculation using the "traditional" ASOA precursors, we used the two-product model framework (Pye and Seinfeld, 2010). Benzene, toluene, and xylene are oxidized with OH and converted to peroxy radicals. These peroxy radicals react with $HO_2$ or NO, resulting in non-volatile ASOA ($HO_2$ pathway, ASOAN species in GEOS-Chem) or semi-volatile ASOA tracers (NO pathway, ASOA1-3 in GEOS-Chem). As is the case for monoterpene and sesquiterpene SOA above, GEOS-Chem calculates online partitioning and dry/wet deposition processes for semi-volatile ASOA tracers. Other conditions including mortality calculation are kept the same as the base simulation above.

**S4 Ozone Sensitivity to ASOA Simulations**

A potential issue in the attribution of premature mortality to AOSA is that reducing emissions that lead to ASOA is that this may impact ozone concentrations. A sensitivity analysis was conducted, where the ASOA emissions were reduced by 20% (Fig. S14). In general, there is a less than 1% reduction in total ozone concentration in the boundary layer. This is due to the fact that the most important AVOCs that contribute to ozone formation are light alkenes (e.g., ethylene and propylene, Fig. 2), which are not ASOA precursors. Though the reaction rate constant of the ASOA precursors is generally high (Table S12), the concentration of the precursors is low and they thus account for a low percentage of the total ozone production potential (Table S5 through Table S9). For example, the measured OH reactivity (Sect. 3) for two different urban regions was between 15 to 25 $s^{-1}$ (Griffith et al., 2016; Whalley et al., 2016)

while the OH reactivity for the ASOA precursors for the same region was between 2 to 4 $s^{-1}$. The small contribution to the OH reactivity is in line to the minimal impact to the ozone concentration observed in Fig. S14.

**S5 Error Analysis of Observations**

The errors that will be discussed here are in reference to Fig. 2 and Fig. 4 and Table S4

either come from the 1σ uncertainty in the slopes (the SOA versus $O_x$, HCHO, or PAN values) or propagation of uncertainty in observations. For SOA, we estimate the 1σ uncertainty of ~15%, which is lower than the typical 1σ uncertainty of the AMS (Bahreini et al., 2009) due to the careful calibrations and excellent intercomparisons in the various campaigns (see Table 1 for references for the AMS comparisons). For ΔCO, the largest uncertainty is associated with the

CO background (Hayes et al., 2013; Nault et al., 2018), and is estimated to be ~10% at 0.5

photochemical equivalent days (Hayes et al., 2013). The uncertainty in the emission ratios is

~10% (Wang et al., 2014; de Gouw et al., 2017); though, it may be higher for the values calculated here due to the uncertainty in CO background, rate constants, and photochemical age.

Therefore, for Fig. 2a, the uncertainty in the y-values is 18% and the uncertainty in the x-values is 10%. For Fig. 4, the uncertainty in the measurement is 21%.

Another potential source of uncertainty may stem from the fit of the data in Fig. 2a, as the data point from Seoul (KORUS-AQ) could be impacting the fit due to the difference in its value compared to the other locations. Statistical analysis for the influence of the data from Seoul on the figure was conducted, including a T-test, Cook's Distance test, and Difference in Fits test (Table S11). All three statistical tests show that the data from Seoul (and all the data in general)

is not overly influencing the reported slope.

A further potential source of uncertainty in this analysis is the calculated VOC emission ratios for the studies that did not have ratios published previously (Houston 2000, London,

Houston 2013, and Seoul). To investigate how well Eq. 3 does in estimating the VOC emission ratios, a comparison of the estimated VOC emission ratios versus previously published ratios for two different cities, Mexico City (Apel et al., 2010; Bon et al., 2011) and Los Angeles (de Gouw et al., 2017) was made (Table S10). Also, for Mexico City, two locations, an urban and a suburban site, were compared both against each other (Apel et al., 2010; Bon et al., 2011) and the calculated values from Eq. 3.

First, as shown in Table S10, even for the same location (suburban Mexico City), different values in the emission ratio, especially for the alkanes, can be observed, by as much as a factor of 7. This can be partially explained by differences in how the emission ratios were determined. For both Apel et al. (2010) and Bon et al. (2011), the authors took the slope of

VOCs versus CO and used different regression techniques and different time periods. Comparing their technique with ours, we generally estimate VOC emission ratios within 50% of the reported values, and the estimation improves for shorter lived compounds (e.g., aromatics). However, de

Gouw et al. (2017) more carefully took chemistry into consideration for any potential losses of the VOCs prior to observation to determine emission ratios, similar to this study. We believe the comparison with de Gouw et al. (2017) provides a more useful comparison in the method presented here. We find, at most, a 30% difference in the emission ratios, with an average difference of 4±15% for all compounds. Thus, from this analysis, we conclude that (1) there is large variability in VOC emission ratios across urban areas around the world, which has been highlighted in other studies (Warneke et al., 2007), and (2) the method that considers losses of

VOCs is the more accurate procedure to estimate VOC emissions and leads to the best reproducibility across studies and lowest uncertainty (< 30%, ~4% on average).

**Supporting Information Tables**

**Table S1.** List of instruments whose observations are used in this study. In some cases $\Delta$SOA/$\Delta$CO (Table S4), SOA versus $O_x$ slope (Table S4), or VOC emission ratios (Table S5 through Table S8) had already been reported, and, in those cases, we use the previous literature reports in our analyses.

| Location | SOA | $O_x$ | HCHO | PAN | VOCs | CO |
|---|---|---|---|---|---|---|
| Houston, TX, USA (2000) | Q-AMS[a] | CL & UV Absorpion[b] | DOAS[c] | GC-ECD[d] | GC-FID, GC-MS[e] | Infrared Absortion[f] |
| Mexico City, Mexico (2006) | HR-ToF-AMS[g] | CL[h] | TDLAS[i] | CIMS[j] | WAS[k] | UV RF[l] |
| Los Angeles, CA, USA (2010) | HR-ToF-AMS[g] | CL & UV Absorption[m] | Average of DOAS[c] & Hantzsch Reaction[n] | GC-ECD[d] | GC-MS[o] | UV RF[l] |
| Beijing, China (2011) | HR-ToF-AMS[g] | CL & UV Absorption[p] | PTR-MS[q] | GC-ECD[r] | GC-FID[s] | IR Absorption[p] |
| London, UK (2012) | C-ToF-AMS[t] | CL & UV Absorption[u] | Hantzsch Reaction[n] | GC-ECD[v] | GC-FID & GC×GC-FID[w] | UV RF[l] |
| Houston, TX, USA (2013) | HR-ToF-AMS[g] | CL[x] | Average of LIF[y] & CAMS[z] | CIMS[j] | WAS[k] | DACOM[aa] |
| Seoul, South Korea (2016) | HR-ToF-AMS[g] | CL[h] | CAMS[z] | CIMS[j] | WAS[k] | DACOM[aa] |

[a]Quadrupole Aerosol Mass Spectrometer (Q-AMS) (Jayne et al., 2000)
[b]Chemiluminescence (CL) and UV Absorption (Williams et al., 1997)
[c]Differential Optical Absorption Spectrometry (DOAS) (Stutz and Platt, 1996, 1997)
[d]Gas chromatography-electron capture detector (GC-ECD) (Williams et al., 2000; Roberts et al., 2002)
[e]Gas chromatography-flame ionization detector (GC-FID) and gas chromatography mass spectrometer (Roberts et al., 2001)
[f]TECO Model 48s IR gas-filter
[g]High Resolution Time-of-Flight Aerosol Mass Spectrometer (HR-ToF-AMS) (DeCarlo et al., 2006)
[h]Chemiluminescence (CL) and UV Absorption (Weinheimer et al., 1994)
[i]Tunable diode laser absorption spectroscopic (TDLAS) measurements (Fried et al., 2003)
[j]Chemical ionization mass spectrometer (CIMS) (Huey L Tanner D Slusher D Dibb J Arimoto R Chen G Davis D Buhr M Nowak J Mauldin R Eisele F, 2004; Slusher et al., 2004; Kim et al., 2007)
[k]Whole air sample, followed by analysis with GC-FID and/or GC-MS (Blake et al., 2003)
[l]UV Resonance Fluorescence (RF) (Gerbig et al., 1999)

[m]Chemiluminescence (CL) and UV Absorption (Hayes et al., 2013)
[n]Hantzsch reaction (Cárdenas et al., 2000)
[o]Gas chromatograph mass spectrometer (Gilman et al., 2010)
[p]Chemiluminescence (CL), UV Absorption, and IR Absorption (Hu et al., 2016)
[q]Proton transfer reaction mass spectrometer (PTR-MS) (Warneke et al., 2011)
[r]Gas chromatography electron capture detector (GC-ECD) (Zhang et al., 2017)
[s]Gas chromatography flame ionization detector (GC-FID) (Wang et al., 2014)
[t]Compact Time-of-Flight Aerosol Mass Spectrometer (C-ToF-AMS) (Drewnick et al., 2005)
[u]Chemiluminescence (CL) and UV Absorption (Whalley et al., 2016)
[v]Gas chromatography electron capture detector (GC-ECD) (Whalley et al., 2016)
[w]Gas chromatography flame ionization detector (GC-FID) (Dunmore et al., 2015)
[x]Chemiluminescence (CL) (Ryerson et al., 1999; Pollack et al., 2010)
[y]Laser induced fluorescence (LIF) (Cazorla et al., 2015)
[z]Compact Atmospheric Multi-species Spectrometer (CAMS) difference frequency absorption
spectrometer (Weibring et al., 2010)
[aa]Tunable diode laser absorption spectroscopy (Sachse et al., 1987)

**Table S2.** Concentrations of PM$_1$ components shown in Fig. 1. References for the measurements can be found in Table 1.

| Dataset Location | Average Concentration (µg sm$^{-3}$) of submicron aerosol under standard temperature and pressure | | | | |
|---|---|---|---|---|---|
| | SOA | HOA | SO$_4$ | NO$_3$ | NH$_4$ |
| Houston, TX, USA (2000) | 2.7 | 0.7 | 4.9 | 0.4 | 1.5 |
| Northeast USA (2002) | 4.9 | 0.5 | 2.0 | 0.3 | 0.7 |
| Tokyo, Japan (2004) | 6.0 | 1.5 | 4.4 | 0.9 | 4.0 |
| Mexico City, Mexico (2006) | 11.2 | 4.8 | 1.9 | 6.0 | 2.5 |
| Paris, France (2009) | 1.9 | 1.1 | 1.2 | 0.5 | 0.6 |
| Los Angeles, CA, USA (2010) | 5.0 | 2.0 | 2.9 | 3.6 | 2.1 |
| Changdao Island, China (2011) | 9.4 | 4.4 | 8.3 | 12.2 | 6.5 |
| Beijing, China (2011) | 17.1 | 8.9 | 22.0 | 16.8 | 13.7 |
| London, UK (2012) | 2.7 | 1.6 | 1.4 | 2.7 | 1.3 |
| Houston, TX, USA (2013) | 3.7 | 0.0 | 2.7 | 0.1 | 0.6 |
| New York City, NY, USA (2015) | 0.8 | 0.7 | 1.2 | 1.4 | 0.4 |
| Seoul, South Korea (2016) | 11.9 | 1.3 | 5.0 | 7.9 | 4.4 |

**Table S3.** Table summarizing the results of recent GEOS-Chem performance evaluations for
modeling BSOA.

| Study | Observed Data | Species | Details |
|---|---|---|---|
| Fisher et al. (2016)[a] | SEAC⁴RS, below 1 km (spatial pattern), below 500 m (bias) | Isoprene | Spatial patterns well captured, and biases are +34% for isoprene and +3% for monoterpenes |
| | | Monoterpene | |
| | | Organic Nitrates from Isoprene | Spatial patterns well captured, and biases are -0.6% for first- and -35% for second-generation isoprene nitrates |
| | SEAC⁴RS, 0 - 4 km vertical profiles | Isoprene | Agreed well but GEOS-Chem somewhat overestimated observed concentrations near 1km |
| | | Monoterpene | |
| | | HCHO | |
| | | Organic Nitrates from Isoprene | Agreed within measurement uncertainties |
| | SOAS, at the surface | Isoprene | Underestimated isoprene and monoterpenes (-28% and -54%), but overestimated first- and second- generation isoprene nitrates (+85% and +43%) |
| | | Monoterpene | |
| | | HCHO | |
| | | Organic Nitrates from Isoprene | |
| Travis et al. (2016) | SEAC⁴RS, 0 - 12 km | First Generation from Isoprene Nitrates | Good agreement for ISOPOOH and ISOPN, underestimation of HPALDs by a factor of two |
| | | ISOPOOH | |
| | | HPALDS | |
| Marais et al. (2016) | SOAS, at the surface | IEPOX-SOA | Good agreement for isoprene derived aerosols, mean concentrations were almost the same |
| | | ISOPOOH-SOA | |
| | SEAC⁴RS, below 2 km (spatial pattern) | IEPOX-SOA | Spatial patterns well captured |

[a]This study decreased isoprene emissions by 15% and doubled monoterpene emissions of
MEGANv2.1.

**Table S3 cont.**

| Study | Observed Data | Species | Details |
|---|---|---|---|
| Kaiser et al. (2018)[a] | SEAC[4]RS | Isoprene | All were overestimated, except for first generation isoprene nitrates |
| | | HCHO | |
| | | ISOPOOH | |
| | | MVK + MACR | |
| | | First Generation Isoprene Nitrates | |
| Pai et al. (2020) | 15 airborne campaigns (SEAC[4]RS, GoAmazon, SENEX, OP3, etc.) | OA under biognic dominant conditions | Slight overestimation, but generally very similar in magnitude |

[a]NEI $NO_x$ emissions other than power plants decreased by 60%, soil $NO_x$ emissions were reduced by 50% across the Midwestern US. With the decrease of $NO_x$ emissions, ISOPOOH concentrations were increased in GEOS-Chem.

**Table S4**. Dilution-corrected SOA concentrations at 0.5 equivalent days and slopes of SOA
versus $O_x$, HCHO, and PAN used in Fig. 2 and Fig. 3. References for the values can be found
either in Table 1 or found in Fig. S2 through Fig. S4. Uncertainty is 1σ, and either represents
propagation in uncertainty in measurements (see Sect. S5) for ΔSOA/ΔCO or uncertainty in
slopes for SOA versus the three photochemical species.

| Dataset Location | ΔSOA/ΔCO at 0.5 eq. days | SOA vs. $O_x$ Slopes | SOA vs. HCHO Slopes | SOA vs. PAN Slopes |
|---|---|---|---|---|
| Houston, TX, USA (2000) | | 0.04±0.01[a] | 0.32±0.08 | 1.41±0.46 |
| Northeast USA (2002) | 16±3[b] 48±9[c] | | | |
| Mexico City, Mexico (2003) | | 0.14±0.01[a] | | |
| Tokyo, Japan (2004) | | 0.19±0.01[a] | | |
| Mexico City, Mexico (2006) | 58±10 | 0.16±0.01 | 1.60±0.06 | 5.60±0.30 |
| Paris, France (2009) | | 0.14±0.01[a] | | |
| Pasadena, CA, USA (2010) | 59±11 | 0.16±0.01 | 1.93±0.02 | 5.41±0.12 |
| Changdao Island, China (2011) | 23±4 | | | |
| Beijing, China (2011) | 31±6 | 0.21±0.01 | 3.90±0.15 | 7.42±0.46 |
| London, UK (2012) | 54±10 | 0.13±0.01 | 0.36±0.02 | 3.37±0.41 |
| Houston, TX, USA (2013) | | 0.16±0.01 | 1.52±0.13 | 6.92±0.58 |
| New York City, NY, USA (2015) | 33±6 | | | |
| Seoul, South Korea (2016) | 107±19 | 0.29±0.02 | 3.73±0.26 | 10.13±0.52 |

[a]Missing reported uncertainty; therefore, assuming ±0.01, as that is typical for other campaigns
[b]From de Gouw et al. (2005). [c]From Kleinman et al. (2007).

**Table S5**. Emission ratios of BTEX aromatics used in this study. If no reference is listed, then
the emission ratio was calculated using Eq. 3.

| Dataset Location | Emission Ratios (ppbv aromatic/ppmv CO) | | | | | References |
|---|---|---|---|---|---|---|
| | Benzene | Toluene | Ethylbenzene | m+p-xylene | o-xylene | |
| Houston, TX, USA (2000) | 2.6 | 3.5 | 0.6 | 2.8 | 0.8 | |
| NE USA, Ship (2002) | 0.9 | 2.0 | 0.2 | 0.6 | 0.3 | Baker et al. (2008) |
| NE USA, Aircraft (2002) | 0.8 | 2.9 | 0.4 | 1.2 | 0.5 | Warneke et al. (2007) |
| Mexico City, Mexico (2006) | 0.9 | 7.5 | 0.9 | 1.1 | 0.4 | Apel et al. (2010) |
| Los Angeles, CA, USA (2010) | 1.3 | 3.4 | 0.6 | 2.1 | 0.8 | de Gouw et al. (2017) |
| Changdao Island, China (2011) | 2.3 | 1.9 | 0.5 | 1.3 | 0.4 | Yuan et al. (2013) |
| Beijing, China (2011) | 1.2 | 2.4 | 1.0 | 1.6 | 0.6 | Wang et al. (2014) |
| London, UK (2012) | 1.8 | 6.3 | 1.2 | 2.2 | 1.1 | |
| Houston, TX, USA (2013) | 2.3 | 3.0 | 0.6 | 3.9 | 1.2 | |
| New York City, NY, USA (2015) | 0.8 | 2.9 | 0.4 | 1.2 | 0.5 | Warneke et al. (2007)[a] |
| Seoul, South Korea (2016) | 1.1 | 13.1 | 2.4 | 3.3 | 2.3 | |

[a]Using the emissions from Warneke et al. (2007) instead of Schroder et al. (2018) as Schroder et
al. found significant uncertainty in the emissions calculated from observations.

**Table S6**. Emission ratios of alkanes used in this study. If no reference is listed, then the emission ratio was calculated using Eq. 3.

| Dataset Location | Emission Ratios (ppbv alkane/ppmv CO) | | | | | | | References |
|---|---|---|---|---|---|---|---|---|
| | Ethane | Propane | n-Butane | i-Butane | n-Pentane | i-Pentane | n-Hexane | |
| Houston, TX, USA (2000) | 40.9 | 24.3 | 9.0 | 14.7 | 3.1 | 10.0 | 3.1 | |
| NE USA, Ship (2002) | 8.3 | 2.3 | 1.8 | 1.3 | 1.0 | 2.8 | 0.9 | Baker et al. (2008) |
| NE USA, Aircraft (2002) | 9.9 | 9.0 | 2.4 | 1.3 | 2.0 | 5.4 | 0.6 | Warneke et al. (2007) |
| Mexico City, Mexico (2006) | 7.4 | 41.5 | 15.1 | 4.8 | 2.1 | 2.7 | 1.5 | Apel et al. (2010) |
| Los Angeles, CA, USA (2010) | 16.5 | 13.4 | 5.0 | 3.2 | 3.4 | 8.7 | 1.4 | de Gouw et al. (2017) |
| Changdao Island, China (2011) | 7.7 | 4.5 | 2.5 | 1.2 | 1.0 | 1.5 | 0.5 | Yuan et al. (2013) |
| Beijing, China (2011) | 4.3 | 3.9 | 2.5 | 2.5 | 1.2 | 2.0 | 0.6 | Wang et al. (2014) |
| London, UK (2012) | 33.0 | 17.8 | 17.3 | 8.4 | 4.6 | 11.3 | 1.3 | |
| Houston, TX, USA (2013) | 86.5 | 37.3 | 14.6 | 10.6 | 7.0 | 10.5 | 3.0 | |
| Seoul, South Korea (2016) | 16.1 | 0.4 | 6.0 | 3.4 | 3.1 | 3.7 | 1.7 | |

**Table S7**. Emission ratios of alkenes used in this study. If no reference is listed, then the emission ratio was calculated using Eq. 3.

| Dataset Location | Emission Ratios (ppbv alkene/ppmv CO) | | References |
|---|---|---|---|
| | Ethene | Propene | |
| Houston, TX, USA (2000) | 24.4 | 28.4 | |
| NE USA, Ship (2002) | 4.4 | 1.1 | Baker et al. (2008) |
| NE USA, Aircraft (2002) | 4.9 | 1.4 | Warneke et al. (2007) |
| Mexico City, Mexico (2006) | 8.4 | 2.6 | Apel et al. (2010) |
| Los Angeles, CA, USA (2010) | 11.2 | 4.1 | de Gouw et al. (2017) |
| Changdao Island, China (2011) | 5.3 | 1.4 | Yuan et al. (2013) |
| Beijing, China (2011) | 4.4 | 1.4 | Wang et al. (2014) |
| London, UK ()2012) | 10.3 | 6.2 | |
| Houston, TX, USA (2013) | 12.0 | 15.8 | |
| Seoul, South Korea (2016) | 5.4 | 2.1 | |

**Table S8**. Emission ratios of non-BTEX aromatics used in this study. If no reference is listed, then the emission ratio was calculated using Eq. 3.

| Dataset Location | Emission Ratios (ppbv aromatic/ppmv CO) | | | References |
|---|---|---|---|---|
| | Trimethylbenzenes | Ethyltoluenes | Propylbenzene | |
| NE USA, Aircraft (2002) | 0.71 | 0.58 | 0.14 | Warneke et al. (2007) |
| Los Angeles, CA, USA (2010) | 1.47 | 0.56 | 0.13 | de Gouw et al.(2017) |
| Beijing, China (2011) | 0.57 | 0.41 | 0.09 | Wang et al. (2014) |
| London, UK (2012) | 0.49 | 0.23 | 0.58 | |
| New York City, NY, USA (2015) | 0.71 | 0.58 | 0.14 | Warneke et al. (2007) |

**Table S9**. Normalized mass concentration of primary organic aerosol (POA/CO) measured in various campaigns, used to determine SVOC emission ratios.

| Dataset Location | Normalized Mass Concentration ($\mu g\ sm^{-3}\ ppmv^{-1}$) | | References |
|---|---|---|---|
| | HOA/CO | Other POA/CO | |
| NE USA (2002) | 12.2 | - | de Gouw et al. (2005) |
| Los Angeles, CA, USA (2010) | 5.3 | 7.7 | Hayes et al. (2013) |
| Beijing, China (2011) | 6.1 | 9.9 | Hu et al. (2016) |
| London, UK (2012) | 17.9 | 14.1 | Young et al. (2015) |
| New York City, NY, USA (2015) | 5.6 | 14.4 | Schroder et al. (2018) |

**Table S10.** Comparison of estimated VOC emission ratios from two studies from Mexico City (Apel et al., 2010; Bon et al., 2011), one study from Los Angeles (de Gouw et al., 2017), and this study.

| VOC Ratio | Apel et al. (2010) Downtown MC | This Study | Apel et al. (2010) Suburbs MC | Bon et al. (2011) Outskirt MC | This Study | de Gouw et al. (2017) LA | This Study |
|---|---|---|---|---|---|---|---|
| Ethane | 7.4 | 8.2 | 3.0 | 21.5 | 8.2 | 16.5 | 18.9 |
| Propane | 41.5 | 36.9 | 49.3 | 61.7 | 38.4 | 13.4 | 14.0 |
| n-Butane | 15.1 | 14.9 | 15.3 | 21.7 | 14.1 | 5.0 | 5.7 |
| i-Butane | 4.8 | 4.8 | 5.3 | 7.2 | 4.9 | 3.2 | 3.5 |
| n-Pentane | 2.1 | 2.9 | 2.1 | 2.5 | 2.1 | 3.4 | 3.4 |
| i-Pentane | 2.7 | 3.6 | 3.2 | 3.3 | 3.1 | 8.7 | 7.8 |
| n-Hexane | 1.5 | 1.9 | 1.3 | 1.5 | 1.2 | 1.4 | 1.7 |
| Ethene | 8.4 | 6.1 | 7.9 | 7.0 | 7.1 | 11.2 | 9.6 |
| Propene | 2.6 | 1.3 | 2.9 | 3.0 | 1.6 | 4.1 | 3.9 |
| Benzene | 0.9 | 1.0 | 1.2 | 1.2 | 1.3 | 1.3 | 1.4 |
| Toluene | 7.5 | 9.2 | 5.2 | 4.2 | 4.1 | 3.4 | 3.0 |
| Ethylbenzene | 0.9 | 0.8 | 0.4 | 4.3* | 0.4 | 0.6 | 0.6 |
| m+p-Xylene | 1.1 | 0.7 | 0.5 | No Data | 0.4 | 2.1 | 1.9 |
| o-Xylene | 0.4 | 0.2 | 0.2 | No Data | 0.2 | 0.8 | 0.7 |
| Trimethylbenzenes | No Data | No Data | No Data | No Data | No Data | 1.6 | 1.1 |
| Ethyltoluenes | No Data | No Data | No Data | No Data | No Data | 0.6 | 0.4 |
| Propylbenzene | No Data | No Data | No Data | No Data | No Data | 0.1 | 0.1 |

*In Bon et al. (2011), they reported the sum of C8 aromatics, which is the sum of ethylbenzene and xylenes

**Table S11**. Statistical analysis of the data used in Fig. 2 to determine if any point is influencing the slope, using the T-test, Cook's Distance test, and Difference in Fits test. For the T-test, the point is influential if the t value is < 0.05 while for the Cook's Distance and Difference in Fits test, the point is influential if the value is > 1.

| Campaign | T-test | Cook's Distance | Difference in Fits |
|---|---|---|---|
| NE US Ship | 0.63 | 0.06 | -0.29 |
| NE US Aircraft | 0.12 | 0.27 | 0.73 |
| Mexico City | 0.39 | 0.06 | 0.33 |
| Los Angeles | 0.32 | 0.08 | 0.38 |
| Changdao Island, China | 0.41 | 0.09 | -0.38 |
| Beijing | 0.42 | 0.06 | -0.32 |
| London | 0.31 | 0.13 | -0.48 |
| NYC | 0.90 | 0.00 | -0.05 |
| Seoul | 0.99 | 0.00 | 0.01 |

 **Table S12**. Rate constants used throughout this study.

| Compound | Rate Constant (cm$^3$ molec.$^{-1}$ s$^{-1}$) | References |
|---|---|---|
| *Alkanes* | | |
| Ethane | $6.9\times10^{-12}\times\exp(-1000/T)$ | Atkinson et al. (2006) |
| Propane | $7.6\times10^{-12}\times\exp(-585/T)$ | Atkinson et al. (2006) |
| n-Butane | $9.8\times10^{-12}\times\exp(-425/T)$ | Atkinson et al. (2006) |
| i-Butane | $1.17\times10^{-17}\times T^2\times\exp(213/T)$ | Atkinson and Arey (2003) |
| n-Pentane | $2.52\times10^{-17}\times T^2\times\exp(158/T)$ | Atkinson and Arey (2003) |
| i-Pentane | $3.6\times10^{-12}$ | Atkinson and Arey (2003) |
| n-Hexane | $2.54\times10^{-14}\times T\times\exp(-112/T)$ | Atkinson and Arey (2003) |
| *Alkenes* | | |
| Ethene | $7.84\times10^{-12,a}$ | Atkinson et al. (2006) |
| Propene | $2.86\times10^{-11,a}$ | Atkinson et al. (2006) |
| *Aromatics* | | |
| Benzene | $2.3\times10^{-12}\times\exp(-190/T)$ | Atkinson et al. (2006) |
| Toluene | $1.8\times10^{-12}\times\exp(340/T)$ | Atkinson et al. (2006) |
| Ethylbenzene | $7\times10^{-12}$ | Atkinson and Arey (2003) |
| m+p-xylene | $1.87\times10^{-11,b}$ | Atkinson and Arey (2003) |
| o-xylene | $1.36\times10^{-11}$ | Atkinson and Arey (2003) |
| Trimethylbenzenes | $2.73\times10^{-12}\times\exp(730/T)$ | Bohn and Zetzsch (2012) |
| Ethyltoluenes | $1.2\times10^{-11}$ | Atkinson and Arey (2003) |
| Propylbenzene | $5.8\times10^{-12}$ | Atkinson and Arey (2003) |
| *S/IVOCs* | | |
| IVOCs C* = 4 - 6 | $2\times10^{-11}$ | Jathar et al. (2014) |
| IVOCs C* = 3 | $3\times10^{-11}$ | McDonald et al. (2018) |
| SVOCs & "aging" | $4\times10^{-11}$ | Tsimpidi et al. (2010) |
| *NO$_x$/NO$_y$* | | |
| OH + NO$_2$ | $1.23\times10^{-11,a}$ | Mollner et al. (2010) |

aShowing the rate constant at 298 K, 1013 hPa. However, for this study, we used the temperature and pressure dependent formulation listed in each respective reference.

bThis is the average of m-xylene and p-xylene rate constants.

[a]Showing the rate constant at 298 K, 1013 hPa. However, for this study, we used the temperature and pressure dependent formulation listed in each respective reference.

[b]This is the average of m-xylene and p-xylene rate constants.

**Table 13.** Parameters for VOC, IVOC, and SVOC aerosol yields. The yields are taken from Ma et al. (2017).

| Compound | Stoichiometric SOA yield High-NOx, 298 K ($\mu$g m$^{-3}$) | | | | |
|---|---|---|---|---|---|
| | 0.1 | 1 | 10 | 100 | 1000 |
| Benzene | | | | | |
| Toluene | N/A | 0.276 | 0.002 | 0.431 | 0.202 |
| Ethyltoluene | | | | | |
| Propylbenzenes | | | | | |
| Xylenes | N/A | 0.310 | 0.000 | 0.420 | 0.209 |
| Trimethylbenzenes | | | | | |
| IVOC C* = 6 | 0.007 | 0.090 | 0.206 | 0.350 | 0.00 |
| IVOC C* = 5 | 0.0498 | 0.0814 | 0.456 | 0.278 | 0.00 |
| IVOC C* = 4 | 0.053 | 0.103 | 0.464 | 0.266 | 0.00 |
| IVOC C* = 3 | 0.064 | 0.0914 | 0.562 | 0.209 | 0.00 |
| HOA C* = 2 | N/A | N/A | 0.28 | N/A | N/A |
| HOA C* = 1 | N/A | 0.18 | N/A | N/A | N/A |
| HOA C* = 0 | 0.12 | N/A | N/A | N/A | N/A |
| COA C* = 2 | N/A | N/A | 0.1881 | N/A | N/A |
| COA C* = 1 | N/A | 0.1188 | N/A | N/A | N/A |
| COA C* = 0 | 0.0594 | N/A | N/A | N/A | N/A |

**Table S14**. Table of GBD parameters, which is the mean of the draw values (see associated file)
from the IHME website:
http://ghdx.healthdata.org/record/global-burden-disease-study-2010-gbd-2010-ambient-air-pollut
ion-risk-model-1990-2010.

| Parameter | IHD | Stroke | COPD | LC | ALRI |
|---|---|---|---|---|---|
| $\alpha$ | 1.4273 | 1.2641 | 15.224 | 114.74 | 2.2023 |
| $\beta$ | 0.04764 | 0.00722 | 0.00095 | 0.000141 | 0.000284 |
| $\rho$ | 0.376 | 1.314 | 0.684 | 0.741 | 1.183 |
| $PM_{2.5,Threshold}$ | 7.462 | 7.387 | 7.374 | 7.380 | 7.283 |

**Table S15**. Table of GEMM parameters. The GEMM parameters are from Burnett et al. (2018),
with the Chinese male cohort.

| Cause of Death | Age Range (years) | θ | Standard Error θ | α | μ | π |
|---|---|---|---|---|---|---|
| | >25 | 0.1430 | 0.01807 | 1.6 | 15.5 | 36.8 |
| | 27.5 | 0.1585 | 0.01477 | 1.6 | 15.5 | 36.8 |
| | 32.5 | 0.1577 | 0.01470 | 1.6 | 15.5 | 36.8 |
| | 37.5 | 0.1570 | 0.01463 | 1.6 | 15.5 | 36.8 |
| | 42.5 | 0.1558 | 0.01450 | 1.6 | 15.5 | 36.8 |
| | 47.5 | 0.1532 | 0.01425 | 1.6 | 15.5 | 36.8 |
| NCD + LRI | 52.5 | 0.1499 | 0.01394 | 1.6 | 15.5 | 36.8 |
| | 57.5 | 0.1462 | 0.01361 | 1.6 | 15.5 | 36.8 |
| | 62.5 | 0.1421 | 0.01325 | 1.6 | 15.5 | 36.8 |
| | 67.5 | 0.1374 | 0.01284 | 1.6 | 15.5 | 36.8 |
| | 72.5 | 0.1319 | 0.01234 | 1.6 | 15.5 | 36.8 |
| | 77.5 | 0.1253 | 0.01174 | 1.6 | 15.5 | 36.8 |
| | 85 | 0.1141 | 0.01071 | 1.6 | 15.5 | 36.8 |
| | >25 | 0.2969 | 0.01787 | 1.9 | 12 | 40.2 |
| | 27.5 | 0.5070 | 0.02458 | 1.9 | 12 | 40.2 |
| | 32.5 | 0.4762 | 0.02309 | 1.9 | 12 | 40.2 |
| | 37.5 | 0.4455 | 0.02160 | 1.9 | 12 | 40.2 |
| IHD | 42.5 | 0.4148 | 0.02011 | 1.9 | 12 | 40.2 |
| | 47.5 | 0.3841 | 0.01862 | 1.9 | 12 | 40.2 |
| | 52.5 | 0.3533 | 0.01713 | 1.9 | 12 | 40.2 |
| | 57.5 | 0.3226 | 0.01564 | 1.9 | 12 | 40.2 |
| | 62.5 | 0.2919 | 0.01415 | 1.9 | 12 | 40.2 |

 **Table 15 cont.**

| Cause of Death | Age Range (years) | ϴ | Standard Error ϴ | α | μ | π |
|---|---|---|---|---|---|---|
| IHD | 67.5 | 0.2612 | 0.01266 | 1.9 | 12 | 40.2 |
| | 72.5 | 0.2304 | 0.01117 | 1.9 | 12 | 40.2 |
| | 77.5 | 0.1997 | 0.00968 | 1.9 | 12 | 40.2 |
| | 85 | 0.1536 | 0.00745 | 1.9 | 12 | 40.2 |
| Stroke | >25 | 0.2720 | 0.07697 | 6.2 | 16.7 | 23.7 |
| | 27.5 | 0.4513 | 0.11919 | 6.2 | 16.7 | 23.7 |
| | 32.5 | 0.4240 | 0.11197 | 6.2 | 16.7 | 23.7 |
| | 37.5 | 0.3966 | 0.10475 | 6.2 | 16.7 | 23.7 |
| | 42.5 | 0.3693 | 0.09752 | 6.2 | 16.7 | 23.7 |
| | 47.5 | 0.3419 | 0.09030 | 6.2 | 16.7 | 23.7 |
| | 52.5 | 0.3146 | 0.08307 | 6.2 | 16.7 | 23.7 |
| | 57.5 | 0.2872 | 0.07585 | 6.2 | 16.7 | 23.7 |
| | 62.5 | 0.2598 | 0.06863 | 6.2 | 16.7 | 23.7 |
| | 67.5 | 0.2325 | 0.06190 | 6.2 | 16.7 | 23.7 |
| | 72.5 | 0.2051 | 0.05418 | 6.2 | 16.7 | 23.7 |
| | 77.5 | 0.1778 | 0.04695 | 6.2 | 16.7 | 23.7 |
| | 85 | 0.1368 | 0.03611 | 6.2 | 16.7 | 23.7 |
| COPD | >25 | 0.2510 | 0.06762 | 6.5 | 2.5 | 3.2 |
| Lung Cancer | >25 | 0.2942 | 0.06147 | 6.2 | 9.3 | 29.8 |
| LRI | >25 | 0.4468 | 0.11735 | 6.4 | 5.7 | 8.4 |

**Table S16**. Calculated premature mortality from PM with all aerosol (base mortality) and removing ASOA, using the IER method.

| Location[a] | Base Mortality | Mortality reduced due to removing ASOA | Percent mortality reduced due to removing ASOA |
|---|---|---|---|
| North America | 43,408 | 18,479 | 43% |
| Central America | 11,808 | 3,395 | 29% |
| South America | 31,214 | 10,100 | 32% |
| Africa | 258,294 | 14,869 | 6% |
| Western Europe | 305,754 | 31,880 | 10% |
| Eastern Europe | 195,749 | 16,003 | 8% |
| South Asia | 938,967 | 75,085 | 8% |
| Southeastern Asia | 135,433 | 31,886 | 24% |
| East Asia | 1,315,720 | 122,190 | 9% |
| Oceania | 95 | 27 | 28% |
| Rest of the World | 72,385 | 13,337 | 18% |
| Total | 3,308,957 | 337,224 | 10% |

[a]Locations defined by:

http://themasites.pbl.nl/tridion/en/themasites/_disabled_image/background/regions/index-2.html

**Table S17**. Calculated premature mortality from PM with all aerosol (base mortality) and
removing ASOA, using the GEMM method.

| Location[a] | Base Mortality | Mortality reduced due to removing ASOA | Percent mortality reduced due to removing ASOA |
|---|---|---|---|
| North America | 178,793 | 24,892 | 14% |
| Central America | 58,516 | 7,298 | 12% |
| South America | 145,395 | 22,372 | 15% |
| Africa | 765,946 | 34,528 | 5% |
| Western Europe | 768,991 | 50,427 | 7% |
| Eastern Europe | 465,341 | 25,552 | 5% |
| South Asia | 2,285,903 | 166,228 | 7% |
| Southeastern Asia | 347,191 | 50,802 | 15% |
| East Asia | 2,487,349 | 220,264 | 9% |
| Oceania | 3,375 | 428 | 13% |
| Rest of the World | 269,769 | 35,051 | 13% |
| Total | 7,776,570 | 638,219 | 8% |

[a]Locations defined by:
http://themasites.pbl.nl/tridion/en/themasites/_disabled_image/background/regions/index-2.html

**Table S18**. List of total final consumption, in millions of tonnes of oil equivalent, of oil products
and oil, for each organization. Total final consumption includes imports, and does not include
exports (IEA, 2019).

| Organization | Industry | Transportation | Non-Energy |
|---|---|---|---|
| World | 307 | 2533 | 645 |
| OECD | 89 | 1147 | 326 |
| Africa | 18.4 | 115.4 | 7.9 |
| Non-OECD | 28.3 | 135 | 20 |
| Middle East | 33.5 | 126.3 | 47.5 |
| Non-OECD Europe and Eurasia | 35 | 101 | 53 |

**Supplemental figures for this study**

[Figure]

**Figure S1**. Regression plot of SOA versus HCHO from different campaigns around the world
that have not been previously published. Note, for (c), HCHO is 1.24×Hantzsch HCHO, to
account for the differences between the two HCHO measurements during CalNex. Note, for (a),
SOA is 0.5×OA, estimated from Young et al. (2015), and for (f), SOA is 0.8×OA, estimated from
DeCarlo et al. (2010).

[Figure]

**Figure S2**. Regression plot of SOA versus PAN from different campaigns around the world that have not been previously published. Note, for (a), SOA is 0.5×OA, estimated from Young et al. (2015), and for (f), SOA is 0.8×OA, estimated from DeCarlo et al. (2010).

[Figure]

**Figure S3**. Regression plot of SOA versus Ox from different campaigns around the world that
have not been previously published. Note, for (a), SOA is 0.5×OA, estimated from Young et al.
(2015).

[Figure]

**Figure S4**. Comparison of HCHO measured by the DOAS (Stutz and Platt, 1996, 1997) and
Hantzsch reaction (Cárdenas et al., 2000) methods during the CalNex 2010 study in Pasadena,
CA, ground site (Ryerson et al., 2013).

[Figure]

**Figure S5.** (a) Annually average CO emissions from HTaP. (b) Annually average benzene, toluene, and xylenes (BTX) emissions, weighted by their OH reaction rate

$$( E_{weight} = N \frac{\sum_i E_i k_{OH,i}}{\sum_i k_{OH,i}} ,\ i = B, T, X\ ;\ N{=}3 ).$$

[Figure]

**Figure S6.** Emission ratio versus saturation concentration ($\log_{10}(c^*)$) for (a) Los Angeles, (b) NE US, aircraft, (c) Beijing, and (d) London. The emission ratios for VOCs ($\log_{10}(c^*) \geq 7$) were taken from de Gouw et al. (2017) and Ma et al. (2017) for Los Angeles, Warneke et al. (2007) for NE US, aircraft, and Wang et al. (2014) for Beijing while the VOC emission ratio for London is from Table S6 to Table S8. For VOCs between $\log_{10}(c^*)$ of 3 and 6 (IVOCs), the volatility distribution from McDonald et al. (2018), along with the ratio of IVOC to BTEX from Figure SI-6 and the emission ratio of BTEX (Table S6), were used to determine the emission ratio versus saturation concentration. Finally, for VOCs between $\log_{10}(c^*)$ 0 and 2 (SVOCs), the volatility distributions from Robinson et al. (2007) for non-fossil fuel POA and from Worton et al. (2014) for fossil fuel POA were used to convert the normalized POA mass concentration (Table S9) to VOC emission ratios. Note, the emission ratio versus saturation concentration for New York City, 2015, was similar to (b), as the emissions were similar (Fig. 5) and the BTEX for New York City is the same as NE US (Table S5).

[Figure]

**Figure S7.** 2-D VBS space defined by oxygen to carbon (O:C) ratio and saturation concentration [$\log_{10}(c^*)$] for different oxidation mechanisms and primary sources of OA precursors. Dashed boxes represent primary emissions, while the full boxes represent the secondary oxidation products. (A) and (B) represent different parameterizations for treating traditional anthropogenic and biogenic sources of SOA. Both parameterizations depict the oxidation of an 8-carbon precursor VOC. (A) represents the TSI, or aging, parameterization; (B) represents the MA, or wall-loss corrected, parameterization. (C) Represents the initial oxidation and aging pathway of P-IVOCs following the ZHAO parameterization. It should be noted that the carbon number corresponds to first generation aging and subsequent oxidation results in a 0.25 reduction in carbon number. (D) Represents the decadal aging of SVOCs by hydroxyl radicals. In (D), the full aging pathway of only the C21 species is depicted as an example, though all primary species are allowed to age until the $\log_{10}(c^*)$ = -2 bin. All emitted P-SVOC species undergo the same decadal aging scheme which begins from the saturation concentration bin of the emitted species.

[Figure]

**Figure S8**. CO emissions for the cities investigated here from HTAP (Janssens-Maenhout et al., 2015).

[Figure]

**Figure S9**. (top) Total deaths associated to PM$_{2.5}$ (left) per 10×10 km$^2$ area and (right) summed up for each country, using the Integrated Exposure-Response (IER) method (Burnett et al., 2014). These values are derived from satellite. (bottom) Same as above, but using the Global Exposure Mortality Model (GEMM) (Burnett et al., 2018) for PM$_{2.5}$ per 10×10 km$^2$ area (left) and summed up for each country (right). Premature mortality was determined with PM$_{2.5}$ derived by the methods described in van Donkelaar (2015), which includes satellite and ground-based observations of aerosol.

[Figure]

**Figure S10**. Same as Fig. 8, where top are the results per 10×10 km² area for the attribution of
premature mortality to ASOA (people yr⁻¹, left) and fractional attribution of premature mortality
to ASOA for one year (right) by the IER method. See Fig. 8 for per country comparison.

[Figure]

**Figure S11.** Comparison of satellite retrieved PM$_{2.5}$ (upper left) versus modeled PM$_{2.5}$ (upper right). (Bottom) Fractional contribution of ASOA to total modeled PM$_{2.5}$.

[Figure]

**Figure S12**. Same as Fig. S10, but using the GEMM from Burnett et al. (2018). (top). (Left)
Attribution of premature mortality to ASOA per $10\times10$ km$^2$ area (people yr$^{-1}$) and (Right)
fractional attribution of preamture mortality to ASOA per $10\times10^2$ km for one year.

[Figure]

**Figure S13.** Same as Fig. S12 but summed up for each country for the (left) attribution of premature mortality to ASOA (people yr$^{-1}$) and (right) the fractional attribution of premature mortality to ASOA for one year.

[Figure]

**Figure S14**. Comparison for surface level ozone upon reducing SOA precursors by 20%.

[Figure]

[Figure]

**Figure S15.** (top) Fractional contribution of CO emissions from residential sources to total emission sources from HTaP. (bottom) Fractional contribution of BTEX emissions from residential sources to total emission sources from HTAP. Residential sources include small-scale combustion, such as heating and cooking, which may include solid-fuel emissions.